# When can Regression-Adjusted Control Variates Help?
## Rare Events, Sobolev Embedding and Minimax Optimality

**Jose Blanchet**
Department of MS&E and ICME
Stanford University
Stanford, CA 94305
`jose.blanchet@stanford.edu`

**Haoxuan Chen**
ICME
Stanford University
Stanford, CA 94305
`haoxuanc@stanford.edu`

**Yiping Lu**
Courant Institute of Mathematical Sciences
New York University
New York, NY 10012
`yiping.lu@nyu.edu`

**Lexing Ying**
Department of Mathematics and ICME
Stanford University
Stanford, CA 94305
`lexing@stanford.edu`

## Abstract

This paper studies the use of a machine learning-based estimator as a control variate for mitigating the variance of Monte Carlo sampling. Specifically, we seek to uncover the key factors that influence the efficiency of control variates in reducing variance. We examine a prototype estimation problem that involves simulating the moments of a Sobolev function based on observations obtained from (random) quadrature nodes. Firstly, we establish an information-theoretic lower bound for the problem. We then study a specific quadrature rule that employs a nonparametric regression-adjusted control variate to reduce the variance of the Monte Carlo simulation. We demonstrate that this kind of quadrature rule can improve the Monte Carlo rate and achieve the minimax optimal rate under a sufficient smoothness assumption. Due to the Sobolev Embedding Theorem, the sufficient smoothness assumption eliminates the existence of rare and extreme events. Finally, we show that, in the presence of rare and extreme events, a truncated version of the Monte Carlo algorithm can achieve the minimax optimal rate while the control variate cannot improve the convergence rate.

## 1 Introduction

In this paper, we consider a nonparametric quadrature rule on (random) quadrature points based on regression-adjusted control variate [1, 2, 3, 4]. To construct the quadrature rule, we partition our available data into two halves. The first half is used to construct a nonparametric estimator, which is then utilized as a control variate to reduce the variance of the Monte Carlo algorithm implemented over the second half of our data. Traditional and well-known results [1, Chapter 5.2] show that the optimal linear control variate can be obtained via Ordinary Least Squares regression. In this paper, we investigate a similar idea for constructing a quadrature rule [3, 5, 6, 7, 8, 9, 10], which uses a non-parametric machine learning-based estimator as a regression-adjusted control variate. We aim to answer the following two questions:

> *Is using optimal nonparametric machine learning algorithms to construct control variates an optimal way to improve Monte Carlo methods? What are the factors that determine the effectiveness of the control variate?*

37th Conference on Neural Information Processing Systems (NeurIPS 2023).

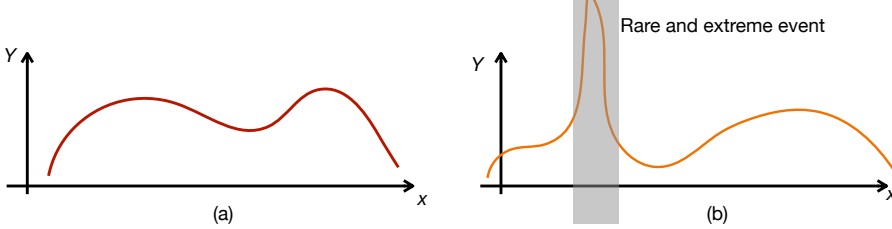

Figure 1: According to the Sobolev Embedding Theorem [11], the Sobolev space $W^{s,p}$ can be embedded in $L^{p^*}$, where $\frac{1}{p^*} = \frac{1}{p} - \frac{s}{d}$. When $s$ is large enough, as shown in (a), the smoothness assumption can rule out the existence of rare and extreme events. When $s$ is not sufficiently large, specifically $s < \frac{2dq-dp}{2pq}$, there may exist a peak (*a.k.a* rare and extreme event) that makes the Monte Carlo simulation hard. Under such circumstances, the function's $2q$-th moment is unbounded.

To understand the two questions, we consider a basic but fundamental prototype problem of estimating moments of a Sobolev function from its values observed on (random) quadrature nodes, which has a wide range of applications in Bayesian inference, the study of complex systems, computational physics, and financial risk management [1]. Specifically, we estimate the $q$-th moment $\int_\Omega f(x)^q dx$ of $f$ based on values $f(x_1), \cdots, f(x_n)$ observed on $n$ (random) quadrature nodes $x_1, \cdots, x_n \in \Omega$ for a function $f$ in the Sobolev space $W^{s,p}(\Omega)$, where $\Omega \subset \mathbb{R}^d$. The parameter $q$ here is introduced to characterize the rare events' extremeness for estimation. To verify the effectiveness of the non-parametric regression adjusted quadrature rule, we first study the statistical limit of the problem by providing a minimax information-theoretic lower bound of magnitude $n^{\max\{(\frac{1}{p} - \frac{s}{d})q - 1, -\frac{s}{d} - \frac{1}{2}\}}$.

We also provide matching upper bounds for different levels of function smoothness. Under the sufficient smoothness assumption that $s > \frac{d(2q-p)}{2pq}$, we find that the non-parametric regression adjusted control variate $\hat{f}$ can improve the rate of classical Monte Carlo algorithm and help us attain a minimax optimal upper bound. In (3.4) below, we bound variance $\int_\Omega (f^q - \hat{f}^q)^2$ of the Monte Carlo target by the sum of the semi-parametric influence part $\int_\Omega f^{2q-2}(f - \hat{f})^2$ and the propagated estimation error $\int_\Omega (f - \hat{f})^{2q}$. Although the optimal algorithm in this regime remains the same, we need to consider three different cases to derive an upper bound on the semi-parametric influence part, which is the main contribution of our proof. We propose a new proof technique that embeds the square of the influence function $(qf^{q-1})^2$ and estimation error $(f - \hat{f})^2$ in appropriate spaces via the Sobolev Embedding Theorem [11]. The two norms used for evaluating $(f^{q-1})^2$ and $(f - \hat{f})^2$ should be dual norms of each other. Also, we should select the norm for evaluating $(f - \hat{f})^2$ in a way that it's easy to estimate $f$ under the selected norm, which helps us control the error induced by $(f - \hat{f})^2$. A detailed explanation of how to select the proper norms in different cases via the Sobolev Embedding Theorem is exhibited in Figure 2. In the first regime when $s > \frac{d}{p}$, we can directly embed $f$ in $L^\infty(\Omega)$ and attain a final convergence rate of magnitude $n^{-\frac{s}{d} - \frac{1}{2}}$. For the second regime when $\frac{d(2q-p)}{p(2q-2)} < s < \frac{d}{p}$, the smoothness parameter $s$ is not large enough to ensure that $f \in L^\infty(\Omega)$. Thus, we evaluate the estimation error $(f - \hat{f})^2$ under the $L^{\frac{p}{2}}$ norm and embed the square of the influence function $(qf^{q-1})^2$ in the dual space of $L^{\frac{p}{2}}(\Omega)$. Here the validity of such embedding is ensured by the lower bound $\frac{d(2q-p)}{p(2q-2)}$ on $s$. Moreover, the semi-parametric influence part is still dominant in the second regime, so the final convergence rate is the same as that of the first case. In the third regime, when $\frac{d(2q-p)}{2pq} < s < \frac{d(2q-p)}{p(2q-2)}$, the semi-parametric influence no longer dominates and the final converge rate transits from $n^{-\frac{s}{d} - \frac{1}{2}}$ to $n^{q(\frac{1}{p} - \frac{s}{d}) - 1}$.

When the sufficient smoothness assumption breaks, *i.e.* $s < \frac{d(2q-p)}{2pq}$, according to the Sobolev Embedding Theorem [11], the Sobolev space $W^{s,p}$ is embedded in $L^{\frac{dp}{d-sp}}$ and $\frac{dp}{d-sp} < 2q$. This indicates that rare and extreme events might be present, and they are not even guaranteed to have bounded $L^{2q}$ norm, which makes the Monte Carlo estimate of the $q$-th moment have infinite variance. Under this scenario, we consider a truncated version of the Monte Carlo algorithm, which can be proved to attain the minimax optimal rate of magnitude $n^{q(\frac{1}{p} - \frac{s}{d}) - 1}$. In contrast, the usage of

regression-adjusted control variates does not improve the convergence rate under this scenario. Our results reveal how the existence of rare events will change answers to the questions raised at the beginning of the section.

We also use the estimation of a linear functional as an example to investigate the algorithm's adaptivity to the noise level. In this paper, we provide minimax lower bounds for estimating the integral of a fixed function with a general assumption on the noise level. Specifically, we consider all estimators that have access to observations $\{x_i, f(x_i) + \epsilon_i\}_{i=1}^n$ of some function $f$ that is $s$-Hölder smooth, where $x_i \overset{\text{i.i.d}}{\sim} \text{Uniform}([0,1]^d)$ and $\epsilon_i \overset{\text{i.i.d}}{\sim} n^{-\gamma}\mathcal{N}(0,1)$ for some $\gamma > 0$. Based on the method of two fuzzy hypotheses, we present a lower bound of magnitude $n^{\max\{-\frac{1}{2}-\gamma, -\frac{1}{2}-\frac{s}{d}\}}$, which exhibits a smooth transition from the Monte Carlo rate to the Quasi-Monte Carlo rate. At the same time, our information-theoretic lower bound also matches the upper bound built for quadrature rules taking use of non-parametric regression-adjusted control variates.

## 1.1 Related Work

**Regression-Adjusted Control Variate**    The control variate method is a technique used for variance reduction in Monte-Carlo simulation. Consider the task of estimating the expectation $\mathbb{E}X$ for some random variable $X$. The idea of control variate method is to introduce another random variable $Y$ correlated with the random variable $X$, such that the random variable $X - Y$ has smaller variance than $X$. Since $\mathbb{E}X = \mathbb{E}[X - Y] + \mathbb{E}[Y]$ and $\mathbb{E}[Y]$ is deterministic, one may obtain a variance reduced estimator of $\mathbb{E}[X]$ by summing up $\mathbb{E}[Y]$ and an empirical estimate of $\mathbb{E}[X - Y]$. Such a random variable $Y$ is called a control variate. Regression-adjusted control variate, in particular, refers to the case when $Y$ is obtained by applying regression methods to observed data samples of $X$.

Regression-adjusted control variates have shown both theoretical and empirical improvements in a wide range of applications, including the construction of confidence intervals [12, 13], randomized trace-estimation [14, 15], dimension reduction [16], causal inference [17], light transport simulation [18], MCMC simulation [19], estimation of the normalizing factor [10] and gradient estimation [20, 21]. It is also used as a technique for proving the approximation bounds on two-layer neural networks in the Barron space [22].

Regarding literature most related to our work, we mention [3, 7, 8, 10], which also study the theoretical properties of nonparametric control variate estimator. However, the theoretical analysis in [3, 7] does not provide a specific convergence rate in the Reproducing Kernel Hilbert Space, which requires a high level of smoothness for the underlying function. In contrast to prior work, our research delves into the effectiveness of a non-parametric regression-adjusted control variate in boosting convergence rates across various degrees of smoothness assumptions and identifies the key factor that determines the efficacy of these control variates.

**Quadrature Rule**    There is a long literature on building quadrature rules in the Reproducing Kernel Hilbert Space, including Bayes–Hermite quadrature [23, 24, 25, 26, 27], determinantal point processes [28, 29, 30, 31], Nyström approximation [32, 33], kernel herding[34, 35, 36] and kernel thinning [37, 38, 39]. Nevertheless, the quadrature points chosen in these studies all have the ability to reconstruct the function's information, which results in a suboptimal rate for estimating the moments.

**Functional Estimation**    There are also lines of research that investigated the optimal rates of estimating both linear [8, 40, 41, 42, 43, 44, 45, 46, 47, 48, 49, 50] and nonlinear [51, 52, 53, 54, 55, 56, 57, 58, 59, 60, 61, 62, 63, 64] functionals, such as integrals and the $L^q$ norm. However, as far as the authors know, previous works on this topic have assumed sufficient smoothness, which rules out the existence of rare and extreme events that are hard to simulate. Additionally, existing proof techniques are only applicable in scenarios where there is either no noise or a constant level of noise present. We have developed a novel and unified proof technique that leverages the method of two fuzzy hypotheses, which allows us to account for not only rare and extreme events but also different levels of noise.

## 1.2 Contribution

- We determine all the regimes when a quadrature rule utilizing a nonparametric estimator as a control variate to reduce the Monte Carlo estimate's variance can boost the convergence rate of estimating the moments of a Sobolev function. Under sufficient smoothness assumption,

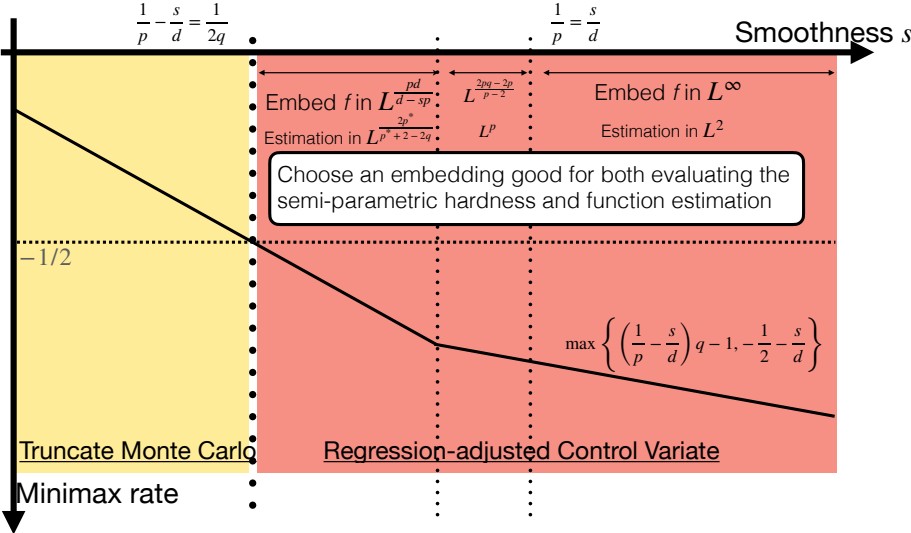

Figure 2: We summarize the minimax optimal rates and the corresponding optimal algorithms with respect to the function smoothness here. When the function is smooth enough, regression-adjusted control variates can improve the Monte Carlo rate. However, when there exist rare and extreme events that are hard to simulate, truncating the Monte Carlo estimate directly yields a minimax optimal algorithm. Above the transition point of algorithm selection is $s = \frac{d(2q-p)}{2pq}$, while the transition point of the optimal convergence rate is $s = \frac{d(2q-p)}{p(2q-2)}$. To build the optimal convergence guarantee for any algorithm that utilizes a regression-adjusted control variate $\hat{f}$, we need to embed the square of the influence function $(qf^{q-1})^2$ in an appropriate space via the Sobolev Embedding Theorem and evaluate the estimation error $(f - \hat{f})^2$ under the dual norm of the norm associated with the chosen space, which allows us to achieve optimal semi-parametric efficiency. Our selections of the metrics in different regimes are shown in this figure.

which rules out the existence of rare and extreme events due to the Sobolev Embedding Theorem, the regression-adjusted control variate improves the convergence rate and achieves the minimax optimal rate. Without the sufficient smoothness assumption, however, there may exist rare and extreme events that are hard to simulate. In this circumstance, we discover that a truncated version of the Monte Carlo method is minimax optimal, while regression-adjusted control variate can't improve the convergence rate.

- As far as the authors know, our paper is the first work considering this problem without assuming that the underlying function $f$ is uniformly bounded. All previous work assumed that $s > \frac{d}{p}$, which implies $f \in L^\infty(\Omega)$ and neglects the possibility of spike functions. As a result, they were unable to discover the transition between the two regimes described above. Under the assumption that $s > \frac{d(2q-p)}{2pq}$, the main difficulty in establishing the convergence guarantee lies in determining the right evaluation metric for function estimation. To select a suitable metric, we introduce a new proof technique by embedding the influence function into an appropriate space via the Sobolev Embedding Theorem and evaluating the function estimation in the corresponding dual norm to achieve optimal semi-parametric efficiency. Our selection of the proper embedding metrics is shown in Figure 2.

- To study how the regression adjusted control variate adapts to the noise level, we examine the linear functionals, *i.e.* the definite integral. We prove that this method is minimax optimal regardless of the level of noise present in the observed data.

## 1.3 Notations

Let $\| \cdot \|$ be the standard Euclidean norm and $\Omega = [0, 1]^d$ be the unit cube in $\mathbb{R}^d$ for any fixed $d \in \mathbb{N}$. Also, let $\mathbb{1} = \mathbb{1}\{\cdot\}$ denote the indicator function, *i.e*, for any event $A$ we have $\mathbb{1}\{A\} = 1$ if $A$ is

true and $\mathbb{1}\{A\} = 0$ otherwise. For any region $R \subseteq \Omega$, we use $V(R) := \int_\Omega \mathbb{1}\{x \in R\}dx$ to denote the volume of $R$. Let $C(\Omega)$ denote the space of all continuous functions $f : \Omega \to \mathbb{R}$ and $\lfloor \cdot \rfloor$ be the rounding function. For any $s > 0$ and $f \in C(\Omega)$, we define the Hölder norm $\| \cdot \|_{C^s(\Omega)}$ by

$$\|f\|_{C^s(\Omega)} := \max_{|k| \le \lfloor s \rfloor} \|D^k f\|_{L^\infty(\Omega)} + \max_{|k| = \lfloor s \rfloor} \sup_{x,y \in \Omega, x \neq y} \frac{|D^k f(x) - D^k f(y)|}{\|x - y\|^{s - \lfloor s \rfloor}}. \tag{1.1}$$

The corresponding Hölder space is defined as $C^s(\Omega) := \left\{ f \in C(\Omega) : \|f\|_{C^s(\Omega)} < \infty \right\}$. When $s = 0$, we have that the two norms $\| \cdot \|_{C^0(\Omega)}$ and $\| \cdot \|_{L^\infty(\Omega)}$ are equivalent and $C^0(\Omega) = L^\infty(\Omega)$. Let $\mathbb{N}_0 := \mathbb{N} \cup \{0\}$ be the set of all non-negative integers. For any $s \in \mathbb{N}_0$ and $1 \le p \le \infty$, we define the Sobolev space $W^{s,p}(\Omega)$ by

$$W^{s,p}(\Omega) := \left\{ f \in L^p(\Omega) : D^\alpha f \in L^p(\Omega), \forall \, \alpha \in \mathbb{N}_0^d \text{ satisfying } |\alpha| \le s \right\}. \tag{1.2}$$

Let $(c)_+$ denote $\max\{c, 0\}$ for any $c \in \mathbb{R}$. Fix any two non-negative sequences $\{a_n\}_{n=1}^\infty$ and $\{b_n\}_{n=1}^\infty$. We write $a_n \lesssim b_n$, or $a_n = O(b_n)$, to denote that $a_n \le Cb_n$ for some constant $C$ independent of $n$. Similarly, we write $a_n \gtrsim b_n$, or $a_n = \omega(b_n)$, to denote that $a_n \ge cb_n$ for some constant $c$ independent of $n$. We use $a_n = \Theta(b_n)$ to denote that $a_n = O(b_n)$ and $a_n = \omega(b_n)$.

# 2 Information-Theoretic Lower Bound on Moment Estimation

**Problem Setup** To understand how the non-parametric regression-adjusted control variate improves the Monte Carlo estimator's convergence rate, we consider a prototype problem that estimates a function's $q$-th moment. For any fixed $q \in \mathbb{N}$ and $f \in W^{s,p}(\Omega)$, we want to estimate the $q$-th moment $I_f^q := \int_\Omega f^q(x)dx$ with $n$ random quadrature points $\{x_i\}_{i=1}^n \subset \Omega$. On each quadrature point $x_i$ ($i = 1, \cdots, n$), we can observe the function value $y_i := f(x_i)$.

In this section, we study the information-theoretic limit for the problem above via the method of two fuzzy hypotheses [65]. We have the following information-theoretic lower bound on the class $\mathcal{H}_n^{f,q}$ that contains all estimators $\hat{H}^q : \Omega^n \times \mathbb{R}^n \to \mathbb{R}$ of the $q$-th moment $I_f^q$.

**Theorem 2.1 (Lower Bound on Estimating the Moment)** *When $p > 2$ and $q < p < 2q$, let $\mathcal{H}_n^f$ denote the class of all the estimators that use $n$ quadrature points $\{x_i\}_{i=1}^n$ and observed function values $\{y_i = f(x_i)\}_{i=1}^n$ to estimate the $q$-th moment of $f$, where $\{x_i\}_{i=1}^n$ are independently and identically sampled from the uniform distribution on $\Omega$. Then we have*

$$\inf_{\hat{H}^q \in \mathcal{H}_n^{f,q}} \sup_{f \in W^{s,p}(\Omega)} \mathbb{E}_{\{x_i\}_{i=1}^n, \{y_i\}_{i=1}^n} \left[ \left| \hat{H}^q\left(\{x_i\}_{i=1}^n, \{y_i\}_{i=1}^n\right) - I_f^q \right| \right] \gtrsim n^{\max\left\{-q\left(\frac{s}{d} - \frac{1}{p}\right) - 1, -\frac{1}{2} - \frac{s}{d}\right\}}. \tag{2.1}$$

**Proof Sketch** Here we give a sketch for our proof of Theorem 2.1. Our proof is based on the method of two fuzzy hypotheses, which is a generalization of the traditional Le Cam's two-point method. In fact, each hypothesis in the generalized method is constructed via a prior distribution. In order to attain a lower bound of magnitude $\Delta$ via the method of two fuzzy hypotheses, one needs to pick two prior distributions $\mu_0, \mu_1$ on the Sobolev space $W^{s,p}(\Omega)$ such that the following two conditions hold. Firstly, the estimators $I_f^q$ differ by $\Delta$ with constant probability under the two priors. Secondly, the TV distance between the two corresponding distributions $\mathbb{P}_0$ and $\mathbb{P}_1$ of data generated by $\mu_0$ and $\mu_1$ is of constant magnitude. In order to prove the two lower bounds given in (2.1), we pick two different pairs of prior distributions as follows:

Below we set $m = \Theta(n^{\frac{1}{d}})$ and divide the domain $\Omega$ into $m^d$ small cubes $\Omega_1, \Omega_2, \cdots, \Omega_{m^d}$, each of which has side length $m^{-1}$. For any $p \in (0, 1)$, we use $v_p, w_p$ to denote the discrete random variables satisfying $\mathbb{P}(v_p = 0) = \mathbb{P}(w_p = -1) = p$ and $\mathbb{P}(v_p = 1) = \mathbb{P}(w_p = 1) = 1 - p$.

(I) For the first lower bound in (2.1), we construct some bump function $g \in W^{s,p}(\Omega)$ satisfying $\text{supp}(g) \subseteq \Omega_1$ and $I_g^q = \int_{\Omega_1} g(x)dx = \Theta(m^{q(-s+\frac{d}{p})-d})$. Now let's take some sufficiently small constant $\epsilon \in (0, 1)$ and pick $\mu_0, \mu_1$ to be discrete measures supported on the two finite sets $\left\{v_{\frac{1+\epsilon}{2}} g\right\}$ and $\left\{v_{\frac{1-\epsilon}{2}} g\right\}$. On the one hand, the difference between the $q$-th moments under $\mu_0$ and $\mu_1$ can be

lower bounded by $\Theta(n^{q(\frac{1}{p}-\frac{s}{d})-1})$ with constant probability. On the other hand, $KL(\mathbb{P}_0\|\mathbb{P}_1)$ can be upper bounded by the KL divergence between $v_{\frac{1+\epsilon}{2}}$ and $v_{\frac{1-\epsilon}{2}}$, which is of constant magnitude.

(II) For the second lower bound in (2.1), we set $M > 0$ to be some sufficiently large constant and $\kappa = \Theta(\frac{1}{\sqrt{n}})$. For any $1 \le j \le m^d$, we construct bump functions $f_j \in W^{s,p}(\Omega)$ satisfying $\text{supp}(f_j) \subseteq \Omega_j$ and $I_{f_j}^k = \int_{\Omega_j} f_j(x)dx = \Theta(m^{-ks-d})$ for any $1 \le j \le m^d$ and $1 \le k \le s$. Now let's pick $\mu_0, \mu_1$ to be discrete measures supported on the two finite sets $\left\{ M + \sum_{j=1}^{m^d} w_j^{(0)} f_j \right\}$ and $\left\{ M + \sum_{j=1}^{m^d} w_j^{(1)} f_j \right\}$, where $\{w_j^{(0)}\}_{j=1}^{m^d}$ and $\{w_j^{(1)}\}_{j=1}^{m^d}$ are independent and identical copies of $w_{\frac{1+\kappa}{2}}$ and $w_{\frac{1-\kappa}{2}}$ respectively. On the one hand, applying Hoeffding's inequality yields that the $q$-th moments under $\mu_0$ and $\mu_1$ differ by $\Theta(n^{-\frac{s}{d}-\frac{1}{2}})$ with constant probability. On the other hand, note that $KL(\mathbb{P}_0\|\mathbb{P}_1)$ can be bounded by the KL divergence between two multivariate discrete distributions $(w_{j_1}^{(0)}, \cdots, w_{j_n}^{(0)})$ and $(w_{j_1}^{(1)}, \cdots, w_{j_n}^{(1)})$, where $\{w_{j_i}^{(0)}\}_{i=1}^n$ and $\{w_{j_i}^{(1)}\}_{i=1}^n$ are independent and identical copies of $w_{\frac{1+\kappa}{2}}$ and $w_{\frac{1-\kappa}{2}}$ respectively. Hence, $KL(\mathbb{P}_0\|\mathbb{P}_1)$ is of constant magnitude.

Combining the two cases above gives us the minimax lower bound in (2.1). We defer a complete proof of Theorem 2.1 to Appendix B.2.

# 3 Minimax Optimal Estimators for Moment Estimation

This section is devoted to constructing minimax optimal estimators of the $q$-th moment. We show that under the sufficient smoothness assumption, a regression-adjusted control variate is essential for building minimax optimal estimators. However, when the given function is not sufficiently smooth, we demonstrate that a truncated version of the Monte Carlo algorithm is minimax optimal, and control variates cannot give any improvement.

## 3.1 Sufficient Smoothness Regime: Non-parametric Regression-Adjusted Control Variate

This subsection is devoted to building a minimax optimal estimator of the $q$-th moment under the assumption that $\frac{s}{d} > \frac{1}{p} - \frac{1}{2q}$, which guarantees that functions in the space $W^{s,p}$ are sufficiently smooth. From the Sobolev Embedding theorem, we know that the sufficient smoothness assumption implies $W^{s,p}(\Omega) \subset L^{p^*}(\Omega) \subset L^{2q}(\Omega)$, where $\frac{1}{p^*} = \frac{1}{p} - \frac{s}{d}$. Given any function $f \in W^{s,p}(\Omega)$ along with $n$ uniformly sampled quadrature points $\{x_i\}_{i=1}^n$ and corresponding observations $\{y_i = f(x_i)\}_{i=1}^n$ of $f$, the key idea behind the construction of our estimator $\hat{H}_C^q$ is to build a nonparametric estimation $\hat{f}$ of $f$ based on a sub-dataset and use $\hat{f}$ as a control variate for Monte Carlo simulation. Consequently, it takes three steps to compute the numerical estimation of $I_f^q$ for any estimator $\hat{H}_C^q : \Omega^n \times \mathbb{R}^n \to \mathbb{R}$. The first step is to divide the observed data into two subsets $\mathcal{S}_1 := \{(x_i, y_i)\}_{i=1}^{\frac{n}{2}}, \mathcal{S}_2 := \{(x_i, y_i)\}_{i=\frac{n}{2}+1}^n$ of equal size and use a machine learning algorithm to compute a nonparametric estimation $\hat{f}_{1:\frac{n}{2}}$ of $f$ based on $\mathcal{S}_1$. Without loss of generality, we may assume that the number of data points is even. Secondly, we treat $\hat{f}_{1:\frac{n}{2}}$ as a control variate and compute the $q$-th moment $I_{\hat{f}}^q$. Using the other dataset $\mathcal{S}_2$, we may obtain a Monte Carlo estimate of $I_f^q - I_{\hat{f}_{1:\frac{n}{2}}}^q$ as follows: $I_f^q - I_{\hat{f}_{1:\frac{n}{2}}}^q \approx \frac{2}{n} \sum_{i=\frac{n}{2}+1}^n \left( y_i^q - \hat{f}_{1:\frac{n}{2}}^q(x_i) \right)$. Finally, combining the estimation of the $q$-th moment $I_{\hat{f}_{1:\frac{n}{2}}}^q = \int_\Omega \hat{f}_{1:\frac{n}{2}}^q(x)dx$ with the estimation of $I_f^q - I_{\hat{f}_{1:\frac{n}{2}}}^q$ gives us the numerical estimation returned by $\hat{H}_C^q$:

$$\hat{H}_C^q\left(\{x_i\}_{i=1}^n, \{y_i\}_{i=1}^n\right) := \int_\Omega \hat{f}_{1:\frac{n}{2}}^q(x)dx + \frac{2}{n} \sum_{i=\frac{n}{2}+1}^n \left( y_i^q - \hat{f}_{1:\frac{n}{2}}^q(x_i) \right). \qquad (3.1)$$

We assume that our function estimation $\hat{f}$ is obtained from an $\frac{n}{2}$-oracle $K_{\frac{n}{2}} : \Omega^{\frac{n}{2}} \times \mathbb{R}^{\frac{n}{2}} \to W^{s,p}(\Omega)$ satisfying Assumption 3.1. For example, there are lines of research [49, 50, 59, 60, 61] considering how the moving least squares method [66, 67] can achieve the convergence rate in (3.2).

**Assumption 3.1 (Optimal Function Estimator as an Oracle)** *Given any function $f \in W^{s,p}(\Omega)$ and $n \in \mathbb{N}$, let $\{x_i\}_{i=1}^n$ be $n$ data points sampled independently and identically from the uniform distribution on $\Omega$. Assume that for $s > \frac{2dq - dp}{2pq}$, there exists an oracle $K_n : \Omega^n \times \mathbb{R}^n \to W^{s,p}(\Omega)$ that estimates $f$ based on the $n$ points $\{x_i\}_{i=1}^n$ along with the $n$ observed function values $\{f(x_i)\}_{i=1}^n$ and satisfies the following bound for any $r$ satisfying $\frac{1}{r} \in \left( \max\{\frac{d-sp}{pd}, 0\}, \max\{\frac{1}{p}, \mathbb{1}\{s > \frac{d}{p}\}\} \right]$:*

$$\left( \mathbb{E}_{\{x_i\}_{i=1}^n} \left[ \|K_n(\{x_i\}_{i=1}^n, \{f(x_i)\}_{i=1}^n) - f\|_{L^r(\Omega)}^r \right] \right)^{\frac{1}{r}} \lesssim n^{-\frac{s}{d} + (\frac{1}{p} - \frac{1}{r})_+}. \tag{3.2}$$

A construction of the desired oracle and a complete proof of the upper bound above (up to logarithm factors) is deferred to Appendix E. Based on the oracle above, we can obtain the following upper bound that matches the information-theoretic lower bound in Theorem 2.1.

**Theorem 3.1 (Upper Bound on Moment Estimation with Sufficient Smoothness)** *Assume that $p > 2$, $q < p < 2q$ and $s > \frac{2dq - dp}{2pq}$. Let $\{x_i\}_{i=1}^n$ be $n$ quadrature points independently and identically sampled from the uniform distribution on $\Omega$ and $\{y_i := f(x_i)\}_{i=1}^n$ be the corresponding $n$ observations of $f \in W^{s,p}(\Omega)$. Then the estimator $\hat{H}_C^q$ constructed in (3.1) above satisfies*

$$\mathbb{E}_{\{x_i\}_{i=1}^n, \{y_i\}_{i=1}^n} \left[ \left| \hat{H}_C^q \left( \{x_i\}_{i=1}^n, \{y_i\}_{i=1}^n \right) - I_f^q \right| \right] \lesssim n^{\max\{-q(\frac{s}{d} - \frac{1}{p}) - 1, -\frac{s}{d} - \frac{1}{2}\}}, \tag{3.3}$$

**Proof Sketch**   Given a non-parametric estimator $\hat{f}$ of the function $f$, we may bound the variance of the Monte Carlo process by $(f^q - \hat{f}^q)^2$ and further upper bound it by the sum of the following two terms:

$$|f^q - \hat{f}^q|^2 \lesssim \underbrace{|f^{q-1}(f - \hat{f})|^2}_{\text{semi-parametric influnce}} + \underbrace{|(f - \hat{f})^q|^2}_{\text{estimation error propagation}}. \tag{3.4}$$

The first term above represents the semi-parametric influence part of the problem, as $qf^{q-1}$ is the influence function for the estimation of the $q$-th moment $f^q$. The second term characterizes how function estimation affects functional estimation. If we consider the special case of estimating the mean instead of a general $q$-th moment, *i.e*, $q = 1$, the semi-parametric influence term will disappear. Consequently, the convergence rate won't transit from $n^{-\frac{1}{2} - \frac{s}{d}}$ to $n^{-q(\frac{s}{d} - \frac{1}{p}) - 1}$ in the special case.

Although the algorithm remains unchanged in the sufficient smooth regime, we need to consider three separate cases to obtain an upper bound on the integral of the semi-parametric influence term $|f^{q-1}(f - \hat{f})|^2$ in (3.4). An illustration of the three cases is given in Figure 2.

From Hölder's inequality, we know that $\int_\Omega f^{2q-2}(x)(f(x) - \hat{f}(x))^2 dx$ can be upper bounded by $\|f^{2q-2}\|_{L^{r'}(\Omega)} \|(f - \hat{f})^2\|_{L^{r*}(\Omega)}$, where $\|\cdot\|_{L^{r'}(\Omega)}$ and $\|\|_{L^{r*}(\Omega)}$ are dual norms. Therefore, the main difficulty here is to embed the function $f$ in different spaces via the Sobolev Embedding Theorem under different assumptions on the smoothness parameter $s$. When the function is smooth enough, *i.e.* $s > \frac{d}{p}$, we embed the function $f$ in $L^\infty(\Omega)$ and evaluate the estimation error $f - \hat{f}$ under the $L^2$ norm. Then our assumption on the oracle (3.2) gives us an upper bound of magnitude $n^{-\frac{2s}{d}}$ on $\|f - \hat{f}\|_{L^2(\Omega)}^2$, which helps us further upper bound the semi-parametric influence part $\int_\Omega f^{2q-2}(x)(f(x) - \hat{f}(x))^2 dx$ by $n^{-\frac{2s}{d}}$ up to constants. When $\frac{d(2q-p)}{p(2q-2)} < s < \frac{d}{p}$, we embed the function $f$ in $L^{\frac{2pq-2p}{p-2}}(\Omega) \subseteq L^{\frac{pd}{d-sp}}(\Omega)$ and evaluate the estimation error $f - \hat{f}$ under the $L^p$ norm. Applying our assumption on the oracle (3.2) again implies that the semi-parametric influence part $\int_\Omega f^{2q-2}(x)(f(x) - \hat{f}(x))^2 dx$ can be upper bounded by $n^{-\frac{2s}{d}}$ up to constants. When $\frac{d(2q-p)}{2pq} < s < \frac{d(2q-p)}{p(2q-2)}$, we embed the function $f$ in $L^{p^*}$ and evaluate the error of the oracle in $L^{\frac{2p^*}{p^*+2-2q}}$, where $\frac{1}{p^*} = \frac{1}{p} - \frac{s}{d}$. Similarly, we can use (3.2) to upper bound the semi-parametric influence part $\int_{x \in \Omega} f^{2q-2}(x)(f(x) - \hat{f}(x))^2 dx$ by $n^{2q(\frac{1}{p} - \frac{s}{d}) - 1}$.

The upper bound on the propagated estimation error $\int_{x \in \Omega} (f(x) - \hat{f}(x))^{2q} dx$ in (3.4) can be derived by evaluating the error of the oracle under the $L^{2q}$ norm. *i.e*, by picking $r = 2q$ in (3.2) above, which yields an upper bound of magnitude $n^{2q(\frac{1}{p} - \frac{s}{d}) - 1}$.

The obtained upper bounds on the semi-parametric influence part and the propagated estimation error above provide us with a clear view of the upper bound on the variance of $f^q - \hat{f}^q$, which is the random variable we aim to simulate via Monte-Carlo in the second stage. Using the standard Monte-Carlo algorithm to simulate the expectation of $f^q - \hat{f}^q$ then gives us an extra $n^{-\frac{1}{2}}$ factor for the convergence rate, which helps us attain the final upper bounds given in (3.3). A complete proof of Theorem 3.1 is given in Appendix C.1.

## 3.2 Beyond the Sufficient Smoothness Regime: Truncated Monte Carlo

In this subsection, we study the case when the sufficient smoothness assumption breaks, *i.e.* $\frac{s}{d} < \frac{1}{p} - \frac{1}{2q}$. According to the Sobolev Embedding theorem, we have that $W_s^p$ is embedded in $L^{\frac{dp}{d-sp}}$. Since $\frac{1}{p} - \frac{s}{d} > \frac{1}{2q}$ implies $\frac{dp}{d-sp} < 2q$, the underlying function $f$ is not guaranteed to have bounded $L^{2q}$ norm, which indicates the existence of rare and extreme events. Consequently, the Monte Carlo estimate of $f$'s $q$-th moment must have infinite variance, which makes it hard to simulate. Here we present a truncated version of the Monte Carlo algorithm that can achieve the minimax optimal convergence rate. For any fixed parameter $M > 0$, our estimator is designed as follows:

$$\hat{H}_M^q\left(\{x_i\}_{i=1}^n, \{y_i\}_{i=1}^n\right) := \frac{1}{n}\sum_{i=1}^n \max\left\{\min\{y_i, M\}, -M\right\}^q. \tag{3.5}$$

In Theorem 3.2, we provide the convergence rate of the estimator (3.5) by choosing the truncation parameter $M$ in an optimal way.

**Theorem 3.2 (Upper Bound on Moment Estimation without Sufficient Smoothness)** *Assuming that $p > 2$, $q < p < 2q$ and $s < \frac{2dq-dp}{2pq}$, we pick $M = \Theta(n^{\frac{1}{p} - \frac{s}{d}})$. Let $\{x_i\}_{i=1}^n$ be $n$ quadrature points independently and identically sampled from the uniform distribution on $\Omega$ and $\{y_i := f(x_i)\}_{i=1}^n$ be the corresponding $n$ observations of $f \in W^{s,p}(\Omega)$. Then we have that the estimator $\hat{H}_M^q$ constructed in (3.5) above satisfies*

$$\mathbb{E}_{\{x_i\}_{i=1}^n, \{y_i\}_{i=1}^n}\left[\left|\hat{H}_M^q\left(\{x_i\}_{i=1}^n, \{y_i\}_{i=1}^n\right) - I_f^q\right|\right] \lesssim n^{-q(\frac{s}{d} - \frac{1}{p}) - 1}. \tag{3.6}$$

**Proof Sketch** The error can be decomposed into bias and variance parts. The bias part is caused by the truncation in our algorithm, which is controlled by the parameter $M$ and can be bounded by $\int_{\{x:|f(x)|>M\}} |f|^q dx$. According to the Sobolev Embedding Theorem, $W^{s,p}(\Omega)$ can be embedded in the space $L^{p^*}$, where $\frac{1}{p^*} = \frac{1}{p} - \frac{s}{d}$. As $|f(x)| > M$ implies $|f(x)|^q < M^{q-p^*}|f(x)|^{p^*}$, the bias can be upper bounded by $M^{q-p^*}$. Similarly, the variance is controlled by $M$ and can be upper bounded by $M^{q-\frac{p^*}{2}}$. Combining the bias and variance bound, we can bound the final error as $M^{q-p^*} + \frac{M^{q-\frac{p^*}{2}}}{\sqrt{n}}$. By selecting $M = \Theta(n^{\frac{1}{p^*}}) = \Theta(n^{\frac{1}{p} - \frac{s}{d}})$, we obtain the final convergence rate $n^{-q(\frac{s}{d} - \frac{1}{p}) - 1}$. A complete proof of Theorem 3.2 is given in Appendix C.2.

**Remark 3.1** *[64] has shown that the convergence rate of the optimal non-parametric regression-based estimation is $n^{-\frac{s}{d} + \frac{1}{p} - \frac{1}{q}}$, which is slower than the convergence rate of the truncated Monte Carlo estimator that we show above.*

# 4 Adapting to the Noise Level: a Case Study for Linear Functional

In this section, we study how the regression-adjusted control variate adapts to different noise levels. Here we consider the linear functional, *i.e.* estimating a function's definite integral via low-noise observations at random points.

**Problem Setup** We consider estimating $I_f = \int_\Omega f(x)dx$, the integral of $f$ over $\Omega$, for a fixed function $f \in C^s(\Omega)$ with uniformly sampled quadrature points $\{x_i\}_{i=1}^n \subset \Omega$. On each quadrature point $x_i$ $(i = 1, \cdots, n)$, we have a noisy observation $y_i := f(x_i) + \epsilon_i$. Here the $\epsilon_i$'s are independently and identically distributed Gaussian noises sampled from $\mathcal{N}(0, n^{-2\gamma})$, where $\gamma \in [0, \infty]$.

## 4.1 Information-Theoretic Lower Bound on Mean Estimation

In this subsection, we present a minimax lower bound (Theorem 4.1) for all estimators $\hat{H} : \Omega^n \times \mathbb{R}^n \to \mathbb{R}$ of the integral $I_f$ of a function $f \in C^s(\Omega)$ when one can only access noisy observations.

**Theorem 4.1 (Lower Bound for Integral Estimation)** *Let $\mathcal{H}_n^f$ denote the class of all the estimators that use $n$ quadrature points $\{x_i\}_{i=1}^n$ and noisy observations $\{y_i = f(x_i) + \epsilon_i\}_{i=1}^n$ to estimate the integral of $f$, where $\{x_i\}_{i=1}^n$ and $\{\epsilon_i\}_{i=1}^n$ are independently and identically sampled from the uniform distribution on $\Omega$ and the normal distribution $\mathcal{N}(0, n^{-2\gamma})$ respectively. Assuming that $\gamma \in [0, \infty]$ and $s > 0$, we have*

$$\inf_{\hat{H} \in \mathcal{H}_n^f} \sup_{f \in C^s(\Omega)} \mathbb{E}_{\{x_i\}_{i=1}^n, \{y_i\}_{i=1}^n} \left[ \left| \hat{H}\left( \{x_i\}_{i=1}^n, \{y_i\}_{i=1}^n \right) - I_f \right| \right] \gtrsim n^{\max\{-\frac{1}{2}-\gamma, -\frac{1}{2}-\frac{s}{d}\}}. \quad (4.1)$$

**Remark 4.1** *Functional estimation is a well-studied problem in the literature of nonparametric statistics. However, current information-theoretic lower bounds for functional estimation [51, 52, 53, 56, 57, 58, 65, 68] assume a constant level of noise on the observed function values. One essential idea for proving these lower bounds is to leverage the existence of the observational noise, which enables us to upper bound the amount of information required to distinguish between two reduced hypotheses. In contrast, we provide a minimax lower bound that is applicable for noises at any level by constructing two priors with overlapping support and assigning distinct probabilities to the corresponding Bernoulli random variables, which separates the two hypotheses. A comprehensive proof of Theorem 4.1 is given in Appendix D.2.*

## 4.2 Optimal Nonparametric Regression-Adjusted Quadrature Rule

In the discussion below, we use the nearest-neighbor method as an example. For any $k \in \{1, 2, \cdots, \frac{n}{2}\}$, the $k$-nearest neighbor estimator $\hat{f}_{k\text{-NN}}$ of $f$ is given by $\hat{f}_{k\text{-NN}}(z) := \frac{1}{k} \sum_{j=1}^k y_{i_j^{(z)}}$, where $\{x_{i_j^{(z)}}\}_{j=1}^{\frac{n}{2}}$ is a permutation of the quadrature points $\{x_i\}_{i=1}^{\frac{n}{2}}$ such that $\|x_{i_1^{(z)}} - z\| \leq \|x_{i_2^{(z)}} - z\| \leq \cdots \leq \|x_{i_{\frac{n}{2}}^{(z)}} - z\|$ holds for any $z \in \Omega$. Moreover, we use $\mathcal{T}_{k,z} := \{x_{i_j^{(z)}}\}_{j=1}^k$ to denote the collection of the $k$ nearest neighbors of $z$ among $\{x_i\}_{i=1}^{\frac{n}{2}}$ for any $z \in \Omega$. For any $1 \leq i \leq \frac{n}{2}$, we take $D_i \subset \Omega$ to be the region formed by all the points whose $k$ nearest neighbors contain $x_i$, i.e, $D_i := \left\{ z \in \Omega : x_i \in \mathcal{T}_{k,z} \right\}$. Our estimator $\hat{H}_{k\text{-NN}}$ can be formally represented as

$$\hat{H}_{k\text{-NN}}\left( \{x_i\}_{i=1}^n, \{y_i\}_{i=1}^n \right) = \underbrace{\sum_{i=1}^{\frac{n}{2}} \frac{V(D_i)}{k} y_i}_{\int_\Omega \hat{f}_{k\text{-NN}}(x)dx} + \underbrace{\frac{2}{n} \sum_{i=\frac{n}{2}+1}^n y_i - \frac{2}{n} \sum_{i=\frac{n}{2}+1}^n \left( \frac{1}{k} \sum_{j=1}^{\frac{n}{2}} \mathbb{1}\{x_i \in D_j\} y_j \right)}_{\frac{2}{n} \sum_{i=\frac{n}{2}+1}^n \left( y_i - \hat{f}_{k\text{-NN}}(x_i) \right)}.$$

In the following theorem, we present an upper bound on the expected risk of the estimator $\hat{H}_{k\text{-NN}}$:

**Theorem 4.2 (Matching Upper Bound for Integral Estimation)** *Let $\{x_i\}_{i=1}^n$ be $n$ quadrature points independently and identically sampled from the uniform distribution on $\Omega$ and $\{y_i := f(x_i) + \epsilon_i\}_{i=1}^n$ be the corresponding $n$ noisy observations of $f \in C^s(\Omega)$, where $\{\epsilon_i\}_{i=1}^n$ are independently and identically sampled from the normal distribution $\mathcal{N}(0, n^{-2\gamma})$. Assuming that $\gamma \in [0, \infty]$ and $s \in (0, 1)$, we have that there exists $k \in \mathbb{N}$ such that the estimator $\hat{H}_{k\text{-NN}}$ constructed above satisfies*

$$\mathbb{E}_{\{x_i\}_{i=1}^n, \{y_i\}_{i=1}^n} \left[ \left| \hat{H}_{k\text{-NN}}\left( \{x_i\}_{i=1}^n, \{y_i\}_{i=1}^n \right) - I_f \right| \right] \lesssim n^{\max\{-\frac{1}{2}-\gamma, -\frac{1}{2}-\frac{s}{d}\}}. \quad (4.2)$$

**Remark 4.2** *Our upper bound in Theorem 4.2 matches our minimax lower bound in Theorem 4.1, which indicates that the regression-adjusted quadrature rule associated with the nearest neighbor estimator is minimax optimal. When the noise level is high ($\gamma < \frac{s}{d}$), the control variate helps*

to improve the rate from $n^{-\frac{1}{2}}$ (the Monte Carlo rate) to $n^{-\frac{1}{2}-\gamma}$ via eliminating **all** the effects of simulating the smooth function. When the noise level is low ($\gamma > \frac{s}{d}$), we show that our estimator $\hat{H}_{k\text{-}NN}$ can achieve the optimal rate of quadrature rules [46]. We defer a complete proof of Theorem 4.2 to Appendix D.3.

## 5 Discussion and Conclusion

In this paper, we have investigated whether a non-parametric regression-adjusted control variate can improve the rate of estimating functionals and its minimax optimality. Using the Sobolev Embedding Theorem, we discover that the existence of infinite variance rare and extreme events will change the answer to this question. We show that when infinite variance rare and extreme events are present, using a non-parametric machine learning algorithm as a control variate does not help to improve the convergence rate, and truncated Monte Carlo is minimax optimal. When the variance of the simulation problem is finite, using a regression-adjusted control variate via an optimal non-parametric estimator is minimax optimal.

The assumptions we made in this paper, such as boundedness of the domain $\Omega$ and constraints on the parameters $p, q$, might be too restrictive for some application scenarios. We left relaxations of these assumptions as future work. One other potential direction is to investigate how to combine importance sampling with regression-adjusted control variates. Also, the study of how regression-adjusted control variates adapt to the noise level for non-linear functionals [62, 63] may be of interest. Moreover, another intriguing project is to analyze how the data distribution's information [3, 7] can be used to achieve both better computational trackability and convergence rate [8].

## Acknowledgments and Disclosure of Funding

Jose Blanchet is supported in part by the Air Force Office of Scientific Research (AFOSR) under award number FA9550-20-1-0397 and the National Science Foundation (NSF) under award number DMS-1915967. Haoxuan Chen is supported by the T. S. Lo Graduate Fellowship Fund. Yiping Lu is supported by the Stanford Interdisciplinary Graduate Fellowship (SIGF). Lexing Ying is supported by the National Science Foundation (NSF) under award number DMS-2011699 and DMS-2208163.

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
