## Appendix

The appendix is organized as follows:

- In Appendix A, we list some notations and standard lemmas used in our proofs.
- Appendix B contains a comprehensive proof of the information-theoretic lower bound on the estimation of $q$-th moments, which is established in Theorem 2.1.
- In Appendix C, we provide a detailed proof of Theorem 3.1 and 3.2, which gives us the minimax optimal upper bound on estimating $q$-th moments.
- Appendix D consists of our proof for the information-theoretic lower bounds and minimax optimal upper bounds on integral estimation and function estimation, which are listed in Theorem 4.1 and 4.2.
- Appendix E provides our construction of the desired function estimator in Assumption 3.1 along with a proof of its convergence rate.

## A  Preliminaries and Basic Tools

### A.1  Preliminaries

This subsection is devoted to presenting some basic notations used in our proofs. For any fixed convex function $f : \mathbb{R}^+ \to \mathbb{R}$ satisfying $f(1) = 0$, we use $D_f(\cdot \| \cdot)$ to denote the corresponding $f$-divergence, *i.e,* $D_f(P\|Q) = \int_{\mathcal{Y}} f\left(\frac{dP}{dQ}\right) dQ$ for any two probability distributions $P$ and $Q$ over some fixed space $\mathcal{Y}$. In particular, when $f(x) = \frac{1}{2}|x-1|$, $D_f(\cdot \| \cdot)$ is the total variation (TV) distance $TV(\cdot \| \cdot)$. When $f(x) = x \log x$, $D_f(\cdot \| \cdot)$ coincides with the Kullback–Leibler (KL) divergence $KL(\cdot \| \cdot)$. Moreover, for any $a \in \mathbb{R}$, we use $\delta_a(\cdot)$ to denote the Dirac delta distribution at point $a$, *i.e,* $\int_{-\infty}^{\infty} f(x)\delta_a(x) = f(a)$ for any function $f : \mathbb{R} \to \mathbb{R}$.

### A.2  Basic Lemmas

In this subsection, we list some basic lemmas that serve as essential tools in our proofs.

**Lemma 1 (Sobolev Embedding Theorem [11])** *For some fixed dimension $d \in \mathbb{N}$, we have that (I) For any $s, t \in \mathbb{N}_0$ and $p, q \in \mathbb{R}$ satisfying $s > t$, $p < d$ and $1 \le p < q \le \infty$, we have $W^{s,p}(\mathbb{R}^d) \subseteq W^{t,q}(\mathbb{R}^d)$ when the relation $\frac{1}{p} - \frac{s}{d} = \frac{1}{q} - \frac{t}{d}$ holds. In the special case when $t = 0$, we have $W^{s,p}(\mathbb{R}^d) \subseteq L^q(\mathbb{R}^d)$ for any $s \in \mathbb{N}$ and $p, q \in \mathbb{R}$ satisfying $1 \le p < q \le \infty$ and $\frac{1}{p} - \frac{s}{d} \le \frac{1}{q}$. (II) For any $\alpha \in (0, 1)$, let $\beta = \frac{d}{1-\alpha} \in (d, \infty]$. Then we have $C^1(\mathbb{R}^d) \cap W^{1,\beta}(\mathbb{R}^d) \subseteq C^\alpha(\mathbb{R}^d)$.*

**Lemma 2 (Hölder's Inequality)** *For any fixed domain $\Omega$ and $p, q \in [1, \infty]$ satisfying $\frac{1}{p} + \frac{1}{q} = 1$, we have that $\|fg\|_{L^1(\Omega)} \le \|f\|_{L^p(\Omega)} \|g\|_{L^q(\Omega)}$ holds for any $f \in L^p(\Omega), g \in L^q(\Omega)$.*

**Lemma 3 (Hoeffding's Inequality)** *Let $X_1, X_2, \cdots, X_n$ be independent random variables satisfying $X_i \in [a_i, b_i]$ for any $1 \le i \le n$. Then for any $\epsilon > 0$, the sum $S_n := \sum_{i=1}^{n} X_i$ of these $n$ random variables satisfies the following inequality:*

$$
\begin{aligned}
\mathbb{P}(S_n \ge \mathbb{E}[S_n] + t) &\le \exp\left(-\frac{2t^2}{\sum_{i=1}^n (b_i - a_i)^2}\right) \\
\mathbb{P}(S_n \le \mathbb{E}[S_n] - t) &\le \exp\left(-\frac{2t^2}{\sum_{i=1}^n (b_i - a_i)^2}\right)
\end{aligned} \tag{A.1}
$$

**Lemma 4 (Data Processing Inequality)** *Given some Markov Chain $X \to Z$, where $X$ and $Z$ are two random variables the measurable spaces $(\mathcal{X}, \mu)$ and $(\mathcal{Z}, \nu)$ respectively. Let $K$ be the transition kernel of the Markov Chain above, i.e, for any $x \in \mathcal{X}$, the probability distribution of $Z$ is given by $K(\cdot, x)$ when conditioned on $X = x$. For any two fixed two distributions $P, Q$ over $\mathcal{X}$ with probability density functions $p, q$, we use $K_P(\cdot)$ and $K_Q(\cdot)$ to denote the corresponding marginal distributions respectively, i.e, $K_P(\cdot) := \int_{\mathcal{X}} K(\cdot, x)p(x)d\mu(x)$ and $K_Q(\cdot) = \int_{\mathcal{X}} K(\cdot, x)q(x)d\mu(x)$. Then we have $D_f(K_P\|K_Q) \le D_f(P\|Q)$ holds for any $f$-divergence $D_f(\cdot \| \cdot)$.*

# B   Proof of Lower Bounds in Section 2

## B.1   A Key Lemma for Building Minimax Optimal Lower Bounds

In this subsection, we firstly present the method of two fuzzy hypotheses, which turns out to be the most essential tool for establishing all the minimax optimal lower bounds in our paper, before giving our complete proof of Theorem 2.1.

**Lemma 5 (Method of Two Fuzzy Hypotheses: Theorem 2.15 (i), [65])** *Let $F : \Theta \to \mathbb{R}$ be some continuous functional defined on the measurable space $(\Theta, \mathcal{U})$ and taking values in $(\mathbb{R}, \mathcal{B}(\mathbb{R}))$, where $\mathcal{B}(\mathbb{R})$ denotes the Borel $\sigma$-algebra on $\mathbb{R}$. Suppose that each parameter $\theta \in \Theta$ is associated with a distribution $\mathbb{P}_\theta$, which together form a collection $\{\mathbb{P}_\theta : \theta \in \Theta\}$ of distributions.*

*For any fixed $\theta \in \Theta$, assume that our observation $\mathbf{X}$ is distributed as $\mathbb{P}_\theta$. Let $\hat{F}$ be an arbitrary estimator of $F(\theta)$ based on $\mathbf{X}$. Let $\mu_0, \mu_1$ be two prior measures on $\Theta$. Assume that there exist constants $c \in \mathbb{R}, \Delta \in (0, \infty)$ and $\beta_0, \beta_1 \in [0, 1)$, such that:*

$$
\begin{aligned}
\mu_0(\theta \in \Theta : F(\theta) \leq c - \Delta) &\geq 1 - \beta_0, \\
\mu_1(\theta \in \Theta : F(\theta) \geq c + \Delta) &\geq 1 - \beta_1.
\end{aligned}
\tag{B.1}
$$

*For $j \in \{0, 1\}$, we use $\mathbb{P}_j(\cdot) := \int \mathbb{P}_\theta(\cdot)\mu_j(d\theta)$ to denote the marginal distribution $\mathbb{P}_j$ associated with the prior distribution $\mu_j$. Then we have the following lower bound:*

$$
\inf_{\hat{F}} \sup_{\theta \in \Theta} \mathbb{P}_\theta(|\hat{F} - F(\theta)| \geq \Delta) \geq \frac{1 - TV(\mathbb{P}_0 || \mathbb{P}_1) - \beta_0 - \beta_1}{2}.
\tag{B.2}
$$

## B.2   Proof of Theorem 2.1 (Information-Theoretic Lower Bound on Moment Estimation)

In this subsection, we give a detailed proof of the two minimax lower bounds established in Theorem 2.1 above via the method of two fuzzy hypotheses (Lemma 5). We start off by introducing some preliminary tools used in our proof. Consider the function $K_0$ defined as follows:

$$
K_0(x) := \prod_{i=1}^{d} \exp\left(-\frac{1}{1 - x_i^2}\right) \mathbb{1}(|x_i| \leq 1), \forall\, x = (x_1, x_2, \cdots, x_d) \in \mathbb{R}^d.
\tag{B.3}
$$

Moreover, we pick some function $K$ satisfying

$$
K(x) := K_0(2x), \ \forall\, x \in \mathbb{R}^d,
\tag{B.4}
$$

From our construction of $K$ and $K_0$ above, we have that $K_0$ is in $C^\infty(\mathbb{R}^d)$ and compactly supported on $[-\frac{1}{2}, \frac{1}{2}]^d$. Furthermore, we set $m = (200n)^{\frac{1}{d}}$ and divide the domain $\Omega$ into $m^d$ small cubes $\Omega_1, \Omega_2, \cdots, \Omega_{m^d}$, each of which has side length $m^{-1}$. For any $1 \leq j \leq m^d$, we use $c_j$ to denote the center of the cube $\Omega_j$. Similar to the proof sketch of Theorem 2.1, below we again use $w_p$ to denote the discrete random variable satisfying $\mathbb{P}(w_p = -1) = p$ and $\mathbb{P}(w_p = 1) = 1 - p$ for any $p \in (0, 1)$. Furthermore, we use $\vec{x} := (x_1, x_2, \cdots, x_n)$ and $\vec{y} := (y_1, y_2, \cdots, y_n)$ to denote the two $n$-dimensional vectors formed by the quadrature points and observed function values, After introducing all preliminaries above, let's present the essential parts of our proof. Given that our lower bound in Theorem 2.1 consists of two terms, our proof is also divided into two parts:

(Case I) For the first lower bound in (2.1), let's consider two functions $g_0$ and $g_1$ defined as follows:

$$
\begin{aligned}
g_0(x) &\equiv 0 \ (\forall\, x \in \Omega), \\
g_1(x) &= \begin{cases} m^{-s+\frac{d}{p}} K(m(x - c_1)) \ (x \in \Omega_1), \\ 0 \ (\text{otherwise}). \end{cases}
\end{aligned}
\tag{B.5}
$$

Clearly we have $g_0 \in W^{s,p}(\Omega)$ and $I_{g_0}^q = 0$. Now let's verify that $g_1 \in W^{s,p}(\Omega)$ for any $m$. Note that the following bound holds for any $t \in \mathbb{N}_0^d$ satisfying $|t| \leq s$:

$$
\begin{aligned}
\|D^t g_1\|_{L^p(\Omega)} &= \left(\int_{\Omega_1} \left| m^{-s+\frac{d}{p}} m^{|t|}(D^t K)(m(x - c_1)) \right|^p dx\right)^{\frac{1}{p}} \\
&= m^{|t|-s+\frac{d}{p}} \left(\int_{[-\frac{1}{2}, \frac{1}{2}]^d} \left|(D^t K)(y)\right|^p \frac{1}{m^d} dy\right)^{\frac{1}{p}} = m^{|t|-s} \|D^t K\|_{L^p([-\frac{1}{2}, \frac{1}{2}]^d)} \lesssim 1.
\end{aligned}
$$

This implies $g_1 \in W^{s,p}(\Omega)$ for any $m$, as desired. Moreover, computing the $q$-th moment of $g_1$ yields

$$
\begin{aligned}
I_{g_1}^q &= \int_\Omega g_1^q(x)dx = \int_{\Omega_1} (m^{-s+\frac{d}{p}}K(m(x-c_1)))^q dx \\
&= m^{-q(s-\frac{d}{p})} \int_{[-\frac{1}{2},\frac{1}{2}]^d} (K(y))^q \frac{1}{m^d} dy = m^{-q(s-\frac{d}{p})-d} \|K\|_{L^q([-\frac{1}{2},\frac{1}{2}]^d)}^q.
\end{aligned}
\tag{B.6}
$$

Now let us take $\epsilon = \frac{1}{2}$ and pick two discrete measures $\mu_0, \mu_1$ supported on the finite set $\{g_0, g_1\} \subset W^{s,p}(\Omega)$ as below:

$$
\begin{aligned}
\mu_0(\{g_0\}) = \frac{1+\epsilon}{2}, \mu_0(\{g_1\}) = \frac{1-\epsilon}{2}, \\
\mu_1(\{g_0\}) = \frac{1-\epsilon}{2}, \mu_1(\{g_1\}) = \frac{1+\epsilon}{2}.
\end{aligned}
\tag{B.7}
$$

On the one hand, by taking $c = \Delta = \frac{1}{2}I_{g_1}^q$ and $\beta_0 = \beta_1 = \frac{1-\epsilon}{2}$, we may use (B.7) to deduce that

$$
\begin{aligned}
\mu_0(f \in W^{s,p}(\Omega) : I_f^q \le c - \Delta) = \mu_0(I_f^q \le 0) \ge \frac{1+\epsilon}{2} = 1 - \beta_0, \\
\mu_1(f \in W^{s,p}(\Omega) : I_f^q \ge c + \Delta) = \mu_1(I_f^q \ge I_{g_1}^q) \ge \frac{1+\epsilon}{2} = 1 - \beta_1.
\end{aligned}
\tag{B.8}
$$

Hence, we have that (B.1) holds true. On the other hand, recall that the quadrature points $\{x_1, \cdots, x_n\}$ are identical and independent samples from the uniform distribution on $\Omega$, which enables us to write the marginal distributions in an explicit form as follows:

$$
\begin{aligned}
\mathbb{P}_0(\vec{x}, \vec{y}) &= \Big(\frac{1+\epsilon}{2} \prod_{i:x_i\in\Omega_1} \delta_0(y_i) + \frac{1-\epsilon}{2} \prod_{i:x_i\in\Omega_1} \delta_{g_1(x_i)}(y_i)\Big) \cdot \prod_{j=2}^{m^d} \Big(\prod_{i:x_i\in\Omega_j} \delta_0(y_i)\Big), \\
\mathbb{P}_1(\vec{x}, \vec{y}) &= \Big(\frac{1-\epsilon}{2} \prod_{i:x_i\in\Omega_1} \delta_0(y_i) + \frac{1+\epsilon}{2} \prod_{i:x_i\in\Omega_1} \delta_{g_1(x_i)}(y_i)\Big) \cdot \prod_{j=2}^{m^d} \Big(\prod_{i:x_i\in\Omega_j} \delta_0(y_i)\Big).
\end{aligned}
\tag{B.9}
$$

In particular, we have $\mathbb{P}_0 = \mathbb{P}_1$ when the set $\{i : x_i \in \Omega_1\}$ is empty. Combing this fact with (B.9) above allows us to compute the KL divergence between $\mathbb{P}_0$ and $\mathbb{P}_1$ as below

$$
\begin{aligned}
KL(\mathbb{P}_0||\mathbb{P}_1) &= \int_\Omega \cdots \int_\Omega \Big(\int_{-\infty}^\infty \cdots \int_{-\infty}^\infty \log\Big(\frac{\mathbb{P}_0(\vec{x},\vec{y})}{\mathbb{P}_1(\vec{x},\vec{y})}\Big) \mathbb{P}_0(\vec{x},\vec{y})dy_1 \cdots dy_n\Big) dx_1 \cdots dx_n \\
&= \int_\Omega \cdots \int_\Omega \Big(\int_{-\infty}^\infty \cdots \int_{-\infty}^\infty \log\Big(\frac{\frac{1+\epsilon}{2}\prod_{i:x_i\in\Omega_1}\delta_0(y_i) + \frac{1-\epsilon}{2}\prod_{i:x_i\in\Omega_1}\delta_{g_1(x_i)}(y_i)}{\frac{1-\epsilon}{2}\prod_{i:x_i\in\Omega_1}\delta_0(y_i) + \frac{1+\epsilon}{2}\prod_{i:x_i\in\Omega_1}\delta_{g_1(x_i)}(y_i)}\Big) \\
&\quad \cdot \Big(\frac{1+\epsilon}{2}\prod_{i:x_i\in\Omega_1}\delta_0(y_i) + \frac{1-\epsilon}{2}\prod_{i:x_i\in\Omega_1}\delta_{g_1(x_i)}(y_i)\Big) \cdot \Big(\prod_{j=2}^{m^d}\prod_{i:x_i\in\Omega_j}\delta_0(y_i)\prod_{i=1}^n dy_i\Big)\prod_{i=1}^n dx_i \\
&= \int_\Omega \cdots \int_\Omega \Big(\int_{-\infty}^\infty \cdots \int_{-\infty}^\infty \log\Big(\frac{\frac{1+\epsilon}{2}\prod_{i:x_i\in\Omega_1}\delta_0(y_i) + \frac{1-\epsilon}{2}\prod_{i:x_i\in\Omega_1}\delta_{g_1(x_i)}(y_i)}{\frac{1-\epsilon}{2}\prod_{i:x_i\in\Omega_1}\delta_0(y_i) + \frac{1+\epsilon}{2}\prod_{i:x_i\in\Omega_1}\delta_{g_1(x_i)}(y_i)}\Big) \\
&\quad \cdot \Big(\frac{1+\epsilon}{2}\prod_{i:x_i\in\Omega_1}\delta_0(y_i) + \frac{1-\epsilon}{2}\prod_{i:x_i\in\Omega_1}\delta_{g_1(x_i)}(y_i)\Big)\prod_{i:x_i\in\Omega_1}dy_i\Big)\prod_{i=1}^n dx_i \\
&= \Big(\log\Big(\frac{1+\epsilon}{1-\epsilon}\Big)\frac{1+\epsilon}{2} + \log\Big(\frac{1-\epsilon}{1+\epsilon}\Big)\frac{1-\epsilon}{2}\Big)\mathbb{P}\Big(\{i : x_i \in \Omega_1\} \ne \varnothing\Big) \\
&= \epsilon \log\Big(\frac{1+\epsilon}{1-\epsilon}\Big)\mathbb{P}\Big(\{i : x_i \in \Omega_1\} \ne \varnothing\Big).
\end{aligned}
\tag{B.10}
$$

Moreover, since the probability that $\{i : x_i \in \Omega_1\} = \varnothing$ equals to $(\frac{m^d-1}{m^d})^n = (\frac{m^d-1}{m^d})^{\frac{m^d}{200}}$, we have

$$
\mathbb{P}\Big(\{i : x_i \in \Omega_1\} \ne \varnothing\Big) = 1 - \Big(1 - \frac{1}{m^d}\Big)^{\frac{m^d}{200}} \le 1 - \Big(\frac{1}{e}\Big(1 - \frac{1}{m^d}\Big)\Big)^{\frac{1}{200}} \le 1 - (2e)^{-\frac{1}{200}}.
\tag{B.11}
$$

Now we may combine (B.10), (B.11) and Pinkser's inequality to upper bound the TV distance between $\mathbb{P}_0$ and $\mathbb{P}_1$ as below:

$$TV(\mathbb{P}_0\|\mathbb{P}_1) \leq \sqrt{\frac{1}{2}KL(\mathbb{P}_0\|\mathbb{P}_1)} \leq \sqrt{\frac{1-(2e)^{-\frac{1}{200}}}{2}\epsilon\log\left(\frac{1+\epsilon}{1-\epsilon}\right)} \leq \sqrt{\frac{3}{100}}\epsilon = \frac{\sqrt{3}}{10}\epsilon. \quad \text{(B.12)}$$

Finally, by substituting (B.6), (B.12), $\Delta = \frac{1}{2}I_{g_1}^q$ and $\beta_0 = \beta_1 = \frac{1-\epsilon}{2} = \frac{1}{4}$ into (B.2) and applying Markov's inequality, we obtain the final lower bound

$$\inf_{\hat{H}^q \in \mathcal{H}_n^{f,q}} \sup_{f \in W^{s,p}(\Omega)} \mathbb{E}_{\{x_i\}_{i=1}^n, \{y_i\}_{i=1}^n}\left[\left|\hat{H}^q\left(\{x_i\}_{i=1}^n, \{y_i\}_{i=1}^n\right) - I_f^q\right|\right]$$

$$\geq \Delta \inf_{\hat{H}^q \in \mathcal{H}_n^{f,q}} \sup_{f \in W^{s,p}(\Omega)} \mathbb{P}_{\{x_i\}_{i=1}^n, \{y_i\}_{i=1}^n}\left[\left|\hat{H}^q\left(\{x_i\}_{i=1}^n, \{y_i\}_{i=1}^n\right) - I_f^q\right| \geq \Delta\right] \quad \text{(B.13)}$$

$$\geq \frac{1}{2}I_{g_1}^q \frac{1 - TV(\mathbb{P}_0\|\mathbb{P}_1) - \beta_0 - \beta_1}{2} \geq \frac{1}{4}\left(1 - \frac{\sqrt{3}}{10}\right)\epsilon I_{g_1}^q$$

$$= \frac{1}{8}\left(1 - \frac{\sqrt{3}}{10}\right)(200n)^{-\frac{q}{d}(s-\frac{d}{p})-1}\|K\|_{L^q([-\frac{1}{2},\frac{1}{2}]^d)}^q \gtrsim n^{-q(\frac{s}{d}-\frac{1}{p})-1},$$

which is exactly the first term in the RHS of (2.1).

(Case II) Now let us proceed to prove the second lower bound in (2.1). For any $1 \leq j \leq m^d$, consider first some function $f_j$ defined as follows

$$f_j(x) = \begin{cases} m^{-s}K(m(x-c_j)) \ (x \in \Omega_j), \\ 0 \ (\text{otherwise}), \end{cases} \quad \text{(B.14)}$$

which satisfies $\text{supp}(f_j) \subseteq \Omega_j$, $f_j \in C^\infty(\Omega)$ and $f_j(x) \geq 0$ ($\forall x \in \Omega$). We further pick two constants $\alpha, M$ satisfying $\alpha := \|K\|_{L^\infty([-\frac{1}{2},\frac{1}{2}]^d)}$ and $M = 3\alpha$. Now consider the following finite set of $2^{m^d}$ functions:

$$\mathcal{S} := \left\{M + \sum_{j=1}^{m^d}\eta_j f_j : \eta_j \in \{\pm 1\}, \ \forall \ 1 \leq j \leq m^d\right\}. \quad \text{(B.15)}$$

We will proceed to verify that any element in $\mathcal{S}$ must be in $W^{s,p}(\Omega)$ for any $m$. Note that for any $\eta_j \in \{\pm 1\}$ ($1 \leq j \leq m^d$) and any $t \in \mathbb{N}_0^d$ satisfying $|t| \leq s$, we have

$$\left\|D^t\left(M + \sum_{j=1}^{m^d}\eta_j f_j\right)\right\|_{L^p(\Omega)}^p \leq \left(M + \left\|\sum_{j=1}^{m^d}\eta_j(D^t f_j)\right\|_{L^p(\Omega)}\right)^p$$

$$\leq 2^p\left(M^p + \left\|\sum_{j=1}^{m^d}\eta_j(D^t f_j)\right\|_{L^p(\Omega)}^p\right) \lesssim M^p + \sum_{j=1}^{m^d}\|D^t f_j\|_{L^p(\Omega_j)}^p$$

$$= M^p + \sum_{j=1}^{m^d}\int_{\Omega_j}\left|m^{-s+|t|}(D^t K)(m(x-c_j))\right|^p dx$$

$$= M^p + m^{(|t|-s)p}\sum_{j=1}^{m^d}\int_{[-\frac{1}{2},\frac{1}{2}]^d}\left|(D^t K)(y)\right|^p \frac{1}{m^d}dy$$

$$\leq M^p + \|D^t K\|_{L^p([-\frac{1}{2},\frac{1}{2}]^d)}^p \lesssim 1.$$

This gives us that $\mathcal{S} \subset W^{s,p}(\Omega)$ for any $m$, as desired. Now let's pick $\kappa = \frac{1}{3}\sqrt{\frac{2}{3n}}$ and take $\{w_j^{(0)}\}_{j=1}^{m^d}$ and $\{w_j^{(1)}\}_{j=1}^{m^d}$ to be independent and identical copies of $w_{\frac{1+\kappa}{2}}$ and $w_{\frac{1-\kappa}{2}}$ respectively. Then we define $\mu_0, \mu_1$ to be two discrete measures supported on the finite set $\mathcal{S}$ such that the following condition holds for any $\eta_j \in \{\pm 1\}$ ($1 \leq j \leq m^d$):

$$\mu_k\left(\left\{M + \sum_{j=1}^{m^d}\eta_j f_j\right\}\right) = \prod_{j=1}^{m^d}\mathbb{P}(w_j^{(k)} = \eta_j), \ k \in \{0,1\}. \quad \text{(B.16)}$$

In order to determine the separation distance $\Delta$ between the two priors $\mu_0$ and $\mu_1$, we need to define two quantities $A := \int_{\Omega_j} (M + f_j(x))^q dx$ and $B := \int_{\Omega_j} (M - f_j(x))^q dx$, which both remain the same for any $1 \leq j \leq m^d$. Now consider deriving a lower bound on the quantity $\Delta' := A - B > 0$. Note that for any fixed $j \in \{1, 2, \cdots, m^d\}$, we have $M > 2\alpha \geq 2m^{-s}\|K\|_{L^\infty([-\frac{1}{2},\frac{1}{2}]^d)} = 2\|f_j\|_{L^\infty(\Omega_j)}$, which implies $M + y > \frac{1}{2}M > 0$ for any $y \in [-\|f_j\|_{L^\infty(\Omega_j)}, \|f_j\|_{L^\infty(\Omega_j)}]$. This helps us obtain the following lower bound on $\Delta'$:

$$
\begin{aligned}
\Delta' &= \int_{\Omega_j} (M + f_j(x))^q dx - \int_{\Omega_j} (M - f_j(x))^q dx = \int_{\Omega_j} \Big( \int_{-f_j(x)}^{f_j(x)} q(M + y)^{q-1} dy \Big) dx \\
&\geq \int_{\Omega_j} \Big( \int_{-f_j(x)}^{f_j(x)} q(\frac{1}{2}M)^{q-1} dy \Big) dx = \frac{q}{2^{q-1}} M^{q-1} \int_{\Omega_j} \Big( 2f_j(x) \Big) dx \gtrsim \int_{\Omega_j} f_j(x) dx \qquad \text{(B.17)} \\
&= \int_{\Omega_j} m^{-s} K(m(x - c_j)) dx = m^{-s} \int_{[-\frac{1}{2},\frac{1}{2}]^d} K(y) \frac{1}{m^d} dy = m^{-s-d}\|K\|_{L^1([-\frac{1}{2},\frac{1}{2}]^d)}.
\end{aligned}
$$

Moreover, let us pick $\lambda = \frac{1}{2}$ and apply Hoeffding's Inequality (Lemma 3) to the bounded random variables $\{w_j^{(0)}\}_{j=1}^{m^d}$ and $\{w_j^{(1)}\}_{j=1}^{m^d}$ to deduce that

$$
\begin{aligned}
\mathbb{P}\Big( \sum_{j=1}^{m^d} w_j^{(0)} \geq -(1-\lambda)m^d\kappa \Big) &\leq \exp\Big( -\frac{2(\lambda m^d\kappa)^2}{4m^d} \Big) = \exp\Big( -\frac{1}{2}\lambda^2\kappa^2 m^d \Big), \\
\mathbb{P}\Big( \sum_{j=1}^{m^d} w_j^{(1)} \leq (1-\lambda)m^d\kappa \Big) &\leq \exp\Big( -\frac{2(\lambda m^d\kappa)^2}{4m^d} \Big) = \exp\Big( -\frac{1}{2}\lambda^2\kappa^2 m^d \Big).
\end{aligned}
\qquad \text{(B.18)}
$$

By taking $c := \frac{m^d}{2}(A + B)$, $\Delta := (1 - \lambda)\kappa m^d(A - B) = (1 - \lambda)\kappa m^d\Delta'$ and $\beta_0 = \beta_1 = \exp\Big( -\frac{1}{2}\lambda^2\kappa^2 m^d \Big)$, we may combine (B.17) and (B.18) justified above to get that

$$
\begin{aligned}
&\mu_0(f \in W^{s,p}(\Omega) : I_f^q \leq c - \Delta) = \mathbb{P}\Big( \sum_{j=1}^{m^d} I_{M+w_j^{(0)}f_j}^q \leq \frac{1 - (1-\lambda)\kappa}{2}m^d A + \frac{1 + (1-\lambda)\kappa}{2}m^d B \Big) \\
&\geq \mathbb{P}\Big( \sum_{j=1}^{m^d} w_j^{(0)} \leq -(1-\lambda)m^d\kappa \Big) = 1 - \mathbb{P}\Big( \sum_{j=1}^{m^d} w_j^{(0)} \geq -(1-\lambda)m^d\kappa \Big) \\
&\geq 1 - \exp\Big( -\frac{1}{2}\lambda^2\kappa^2 m^d \Big) = 1 - \beta_0, \\
&\mu_1(f \in W^{s,p}(\Omega) : I_f^q \geq c + \Delta) = \mathbb{P}\Big( \sum_{j=1}^{m^d} I_{M+w_j^{(1)}f_j}^q \geq \frac{1 + (1-\lambda)\kappa}{2}m^d A + \frac{1 - (1-\lambda)\kappa}{2}m^d B \Big) \\
&\geq \mathbb{P}\Big( \sum_{j=1}^{m^d} w_j^{(1)} \geq (1-\lambda)m^d\kappa \Big) = 1 - \mathbb{P}\Big( \sum_{j=1}^{m^d} w_j^{(0)} \leq (1-\lambda)m^d\kappa \Big) \\
&\geq 1 - \exp\Big( -\frac{1}{2}\lambda^2\kappa^2 m^d \Big) = 1 - \beta_1,
\end{aligned}
\qquad \text{(B.19)}
$$

which indicates that (B.1) holds true. Now let's consider bounding the KL divergence between the two marginal distributions $\mathbb{P}_0, \mathbb{P}_1$ associated with $\mu_0, \mu_1$, respectively. Using the fact that $\{x_1, \cdots, x_n\}$ are identical and independent samples from the uniform distribution on $\Omega$ again allows us to write the marginal distributions in an explicit form as follows:

$$
\begin{aligned}
\mathbb{P}_0(\vec{x}, \vec{y}) &= \prod_{j=1}^{m^d} \Big( \frac{1 + \kappa}{2} \prod_{i:x_i \in \Omega_j} \delta_{M-f_j(x_i)}(y_i) + \frac{1 - \kappa}{2} \prod_{i:x_i \in \Omega_j} \delta_{M+f_j(x_i)}(y_i) \Big), \\
\mathbb{P}_1(\vec{x}, \vec{y}) &= \prod_{j=1}^{m^d} \Big( \frac{1 - \kappa}{2} \prod_{i:x_i \in \Omega_j} \delta_{M-f_j(x_i)}(y_i) + \frac{1 + \kappa}{2} \prod_{i:x_i \in \Omega_j} \delta_{M+f_j(x_i)}(y_i) \Big).
\end{aligned}
\qquad \text{(B.20)}
$$

Furthermore, for any $n$ quadrature points $\{x_i\}_{i=1}^n$, we use $\mathcal{J}_n$ to denote the set of all indices $j$ satisfying that $\Omega_j$ contains at least one of the points in $\{x_i\}_{i=1}^n$, *i.e,*

$$\mathcal{J}_n := \mathcal{J}_n(x_1, \cdots, x_n) = \left\{ j : 1 \le j \le m^d \text{ and } \Omega_j \cap \{x_1, \cdots, x_n\} \ne \varnothing \right\} \tag{B.21}$$

Given that $m^d = 200n > n$, we have $|\mathcal{J}_n| \le n$ for any $n$ quadrature points $\{x_i\}_{i=1}^n$. Using this upper bound on $|\mathcal{J}_n|$ allows us to bound the KL divergence between $\mathbb{P}_0$ and $\mathbb{P}_1$ in the following way:

$$
\begin{aligned}
KL(\mathbb{P}_0\|\mathbb{P}_1) &= \int_\Omega \cdots \int_\Omega \left( \int_{-\infty}^\infty \cdots \int_{-\infty}^\infty \log\left( \frac{\mathbb{P}_0(\vec{x}, \vec{y})}{\mathbb{P}_1(\vec{x}, \vec{y})} \right) \mathbb{P}_0(\vec{x}, \vec{y}) dy_1 \cdots dy_n \right) dx_1 \cdots dx_n \\
&= \int_\Omega \cdots \int_\Omega \left( \int_{-\infty}^\infty \cdots \int_{-\infty}^\infty \log\left( \prod_{j=1}^{m^d} \frac{\frac{1+\kappa}{2} \prod_{i:x_i \in \Omega_j} \delta_{M-f_j(x_i)}(y_i) + \frac{1-\kappa}{2} \prod_{i:x_i \in \Omega_j} \delta_{M+f_j(x_i)}(y_i)}{\frac{1-\kappa}{2} \prod_{i:x_i \in \Omega_j} \delta_{M-f_j(x_i)}(y_i) + \frac{1+\kappa}{2} \prod_{i:x_i \in \Omega_j} \delta_{M+f_j(x_i)}(y_i)} \right) \right. \\
&\quad \left. \cdot \prod_{j=1}^{m^d} \left( \frac{1+\kappa}{2} \prod_{i:x_i \in \Omega_j} \delta_{M-f_j(x_i)}(y_i) + \frac{1-\kappa}{2} \prod_{i:x_i \in \Omega_j} \delta_{M+f_j(x_i)}(y_i) \right) \prod_{i=1}^n dy_i \right) \prod_{i=1}^n dx_i \\
&= \int_\Omega \cdots \int_\Omega \left( \int_{-\infty}^\infty \cdots \int_{-\infty}^\infty \log\left( \prod_{j \in \mathcal{J}_n} \frac{\frac{1+\kappa}{2} \prod_{i:x_i \in \Omega_j} \delta_{M-f_j(x_i)}(y_i) + \frac{1-\kappa}{2} \prod_{i:x_i \in \Omega_j} \delta_{M+f_j(x_i)}(y_i)}{\frac{1-\kappa}{2} \prod_{i:x_i \in \Omega_j} \delta_{M-f_j(x_i)}(y_i) + \frac{1+\kappa}{2} \prod_{i:x_i \in \Omega_j} \delta_{M+f_j(x_i)}(y_i)} \right) \right. \\
&\quad \left. \cdot \prod_{j \in \mathcal{J}_n} \left( \frac{1+\kappa}{2} \prod_{i:x_i \in \Omega_j} \delta_{M-f_j(x_i)}(y_i) + \frac{1-\kappa}{2} \prod_{i:x_i \in \Omega_j} \delta_{M+f_j(x_i)}(y_i) \right) \prod_{i=1}^n dy_i \right) \prod_{i=1}^n dx_i \\
&= \int_\Omega \cdots \int_\Omega \left( \sum_{j \in \mathcal{J}_n} \int_{-\infty}^\infty \cdots \int_{-\infty}^\infty \log\left( \frac{\frac{1+\kappa}{2} \prod_{i:x_i \in \Omega_j} \delta_{M-f_j(x_i)}(y_i) + \frac{1-\kappa}{2} \prod_{i:x_i \in \Omega_j} \delta_{M+f_j(x_i)}(y_i)}{\frac{1-\kappa}{2} \prod_{i:x_i \in \Omega_j} \delta_{M-f_j(x_i)}(y_i) + \frac{1+\kappa}{2} \prod_{i:x_i \in \Omega_j} \delta_{M+f_j(x_i)}(y_i)} \right) \right. \\
&\quad \left. \cdot \left( \frac{1+\kappa}{2} \prod_{i:x_i \in \Omega_j} \delta_{M-f_j(x_i)}(y_i) + \frac{1-\kappa}{2} \prod_{i:x_i \in \Omega_j} \delta_{M+f_j(x_i)}(y_i) \right) \prod_{i:x_i \in \Omega_j} dy_i \right) \prod_{i=1}^n dx_i \\
&= \int_\Omega \cdots \int_\Omega |\mathcal{J}_n| \left( \log\left(\frac{1+\kappa}{1-\kappa}\right)\frac{1+\kappa}{2} + \log\left(\frac{1-\kappa}{1+\kappa}\right)\frac{1-\kappa}{2} \right) \prod_{i=1}^n dx_i \le n\kappa \log\left(\frac{1+\kappa}{1-\kappa}\right).
\end{aligned}
\tag{B.22}
$$

Now we may combine (B.22) and Pinkser's inequality to upper bound the TV distance between $\mathbb{P}_0$ and $\mathbb{P}_1$ as below:

$$TV(\mathbb{P}_0\|\mathbb{P}_1) \le \sqrt{\frac{1}{2}KL(\mathbb{P}_0\|\mathbb{P}_1)} \le \sqrt{\frac{n\kappa}{2}\log\left(\frac{1+\kappa}{1-\kappa}\right)} \le \sqrt{\frac{3n}{2}}\kappa = \frac{1}{3}. \tag{B.23}$$

Finally, by substituting (B.17), (B.23), $\Delta = (1-\lambda)\kappa m^d \Delta'$ and $\beta_0 = \beta_1 = \exp\left(-\frac{1}{2}\lambda^2\kappa^2 m^d\right) = \exp(-\frac{50}{27}) < \frac{1}{6}$ into (B.2) and applying Markov's inequality, we obtain the final lower bound

$$
\begin{aligned}
&\inf_{\hat{H}^q \in \mathcal{H}_n^{f,q}} \sup_{f \in W^{s,p}(\Omega)} \mathbb{E}_{\{x_i\}_{i=1}^n, \{y_i\}_{i=1}^n} \left[ \left| \hat{H}^q\left(\{x_i\}_{i=1}^n, \{y_i\}_{i=1}^n\right) - I_f^q \right| \right] \\
&\ge \Delta \inf_{\hat{H}^q \in \mathcal{H}_n^{f,q}} \sup_{f \in W^{s,p}(\Omega)} \mathbb{P}_{\{x_i\}_{i=1}^n, \{y_i\}_{i=1}^n} \left[ \left| \hat{H}^q\left(\{x_i\}_{i=1}^n, \{y_i\}_{i=1}^n\right) - I_f^q \right| \ge \Delta \right] \\
&\ge (1-\lambda)\kappa m^d \Delta' \frac{1 - TV(\mathbb{P}_0\|\mathbb{P}_1) - \beta_0 - \beta_1}{2} \ge \frac{1}{2}\frac{\sqrt{2}}{3\sqrt{3n}} \cdot (200n) \cdot \frac{\Delta'}{6} \\
&\gtrsim \sqrt{n}\Delta' \gtrsim \sqrt{n}(200n)^{-\frac{s+d}{d}}\|K\|_{L^1([-\frac{1}{2}, \frac{1}{2}]^d)} \gtrsim n^{-\frac{s}{d}-\frac{1}{2}},
\end{aligned}
\tag{B.24}
$$

which is exactly the second term in the RHS of (2.1). Combining the two lower bounds proved in (B.13) and (B.24) concludes our proof of Theorem 2.1.

# C Proof of Upper Bounds in Section 3

## C.1 Proof of Theorem 3.1 (Regression-Adjusted Control Variate)

In this subsection, we present a detailed proof of Theorem 3.1. With the first half of the quadrature points $\{x_i\}_{i=1}^{\frac{n}{2}}$ and observed function values $\{y_i\}_{i=1}^{\frac{n}{2}}$ as inputs, we pick the regression adjusted control variate $\hat{f}_{1:\frac{n}{2}}$ to be the estimator returned by the oracle $K_{\frac{n}{2}}$ specified in Assumption 3.1. Moreover, we use the following expression to denote the variance of the function $\hat{f}_{1:\frac{n}{2}}^q(x) - f^q(x)$ with respect to the uniform distribution on $\Omega$:

$$\text{Var}(\hat{f}_{1:\frac{n}{2}}^q - f^q) := \int_\Omega (f^q(x) - \hat{f}_{1:\frac{n}{2}}^q(x))^2 dx - \left( \int_\Omega (f^q(x) - \hat{f}_{1:\frac{n}{2}}^q(x)) dx \right)^2. \tag{C.1}$$

By plugging in the expression of $\hat{H}_C^q, I_f^q$ and using the fact that $\{x_i\}_{i=1}^n$ are identical and independent copies of the uniform random variable over $\Omega$, we have

$$\mathbb{E}_{\{x_i\}_{i=1}^n, \{y_i\}_{i=1}^n} \left[ \left| \hat{H}_C^q \left( \{x_i\}_{i=1}^n, \{y_i\}_{i=1}^n \right) - I_f^q \right|^2 \right]$$

$$= \mathbb{E}_{\{x_i\}_{i=1}^n} \left[ \left| \int_\Omega \hat{f}_{1:\frac{n}{2}}^q(x) dx + \frac{2}{n} \sum_{i=\frac{n}{2}+1}^n \left( f^q(x_i) - \hat{f}_{1:\frac{n}{2}}^q(x_i) \right) - \int_\Omega f^q(x) dx \right|^2 \right]$$

$$= \mathbb{E}_{\{x_i\}_{i=1}^{\frac{n}{2}}} \left[ \mathbb{E}_{\{x_i\}_{i=\frac{n}{2}+1}^n} \left[ \left| \frac{1}{\frac{n}{2}} \sum_{i=\frac{n}{2}+1}^n \left( f^q(x_i) - \hat{f}_{1:\frac{n}{2}}^q(x_i) - \int_\Omega (f^q(x) - \hat{f}_{1:\frac{n}{2}}^q(x)) dx \right) \right|^2 \right] \right]$$

$$= \mathbb{E}_{\{x_i\}_{i=1}^{\frac{n}{2}}} \left[ \frac{4}{n^2} \sum_{i=\frac{n}{2}+1}^n \mathbb{E}_{x_i} \left[ \left| \left( f^q(x_i) - \hat{f}_{1:\frac{n}{2}}^q(x_i) - \int_\Omega (f^q(x) - \hat{f}_{1:\frac{n}{2}}^q(x)) dx \right) \right|^2 \right] \right]$$

$$= \mathbb{E}_{\{x_i\}_{i=1}^{\frac{n}{2}}} \left[ \frac{4}{n^2} \sum_{i=\frac{n}{2}+1}^n \text{Var}(\hat{f}_{1:\frac{n}{2}}^q - f^q) \right] = \frac{2}{n} \mathbb{E}_{\{x_i\}_{i=1}^{\frac{n}{2}}} \left[ \text{Var}(\hat{f}_{1:\frac{n}{2}}^q - f^q) \right].$$

$$\tag{C.2}$$

From the identity above, we know that it suffices to upper bound the term $\mathbb{E}_{\{x_i\}_{i=1}^{\frac{n}{2}}} \left[ \text{Var}(\hat{f}_{1:\frac{n}{2}}^q - f^q) \right]$. Let $g_{1:\frac{n}{2}} := \hat{f}_{1:\frac{n}{2}} - f$ denote the difference between the estimator $\hat{f}_{1:\frac{n}{2}}$ and underlying function $f$. Then we may further upper bound the expression $\mathbb{E}_{\{x_i\}_{i=1}^{\frac{n}{2}}} \left[ \text{Var}(\hat{f}_{1:\frac{n}{2}}^q - f^q) \right]$ as follows:

$$\mathbb{E}_{\{x_i\}_{i=1}^{\frac{n}{2}}} \left[ \text{Var}(\hat{f}_{1:\frac{n}{2}}^q - f^q) \right] = \mathbb{E}_{\{x_i\}_{i=1}^{\frac{n}{2}}} \left[ \int_\Omega (f^q(x) - \hat{f}_{1:\frac{n}{2}}^q(x))^2 dx - \left( \int_\Omega (f^q(x) - \hat{f}_{1:\frac{n}{2}}^q(x)) dx \right)^2 \right]$$

$$\leq \mathbb{E}_{\{x_i\}_{i=1}^{\frac{n}{2}}} \left[ \int_\Omega \left( f^q(x) - \hat{f}_{1:\frac{n}{2}}^q(x) \right)^2 dx \right] = \mathbb{E}_{\{x_i\}_{i=1}^{\frac{n}{2}}} \left[ \int_\Omega \left( (f(x) + g_{1:\frac{n}{2}}(x))^q - f^q(x) \right)^2 dx \right]$$

$$= \mathbb{E}_{\{x_i\}_{i=1}^{\frac{n}{2}}} \left[ \int_\Omega \left( \int_0^{g_{1:\frac{n}{2}}(x)} q(f(x) + y)^{q-1} dy \right)^2 dx \right]$$

$$\leq \mathbb{E}_{\{x_i\}_{i=1}^{\frac{n}{2}}} \left[ \int_\Omega \left| \int_0^{g_{1:\frac{n}{2}}(x)} 1 dy \right| \left| \int_0^{g_{1:\frac{n}{2}}(x)} q^2 (|f(x) + y|^2)^{q-1} dy \right| dx \right]$$

$$\lesssim \mathbb{E}_{\{x_i\}_{i=1}^{\frac{n}{2}}} \left[ \int_\Omega |g_{1:\frac{n}{2}}(x)| \cdot |g_{1:\frac{n}{2}}(x)| \max \left\{ |f^{2q-2}(x)|, |g_{1:\frac{n}{2}}^{2q-2}(x)| \right\} dx \right]$$

$$\lesssim \mathbb{E}_{\{x_i\}_{i=1}^{\frac{n}{2}}} \left[ \int_\Omega |g_{1:\frac{n}{2}}^{2q}(x)| dx \right] + \mathbb{E}_{\{x_i\}_{i=1}^{\frac{n}{2}}} \left[ \int_\Omega |g_{1:\frac{n}{2}}^2(x) f^{2q-2}(x)| dx \right].$$

$$\tag{C.3}$$

Now let's proceed to bound from above the two expected integrals in the last line of (C.3). For the first expected integral, since $s > \frac{2dq - dp}{2pq} \Rightarrow \frac{1}{2q} > \frac{d - sp}{pd}$, we may apply (3.2) in Assumption 3.1 to

deduce that

$$
\mathbb{E}_{\{x_i\}_{i=1}^{\frac{n}{2}}} \left[ \int_\Omega |g_{1:\frac{n}{2}}^{2q}(x)|dx \right] = \mathbb{E}_{\{x_i\}_{i=1}^{\frac{n}{2}}} \left[ \|\hat{f}_{1:\frac{n}{2}} - f\|_{L^{2q}(\Omega)}^{2q} \right]
$$
$$
\lesssim ((\tfrac{n}{2})^{-\frac{s}{d}+(\frac{1}{p}-\frac{1}{2q})_+})^{2q} \lesssim n^{2q(-\frac{s}{d}+\frac{1}{p}-\frac{1}{2q})} = n^{2q(\frac{1}{p}-\frac{s}{d})-1},
$$

(C.4)

where the last equality above follows from the given assumption that $p < 2q$. Now let's proceed to bound from above the second expected integral in (C.3). Here we define $p^* = (\max\{\frac{1}{p} - \frac{s}{d}, 0\})^{-1}$, *i.e*, $p^* = \frac{pd}{d-sp}$ when $s < \frac{d}{p}$ and $p^* = \infty$ otherwise. From Sobolev Embedding Theorem (Lemma 1), we have that $W^{s,p}(\Omega) \subseteq L^{p^*}(\Omega)$. Based on the value of the smoothness parameter $s$, we have three separate cases as below:

(Case I) When $s \in (\frac{d}{p}, \infty)$, we have $p^* = \infty$ and $f \in W^{s,p}(\Omega) \subset L^\infty(\Omega)$. Since $\hat{f}_{1:\frac{n}{2}}$ and $f$ are both in the Sobolev space $W^{s,p}(\Omega) \subseteq L^\infty(\Omega)$, we may further deduce that $g_{1:\frac{n}{2}} = \hat{f}_{1:\frac{n}{2}} - f \in W^{s,p}(\Omega) \subseteq L^\infty(\Omega) \subseteq L^2(\Omega)$. By picking $r = 2$ in (3.2) of Assumption 3.1, we may use the facts that $p > 2$ and $f \in L^\infty(\Omega)$ to deduce that

$$
\mathbb{E}_{\{x_i\}_{i=1}^{\frac{n}{2}}} \left[ \int_\Omega |g_{1:\frac{n}{2}}^2(x)f^{2q-2}(x)|dx \right] \lesssim \mathbb{E}_{\{x_i\}_{i=1}^{\frac{n}{2}}} \left[ \int_\Omega |g_{1:\frac{n}{2}}^2(x)|dx \right]
$$
$$
= \mathbb{E}_{\{x_i\}_{i=1}^{\frac{n}{2}}} \left[ \|\hat{f}_{1:\frac{n}{2}} - f\|_{L^2(\Omega)}^2 \right] \lesssim \left( n^{-\frac{s}{d}+(\frac{1}{p}-\frac{1}{2})_+} \right)^2 = n^{-\frac{2s}{d}},
$$

(C.5)

which is our final upper bound on the second expected integral in (C.3) under the assumption that $s \in (\frac{d}{p}, \infty)$.

(Case II) When $s \in (\frac{d(2q-p)}{p(2q-2)}, \frac{d}{p})$, we have $p^* = \frac{pd}{d-sp} > \frac{pd}{d-p\frac{d(2q-p)}{p(2q-2)}} = \frac{p(2q-2)}{p-2}$, which implies $f \in W^{s,p}(\Omega) \subseteq L^{p^*}(\Omega) \subseteq L^{\frac{p(2q-2)}{p-2}}(\Omega) \subseteq L^p(\Omega)$. Given that $\frac{p}{p-2} > 1$, we can further deduce that $f^{2q-2} \in L^{\frac{p}{p-2}}(\Omega)$ . Moreover, since $\hat{f}_{1:\frac{n}{2}} \in W^{s,p}(\Omega) \subseteq L^p(\Omega)$, we have that $g_{1:\frac{n}{2}} = \hat{f}_{1:\frac{n}{2}} - f \in L^p(\Omega)$. Given that $p > 2$, we can further deduce that $g_{1:\frac{n}{2}}^2 \in L^{\frac{p}{2}}(\Omega)$. Then we may apply Hölder's inequality (Lemma 2) to $g_{1:\frac{n}{2}}^2 \in L^{\frac{p}{2}}(\Omega)$ and $f^{2q-2} \in L^{\frac{p}{p-2}}(\Omega)$ to obtain that

$$
\mathbb{E}_{\{x_i\}_{i=1}^{\frac{n}{2}}} \left[ \int_\Omega |g_{1:\frac{n}{2}}^2(x)f^{2q-2}(x)|dx \right] = \mathbb{E}_{\{x_i\}_{i=1}^{\frac{n}{2}}} \left[ \|g_{1:\frac{n}{2}}^2 f^{2q-2}\|_{L^1(\Omega)} \right]
$$
$$
\leq \mathbb{E}_{\{x_i\}_{i=1}^{\frac{n}{2}}} \left[ \left\| g_{1:\frac{n}{2}}^2 \right\|_{L^{\frac{p}{2}}(\Omega)} \left\| f^{2q-2} \right\|_{L^{\frac{p}{p-2}}(\Omega)} \right] \leq \left\| f \right\|_{L^{\frac{p(2q-2)}{p-2}}(\Omega)}^{2q-2} \mathbb{E}_{\{x_i\}_{i=1}^{\frac{n}{2}}} \left[ \left\| g_{1:\frac{n}{2}} \right\|_{L^p(\Omega)}^2 \right].
$$

(C.6)

Note that the function $h(t) = t^{\frac{2}{p}}$ is concave and $\frac{1}{p} \in (\frac{d-sp}{pd}, 1]$ when $p > 2$. Hence, applying Jensen's inequality and picking $r = p$ in (3.2) of Assumption 3.1 further allows us to upper bound the last term in (C.6) as follows:

$$
\mathbb{E}_{\{x_i\}_{i=1}^{\frac{n}{2}}} \left[ \|g_{1:\frac{n}{2}}\|_{L^p(\Omega)}^2 \right] = \mathbb{E}_{\{x_i\}_{i=1}^{\frac{n}{2}}} \left[ \left( \|g_{1:\frac{n}{2}}\|_{L^p(\Omega)}^p \right)^{\frac{2}{p}} \right]
$$
$$
\leq \mathbb{E}_{\{x_i\}_{i=1}^{\frac{n}{2}}} \left[ \|g_{1:\frac{n}{2}}\|_{L^p(\Omega)}^p \right]^{\frac{2}{p}} = \mathbb{E}_{\{x_i\}_{i=1}^{\frac{n}{2}}} \left[ \|\hat{f}_{1:\frac{n}{2}} - f\|_{L^p(\Omega)}^p \right]^{\frac{2}{p}}
$$
$$
\lesssim \left( (\tfrac{n}{2})^{-\frac{s}{d}+(\frac{1}{p}-\frac{1}{p})_+} \right)^2 \lesssim n^{-\frac{2s}{d}}.
$$

(C.7)

Substituting (C.7) into (C.6) then gives us the final upper bound on the second expected integral in (C.3) under the assumption that $s \in (\frac{d(2q-p)}{p(2q-2)}, \infty)$:

$$
\mathbb{E}_{\{x_i\}_{i=1}^{\frac{n}{2}}} \left[ \int_\Omega |g_{1:\frac{n}{2}}^2(x)f^{2q-2}(x)|dx \right] \lesssim n^{-\frac{2s}{d}}.
$$

(C.8)

(Case III) When $s \in (\frac{d(2q-p)}{2pq}, \frac{d(2q-p)}{p(2q-2)})$, we have that $s < \frac{d}{p}$, which indicates that $p^* = \frac{pd}{d-sp}$ satisfies $2q < p^* < \frac{p(2q-2)}{p-2}$. Given that $p^* > 2q > 2q - 2$ and $f \in W^{s,p}(\Omega) \subseteq L^{p^*}(\Omega)$, we can deduce

that $f^{2q-2} \in L^{\frac{p^*}{2q-2}}(\Omega)$. Furthermore, note that $p^* > 2q$ implies $\frac{2p^*}{p^*+2-2q} < p^*$ and $p^* < \frac{p(2q-2)}{p-2}$ implies $\frac{2p^*}{p^*+2-2q} > p$. Since $\hat{f}_{1:\frac{n}{2}}$ and $f$ are both in the Sobolev space $W^{s,p}(\Omega) \subseteq L^{p^*}(\Omega)$, we may further deduce that $g_{1:\frac{n}{2}} = \hat{f}_{1:\frac{n}{2}} - f \in W^{s,p}(\Omega) \subseteq L^{p^*}(\Omega) \subseteq L^{\frac{2p^*}{p^*+2-2q}}(\Omega)$. Given that $q \geq 1 \Rightarrow \frac{p^*}{p^*+2-2q} \geq 1$, we have $g_{1:\frac{n}{2}}^2 \in L^{\frac{p^*}{p^*+2-2q}}(\Omega)$. Then we may apply Hölder's inequality (Lemma 2) to $g_{1:\frac{n}{2}}^2 \in L^{\frac{p^*}{p^*+2-2q}}(\Omega)$ and $f^{2q-2} \in L^{\frac{p^*}{2q-2}}(\Omega)$, which yields the following upper bound:

$$
\begin{aligned}
\mathbb{E}_{\{x_i\}_{i=1}^{\frac{n}{2}}}\left[\int_\Omega |g_{1:\frac{n}{2}}^2(x)f^{2q-2}(x)|dx\right] &= \mathbb{E}_{\{x_i\}_{i=1}^{\frac{n}{2}}}\left[\|g_{1:\frac{n}{2}}^2 f^{2q-2}\|_{L^1(\Omega)}\right] \\
&\leq \mathbb{E}_{\{x_i\}_{i=1}^{\frac{n}{2}}}\left[\left\|g_{1:\frac{n}{2}}^2\right\|_{L^{\frac{p^*}{p^*+2-2q}}(\Omega)}\left\|f^{2q-2}\right\|_{L^{\frac{p^*}{2q-2}}(\Omega)}\right] \quad \text{(C.9)}\\
&\leq \left\|f\right\|_{L^{p^*}(\Omega)}^{2q-2}\mathbb{E}_{\{x_i\}_{i=1}^{\frac{n}{2}}}\left[\left\|g_{1:\frac{n}{2}}\right\|_{L^{\frac{2p^*}{p^*+2-2q}}(\Omega)}^2\right].
\end{aligned}
$$

Note that the function $\omega(t) = t^{\frac{p^*+2-2q}{p^*}}$ is concave since $q \geq 1$. Moreover, using the given assumption $s \in (\frac{d(2q-p)}{2pq}, \frac{d(2q-p)}{p(2q-2)})$ we get that $\frac{pd}{d-sp} > 2q$, which further yields

$$
\frac{p^*+2-2q}{2p^*} = \frac{\frac{pd}{d-sp}+2-2q}{2\frac{pd}{d-sp}} > \frac{2}{2\frac{pd}{d-sp}} = \frac{d-sp}{pd},
$$

*i.e,* $\frac{(p^*+2-2q)}{2p^*} \in (\frac{d-sp}{pd}, 1]$. Hence, we may apply Jensen's inequality and (3.2) in Assumption 3.1 to upper-bound the last term in (C.9) as follows:

$$
\begin{aligned}
\mathbb{E}_{\{x_i\}_{i=1}^{\frac{n}{2}}}\left[\left\|g_{1:\frac{n}{2}}\right\|_{L^{\frac{2p^*}{p^*+2-2q}}(\Omega)}^2\right] &= \mathbb{E}_{\{x_i\}_{i=1}^{\frac{n}{2}}}\left[\left(\left\|g_{1:\frac{n}{2}}\right\|_{L^{\frac{2p^*}{p^*+2-2q}}(\Omega)}^{\frac{2p^*}{p^*+2-2q}}\right)^{\frac{p^*+2-2q}{p^*}}\right] \\
&\leq \mathbb{E}_{\{x_i\}_{i=1}^{\frac{n}{2}}}\left[\left\|g_{1:\frac{n}{2}}\right\|_{L^{\frac{2p^*}{p^*+2-2q}}(\Omega)}^{\frac{2p^*}{p^*+2-2q}}\right]^{\frac{(p^*+2-2q)}{p^*}} \\
&= \mathbb{E}_{\{x_i\}_{i=1}^{\frac{n}{2}}}\left[\left\|\hat{f}_{1:\frac{n}{2}}-f\right\|_{L^{\frac{2p^*}{p^*+2-2q}}(\Omega)}^{\frac{2p^*}{p^*+2-2q}}\right]^{\frac{(p^*+2-2q)}{p^*}} \quad \text{(C.10)}\\
&\lesssim \left((\frac{n}{2})^{-\frac{s}{d}+(\frac{1}{p}-\frac{p^*+2-2q}{2p^*})_+}\right)^2 \\
&\lesssim n^{-\frac{2s}{d}+2(\frac{1}{p}-\frac{p^*+2-2q}{2p^*})_+}.
\end{aligned}
$$

In order to simplify the last expression in (C.10), let's recall the fact that $p^* \in (2q, \frac{p(2q-2)}{p-2})$ proved above. This gives us that $p^*(p-2) < p(2q-2) \Rightarrow 2p^* > p(p^*+2-2q)$, *i.e,* $\frac{1}{p} > \frac{p^*+2-2q}{2p^*}$. Then we may simplify the power term in the last expression of (C.10) as follows:

$$
-\frac{2s}{d}+2\left(\frac{1}{p}-\frac{p^*+2-2q}{2p^*}\right)_+ = -\frac{2s}{d}+\frac{2}{p}-\left(1+\frac{2}{p^*}-\frac{2q}{p^*}\right) = \frac{2q}{p^*}-1 = 2q\left(\frac{1}{p}-\frac{s}{d}\right)-1.
$$

Now let's substitute (C.10) into (C.9), which gives us the final upper bound on the second expected integral in (C.3) under the assumption that $s \in (\frac{d(2q-p)}{2pq}, \frac{d(2q-p)}{p(2q-2)})$:

$$
\mathbb{E}_{\{x_i\}_{i=1}^{\frac{n}{2}}}\left[\int_\Omega |g_{1:\frac{n}{2}}^2(x)f^{2q-2}(x)|dx\right] \lesssim n^{2q(\frac{1}{p}-\frac{s}{d})-1}. \quad \text{(C.11)}
$$

Combining the upper bounds derived in (C.4), (C.5), (C.8) and (C.11) finally allows us to upper bound the expected variance $\mathbb{E}_{\{x_i\}_{i=1}^{\frac{n}{2}}}\left[\text{Var}(\hat{f}_{1:\frac{n}{2}}^q - f^q)\right]$ as below:

$$
\mathbb{E}_{\{x_i\}_{i=1}^{\frac{n}{2}}}\left[\text{Var}(\hat{f}_{1:\frac{n}{2}}^q - f^q)\right] \lesssim n^{2q(\frac{1}{p}-\frac{s}{d})-1} + \max\{n^{-\frac{2s}{d}}, n^{2q(\frac{1}{p}-\frac{s}{d})-1}\}. \quad \text{(C.12)}
$$

Finally, substituting (C.12) into C.2) derived at the beginning gives us the final upper bound:

$$\mathbb{E}_{\{x_i\}_{i=1}^n,\{y_i\}_{i=1}^n}\left[\left|\hat{H}_C^q\left(\{x_i\}_{i=1}^n,\{y_i\}_{i=1}^n\right)-I_f^q\right|\right]$$

$$\leq\sqrt{\mathbb{E}_{\{x_i\}_{i=1}^n,\{y_i\}_{i=1}^n}\left[\left|\hat{H}_C^q\left(\{x_i\}_{i=1}^n,\{y_i\}_{i=1}^n\right)-I_f^q\right|^2\right]}=\sqrt{\frac{2}{n}\mathbb{E}_{\{x_i\}_{i=1}^{\frac{n}{2}}}\left[\mathrm{Var}(\hat{f}_{1:\frac{n}{2}}^q-f^q)\right]}$$

$$\lesssim n^{-\frac{1}{2}}\sqrt{n^{2q(\frac{1}{p}-\frac{s}{d})-1}+\max\{n^{-\frac{2s}{d}},n^{2q(\frac{1}{p}-\frac{s}{d})-1}\}}\lesssim\max\{n^{-\frac{s}{d}-\frac{1}{2}},n^{-q(\frac{s}{d}-\frac{1}{p})-1}\}. \tag{C.13}$$

This concludes our proof of Theorem 3.1.

## C.2 Proof of Theorem 3.2 (Truncated Monte Carlo)

In this subsection, we provide a complete proof of Theorem 3.2. For any fixed parameter $M>0$, we may divide $\Omega$ into the following two regions:

$$\Omega_M^+:=\{x\in\Omega:|f(x)|\geq M\},\ \Omega_M^-:=\{x\in\Omega:|f(x)|<M\}, \tag{C.14}$$

where $\Omega_M^+\cap\Omega_M^-=\varnothing$ and $\Omega_M^+\cup\Omega_M^-=\Omega$. Let $f_M(x):=\max\left\{\min\{f(x),M\},-M\right\}$ $(\forall\,x\in\Omega)$ denote a truncated version of the given function $f$, where $M$ is the threshold. Also, we use the following expression to denote the expectation of the $q$-th power of the truncated function $f_M$ with respect to the uniform distribution on $\Omega$:

$$\mathbb{E}(f_M^q(x))=\int_\Omega\max\left\{\min\{f(x),M\},-M\right\}^q dx=\int_{\Omega_M^+}M^q dx+\int_{\Omega_M^-}f(x)^q dx, \tag{C.15}$$

where the last identity in (C.15) above follows from our definition of the two regions defined in (C.14). In a similar way, we can define the variance of the function $f_M^q$ as below:

$$\mathrm{Var}(f_M^q(x))=\mathbb{E}(f_M^{2q}(x))-\mathbb{E}(f_M^q(x))^2$$
$$=\int_\Omega\max\left\{\min\{f(x),M\},-M\right\}^{2q}dx-\left(\int_\Omega\max\left\{\min\{f(x),M\},-M\right\}^q dx\right)^2. \tag{C.16}$$

Furthermore, as $\{x_i\}_{i=1}^n$ are identical and independent samples of the uniform distribution on $\Omega$, we have that for any $1\leq i\leq n$, the following identity holds

$$\mathbb{E}_{\{x_i\}_{i=1}^n,\{y_i\}_{i=1}^n}\left[\hat{H}_M^q\left(\{x_i\}_{i=1}^n,\{y_i\}_{i=1}^n\right)\right]$$

$$=\mathbb{E}_{\{x_i\}_{i=1}^n,\{y_i\}_{i=1}^n}\left[\frac{1}{n}\sum_{i=1}^n\max\left\{\min\{y_i,M\},-M\right\}^q\right] \tag{C.17}$$

$$=\mathbb{E}_{x_i}\left[\max\left\{\min\{f(x_i),M\},-M\right\}^q\right]=\mathbb{E}_{x_i}[f_M^q(x_i)]=\mathbb{E}(f_M^q(x)).$$

Now we may use (C.17) and the bias-variance decomposition to derive an upper bound on the squared expected risk of the estimator $\hat{H}_M^q$ as follows:

$$\mathbb{E}_{\{x_i\}_{i=1}^n,\{y_i\}_{i=1}^n}\left[\left|\hat{H}_M^q\left(\{x_i\}_{i=1}^n,\{y_i\}_{i=1}^n\right)-I_f^q\right|^2\right]$$

$$=\mathbb{E}_{\{x_i\}_{i=1}^n,\{y_i\}_{i=1}^n}\left[\left|\hat{H}_M^q\left(\{x_i\}_{i=1}^n,\{y_i\}_{i=1}^n\right)-\mathbb{E}(f_M^q(x))+\mathbb{E}(f_M^q(x))-I_f^q\right|^2\right]$$

$$\leq 2\mathbb{E}_{\{x_i\}_{i=1}^n,\{y_i\}_{i=1}^n}\left[\left|\frac{1}{n}\sum_{i=1}^n\max\left\{\min\{y_i,M\},-M\right\}^q-\mathbb{E}(f_M^q(x))\right|^2\right] \tag{C.18}$$

$$+2\mathbb{E}_{\{x_i\}_{i=1}^n,\{y_i\}_{i=1}^n}\left[\left|\mathbb{E}(f_M^q(x))-I_f^q\right|^2\right]$$

where the first and the second term in the last line of (C.18) above denotes the variance and the bias part, respectively. Again, we define $p^* = (\max\{\frac{1}{p} - \frac{s}{d}, 0\})^{-1}$, *i.e*, $p^* = \frac{pd}{d-sp}$ when $s < \frac{d}{p}$ and $p^* = \infty$ otherwise. Under the assumption that $s < \frac{2dq-dp}{2pq} < \frac{d}{p}$, we have $p^* = \frac{pd}{d-sp} \in (p, 2q)$. Moreover, from Sobolev Embedding Theorem (Lemma 1), we have that $f \in W^{s,p}(\Omega) \subseteq L^{p^*}(\Omega)$.

On the one hand, since $p < 2q$, we can deduce that $|f(x)|^{2q} \le M^{2q-p^*}|f(x)|^{p^*}$ for any $x \in \Omega_M^-$ and $M^{2q} \le M^{2q-p^*}|f(x)|^{p^*}$ for any $x \in \Omega_M^+$, which helps us upper bound the variance part as below:

$$
\mathbb{E}_{\{x_i\}_{i=1}^n, \{y_i\}_{i=1}^n}\left[\left|\frac{1}{n}\sum_{i=1}^n \max\left\{\min\{y_i, M\}, -M\right\}^q - \mathbb{E}(f_M^q(x))\right|^2\right]
$$

$$
= \mathbb{E}_{\{x_i\}_{i=1}^n}\left[\left|\frac{1}{n}\sum_{i=1}^n \left(f_M^q(x_i) - \mathbb{E}_{x_i}\left[f_M^q(x_i)\right]\right)\right|^2\right]
$$

$$
= \frac{1}{n^2}\sum_{i=1}^n \mathbb{E}_{x_i}\left[\left(f_M^q(x_i) - \mathbb{E}_{x_i}\left[f_M^q(x_i)\right]\right)^2\right] = \frac{1}{n}\mathrm{Var}(f_M^q(x)) \tag{C.19}
$$

$$
\le \frac{1}{n}\mathbb{E}(f_M^{2q}(x)) = \frac{1}{n}\left(\int_{\Omega_M^+} M^{2q}dx + \int_{\Omega_M^-} f(x)^{2q}dx\right)
$$

$$
\le \frac{1}{n}\left(\int_{\Omega_M^+} M^{2q-p^*}|f(x)|^{p^*}dx + \int_{\Omega_M^-} M^{2q-p^*}|f(x)|^{p^*}dx\right)
$$

$$
= \frac{1}{n}\int_\Omega M^{2q-p^*}|f(x)|^{p^*}dx \lesssim \frac{M^{2q-p^*}}{n},
$$

where the last step of (C.19) above follows from the fact that $f \in W^{s,p}(\Omega) \subseteq L^{p^*}(\Omega)$.

On the other hand, using the fact that $p^* > p > q \Rightarrow |f(x)|^q \le M^{q-p^*}|f(x)|^{p^*}$ for any $x \in \Omega_M^+$, we may upper-bound the bias part as follows:

$$
\mathbb{E}_{\{x_i\}_{i=1}^n, \{y_i\}_{i=1}^n}\left[\left|\mathbb{E}(f_M^q(x)) - I_f^q\right|^2\right]
$$

$$
= \left|\int_{\Omega_M^+} M^q dx + \int_{\Omega_M^-} f(x)^q dx - \int_{\Omega_M^+} f^q(x)dx - \int_{\Omega_M^-} f(x)^q dx\right|^2
$$

$$
= \left|\int_{\Omega_M^+}\left(M^q - f^q(x)\right)dx\right|^2 \le \left|\int_{\Omega_M^+}\left|M^q - f^q(x)\right|dx\right|^2 \le \left|\int_{\Omega_M^+}\left(M^q + |f(x)|^q\right)dx\right|^2 \tag{C.20}
$$

$$
\le \left|2\int_{\Omega_M^+}|f(x)|^q dx\right|^2 \lesssim \left|\int_{\Omega_M^+} M^{q-p^*}|f(x)|^{p^*}dx\right|^2 \le M^{2q-2p^*}\left|\int_\Omega |f(x)|^{p^*}dx\right|^2
$$

$$
\lesssim M^{2q-2p^*},
$$

where the last step above again follows from the fact that $f \in W^{s,p}(\Omega) \subseteq L^{p^*}(\Omega)$. By substituting (C.19) and (C.20) into (C.18), we obtain that

$$
\mathbb{E}_{\{x_i\}_{i=1}^n, \{y_i\}_{i=1}^n}\left[\left|\hat{H}_M^q\left(\{x_i\}_{i=1}^n, \{y_i\}_{i=1}^n\right) - I_f^q\right|\right]
$$

$$
\le \sqrt{\mathbb{E}_{\{x_i\}_{i=1}^n, \{y_i\}_{i=1}^n}\left[\left|\hat{H}_M^q\left(\{x_i\}_{i=1}^n, \{y_i\}_{i=1}^n\right) - I_f^q\right|^2\right]} \lesssim \sqrt{\frac{M^{2q-p^*}}{n} + M^{2q-2p^*}}. \tag{C.21}
$$

By balancing the variance part $\frac{M^{2q-p^*}}{n}$ and the bias part $M^{2q-2p^*}$ above, we may get the optimal choice of $M$ as follows: $\frac{M^{2q-p^*}}{n} = M^{2q-2p^*} \Rightarrow M = \Theta(n^{\frac{1}{p^*}})$. Plugging in the optimal choice of

$M$ gives us the final upper bound:

$$\mathbb{E}_{\{x_i\}_{i=1}^n, \{y_i\}_{i=1}^n}\left[\left|\hat{H}_M^q\left(\{x_i\}_{i=1}^n, \{y_i\}_{i=1}^n\right) - I_f^q\right|\right] \lesssim \sqrt{n^{\frac{2q-2p^*}{p^*}}} = n^{\frac{q}{p^*}-1} = n^{-q(\frac{s}{d}-\frac{1}{p})-1},$$

(C.22)

which finishes our proof of Theorem 3.2.

# D   Proof of Minimax Lower and Upper Bounds in Section 4

This section is organized as follows. The first subsection consists of one important lemma used in our proof. In the second subsection, we provide complete proof for the minimax optimal lower bound on the estimation of integrals under any level of noise. In the third subsection, a complete proof for the upper bound on the estimation of integrals is given.

## D.1   A Key Lemma for Establishing the Upper Bound on Integral Estimation

**Lemma 6 (Bound on the Expected $k$-Nearest Neighbor Distance: Theorem 2.4, [69])** *Assume that $x_1, x_2, \cdots, x_n$ are independent and identical samples from the uniform distribution on the domain $\Omega = [0,1]^d$. For any $k \in \{1, 2, \cdots, n\}$ and $z \in \Omega$, we use $x_{i_k^{(z)}}$ to denote the $k$-th nearest neighbor of $z$ among $\{x_i\}_{i=1}^n$. When $z$ is also uniformly distributed over the domain $\Omega$, we have the following upper bound on the expected distance between $z$ and $x_{i_k^{(z)}}$:*

$$\mathbb{E}_{z, \{x_i\}_{i=1}^n}\left[\|z - x_{i_k^{(z)}}\|^2\right] \lesssim \left(\frac{k}{n}\right)^{\frac{2}{d}}.$$

(D.1)

## D.2   Proof of Theorem 4.1 (Lower Bound on Integral Estimation)

Here we present a comprehensive proof of the two lower bounds given in Theorem 4.1 above by applying the method of two fuzzy hypotheses (Lemma 5). Below we again use $\vec{x} := (x_1, x_2, \cdots, x_n)$ and $\vec{y} := (y_1, y_2, \cdots, y_n)$ to denote the two $n$-dimensional vectors formed by the quadrature points and observed function values. Since our lower bound in Theorem 4.1 consists of two terms, we need to prove the two bounds in the following two separate cases:

(Case I) For the first lower bound in (4.1), let's consider two constant functions $g_0$ and $g_1$ defined as follows:

$$g_0(x) \equiv 0 \ (\forall \ x \in \Omega), \ g_1(x) \equiv n^{-\gamma-\frac{1}{2}} \ (\forall \ x \in \Omega)$$

(D.2)

Clearly we have $g_0, g_1 \in C^s(\Omega)$. Then let's take $\mu_k$ to be a Dirac delta measure supported on the set $\{g_j\}$, *i.e*, $\mu_k(\{g_k\}) = 1$, for $k \in \{0,1\}$. By picking $c = \Delta = \frac{1}{2}I_{g_1} = \frac{1}{2}n^{-\gamma-\frac{1}{2}}$ and $\beta_0 = \beta_1 = 0$, we then obtain that

$$\begin{aligned}\mu_0(f \in W^{s,p}(\Omega) : I_f \leq c - \Delta) &= \mu_0(I_f \leq 0) = 1 = 1 - \beta_0, \\ \mu_1(f \in W^{s,p}(\Omega) : I_f \geq c + \Delta) &= \mu_1(I_f \geq I_{g_1}) = 1 = 1 - \beta_1,\end{aligned}$$

(D.3)

which indicates that (B.1) holds true. Now let's consider bounding the KL divergence between the two marginal distributions $\mathbb{P}_0, \mathbb{P}_1$ associated with $\mu_0, \mu_1$, respectively. Given that the quadrature points $\{x_i\}_{i=1}^n$ and the observational noises $\{\epsilon_i\}_{i=1}^n$ are independent and identical samples from the uniform distribution on $\Omega$ and the normal distribution $\mathcal{N}(0, n^{-2\gamma})$, we can write the marginal distributions in an explicit form as follows:

$$\mathbb{P}_0(\vec{x}, \vec{y}) = \prod_{i=1}^n \left(\frac{1}{\sqrt{2\pi}n^{-\gamma}} e^{-\frac{1}{2n^{-2\gamma}}y_i^2}\right), \ \mathbb{P}_1(\vec{x}, \vec{y}) = \prod_{i=1}^n \left(\frac{1}{\sqrt{2\pi}n^{-\gamma}} e^{-\frac{1}{2n^{-2\gamma}}(y_i - n^{-\gamma-\frac{1}{2}})^2}\right).$$

(D.4)

From (D.4) we can see that $\mathbb{P}_0$ and $\mathbb{P}_1$ are two $n$-dimensional normal distributions having the same covariance matrix but different mean vectors. Computing the KL divergence between them and applying Pinsker's inequality then give us that

$$TV(\mathbb{P}_0\|\mathbb{P}_1) \leq \sqrt{\frac{1}{2}KL(\mathbb{P}_0\|\mathbb{P}_1)} = \sqrt{\frac{n(n^{-\gamma-\frac{1}{2}})^2}{4n^{-2\gamma}}} = \frac{1}{2}.$$

(D.5)

Substituting (D.5), $\Delta = \frac{1}{2}I_{g_1} = \frac{1}{2}n^{-\gamma-\frac{1}{2}}$ and $\beta_0 = \beta_1 = 0$ into (B.2) and applying Markov's inequality yield the final lower bound

$$
\inf_{\hat{H}\in\mathcal{H}_n^f} \sup_{f\in C^s(\Omega)} \mathbb{E}_{\{x_i\}_{i=1}^n,\{y_i\}_{i=1}^n}\left[\left|\hat{H}\left(\{x_i\}_{i=1}^n,\{y_i\}_{i=1}^n\right)-I_f\right|\right]
$$

$$
\geq \Delta \inf_{\hat{H}\in\mathcal{H}_n^f} \sup_{f\in C^s(\Omega)} \mathbb{P}_{\{x_i\}_{i=1}^n,\{y_i\}_{i=1}^n}\left[\left|\hat{H}\left(\{x_i\}_{i=1}^n,\{y_i\}_{i=1}^n\right)-I_f\right|\geq\Delta\right] \tag{D.6}
$$

$$
\geq \frac{1}{2}I_{g_1}\frac{1-TV(\mathbb{P}_0||\mathbb{P}_1)-\beta_0-\beta_1}{2} \geq \frac{1}{8}n^{-\gamma-\frac{1}{2}} \gtrsim n^{-\gamma-\frac{1}{2}},
$$

which is exactly the first term in the RHS of (4.1).

(Case II) For the second lower bound in (4.1), our proof is similar to the proof of the second lower bound in Theorem 2.1 presented in Appendix B.2 above. Again, we select $m = (200n)^{\frac{1}{d}}$ and divide the domain $\Omega$ into $m^d$ small cubes $\Omega_1, \Omega_2, \cdots, \Omega_{m^d}$, each of which has side length $m^{-1}$. For any $1 \leq j \leq m^d$, we use $c_j$ to denote center of the cube $\Omega_j$. Then let's consider the same bump function $K$ defined in (B.3) and (B.4) above, which satisfies $\text{supp}(K) \subseteq [-\frac{1}{2},\frac{1}{2}]^d$ and $K \in C^\infty([-\frac{1}{2},\frac{1}{2}]^d)$. In an analogous way, for any $1 \leq j \leq m^d$, we associate each cube $\Omega_j$ with a bump function $f_j$ defined as follows:

$$
f_j(x) = \begin{cases} m^{-s}K(m(x-c_j)) \ (x\in\Omega_j), \\ 0 \ (\text{otherwise}), \end{cases} \tag{D.7}
$$

where $\text{supp}(f_j) \subseteq \Omega_j$, $f_j \in C^\infty(\Omega)$ and $f_j(x) \geq 0$ $(\forall\, x \in \Omega)$. Then let's consider the following finite set of $2^{m^d}$ functions:

$$
\mathcal{S} := \Big\{ \sum_{j=1}^{m^d} \eta_j f_j : \eta_j \in \{\pm 1\}, \, \forall\, 1 \leq j \leq m^d \Big\}. \tag{D.8}
$$

We will first verify that $\mathcal{S} \subseteq C^s(\Omega)$. Fix any element $f_* = \sum_{j=1}^{m^d} \eta_j f_j \in \mathcal{S}$. On the one hand, from our construction of the $f_j$'s given in (D.7) above, we have

$$
\max_{|t|\leq\lfloor s\rfloor} \|D^t f_*\|_{L^\infty(\Omega)} = \max_{|t|\leq\lfloor s\rfloor} m^{-s+|t|}\|D^t K\|_{L^\infty([-\frac{1}{2},\frac{1}{2}]^d)}
$$

$$
\leq \max_{|t|\leq\lfloor s\rfloor} \|D^t K\|_{L^\infty([-\frac{1}{2},\frac{1}{2}]^d)}. \tag{D.9}
$$

On the other hand, for any $1 \leq i \neq j \leq m^d$, we consider the function $\psi_i f_i + \psi_j f_j$, where the scalars $\psi_j, \psi_j \in \{0, \pm 1\}$. Now let's may pick $\beta := \frac{d}{1-\{s\}}$, where $\{s\} = s - \lfloor s\rfloor \in (0,1)$ denotes the fractional part of $s$. Given that $f_j \in C^\infty(\Omega)$, we may upper bound the Sobolev norm $\|\cdot\|_{W^{1,\beta}}$ of the function $D^t(\psi_i f_i + \psi_j f_j)$ for any $t \in \mathbb{N}_0^d$ satisfying $|t| = \lfloor s\rfloor$ as follows:

$$
\left\|D^t(\psi_i f_i + \psi_j f_j)\right\|_{W^{1,\beta}(\Omega)}^\beta = |\psi_i|^\beta\left\|D^t f_i\right\|_{W^{1,\beta}(\Omega_i)}^\beta + |\psi_j|^\beta\left\|D^t f_j\right\|_{W^{1,\beta}(\Omega_j)}^\beta
$$

$$
\leq \left\|D^t f_i\right\|_{L^\beta(\Omega_i)}^\beta + \sum_{r=1}^d\left\|\frac{\partial}{\partial x_r}D^t f_i\right\|_{L^\beta(\Omega_i)}^\beta + \left\|D^{\lfloor s\rfloor} f_j\right\|_{L^\beta(\Omega_j)}^\beta + \sum_{r=1}^d\left\|\frac{\partial}{\partial x_r}D^t f_j\right\|_{L^\beta(\Omega_j)}^\beta
$$

$$
= \sum_{l\in\{i,j\}}\int_{\Omega_l}\left(m^{-s+|t|}D^t K(m(x-c_l))\right)^\beta dx \tag{D.10}
$$

$$
+ \sum_{l\in\{i,j\}}\sum_{r=1}^d\int_{\Omega_l}\left(m^{-s+|t|+1}\frac{\partial}{\partial x_r}D^t K(m(x-c_l))\right)^\beta dx.
$$

From our choice of $\beta$ and assumption on the bump function $K$, we may further upper bound the Sobolev norm $\left\|D^t(\psi_i f_i + \psi_j f_j)\right\|_{W^{1,\beta}(\Omega)}$ as below:

$$\left\|D^t(\psi_i f_i + \psi_j f_j)\right\|_{W^{1,\beta}(\Omega)}^{\beta} \le \sum_{l \in \{i,j\}} m^{-\beta\{s\}} \int_{[-\frac{1}{2},\frac{1}{2}]^d} \left(D^t K(y)\right)^{\beta} \frac{1}{m^d} dy$$

$$+ \sum_{l \in \{i,j\}} dm^{\beta(1-\{s\})} \sup_{|t'| \le \lfloor s \rfloor + 1} \left(\int_{[-\frac{1}{2},\frac{1}{2}]^d} \left(D^{t'} K(y)\right)^{\beta} \frac{1}{m^d} dy\right)$$

$$\le 2m^{-\beta\{s\}-d} \left\|D^t K\right\|_{L^{\beta}([-\frac{1}{2},\frac{1}{2}]^d)}^{\beta} + 2dm^{\beta(1-\{s\})-d} \cdot \sup_{|t'| \le \lfloor s \rfloor + 1} \left\|D^{t'} K\right\|_{L^{\beta}([-\frac{1}{2},\frac{1}{2}]^d)}^{\beta}$$

$$\lesssim \left\|D^t K\right\|_{L^{\beta}([-\frac{1}{2},\frac{1}{2}]^d)}^{\beta} + \sup_{|t'| \le \lfloor s \rfloor + 1} \left\|D^{t'} K\right\|_{L^{\beta}([-\frac{1}{2},\frac{1}{2}]^d)}^{\beta},$$

(D.11)

where the last inequality above follows from our choice of $\beta$. From (D.11) and the second part of the Sobolev Embedding Theorem (Lemma 1), we can deduce that $D^t(\psi_i f_i + \psi_j f_j) \in C^1(\Omega) \cap W^{1,\frac{d}{1-\{s\}}}(\Omega) \subseteq C^{\{s\}}(\Omega)$ and the following inequality holds:

$$\left\|D^t(\psi_i f_i + \psi_j f_j)\right\|_{C^{\{s\}}(\Omega)} \lesssim \left\|D^t(\psi_i f_i + \psi_j f_j)\right\|_{W^{1,\beta}(\Omega)}$$

$$\lesssim \left(\sup_{|t'|=\lfloor s \rfloor} \left\|D^{t'} K\right\|_{L^{\beta}([-\frac{1}{2},\frac{1}{2}]^d)}^{\beta} + \sup_{|t'|=\lfloor s \rfloor+1} \left\|D^{t'} K\right\|_{L^{\beta}([-\frac{1}{2},\frac{1}{2}]^d)}^{\beta}\right)^{\frac{1}{\beta}},$$

(D.12)

Furthermore, combining (D.12) with our construction of the $f_j$'s given in (D.7) above gives us that

$$\max_{|t|=\lfloor s \rfloor} \sup_{x,y \in \Omega, x \ne y} \frac{|D^t f_*(x) - D^t f_*(y)|}{\|x-y\|^{s-\lfloor s \rfloor}}$$

$$\le \max_{\substack{1 \le i \ne j \le k \\ \psi_i, \psi_j \in \{0,\pm 1\}}} \max_{|t|=\lfloor s \rfloor} \sup_{x \ne y \in \Omega} \frac{|D^t(\psi_i f_i + \psi_j f_j)(x) - D^t(\psi_i f_i + \psi_j f_j)(x)|}{\|x-y\|^{\{s\}}}$$

$$\le \max_{\substack{1 \le i \ne j \le k, |t|=\lfloor s \rfloor \\ \psi_i, \psi_j \in \{0,\pm 1\}}} \left\|D^t(\psi_i f_i + \psi_j f_j)\right\|_{C^{\{s\}}(\Omega)}$$

$$\lesssim \left(\sup_{|t'|=\lfloor s \rfloor} \left\|D^{t'} K\right\|_{L^{\beta}([-\frac{1}{2},\frac{1}{2}]^d)}^{\beta} + \sup_{|t'|=\lfloor s \rfloor+1} \left\|D^{t'} K\right\|_{L^{\beta}([-\frac{1}{2},\frac{1}{2}]^d)}^{\beta}\right)^{\frac{1}{\beta}}$$

(D.13)

Finally, adding the two inequalities (D.9) and (D.13) gives us that for any $f_* \in \mathcal{S}$, we have

$$\|f_*\|_{C^s(\Omega)} = \max_{|t| \le \lfloor s \rfloor} \|D^t f_*\|_{L^{\infty}(\Omega)} + \max_{|t|=\lfloor s \rfloor} \sup_{x,y \in \Omega, x \ne y} \frac{|D^t f_*(x) - D^t f_*(y)|}{\|x-y\|^{s-\lfloor s \rfloor}}$$

$$\lesssim \max_{|t| \le \lfloor s \rfloor} \|D^t K\|_{L^{\infty}([-\frac{1}{2},\frac{1}{2}]^d)}$$

$$+ \left(\sup_{|t'|=\lfloor s \rfloor} \left\|D^{t'} K\right\|_{L^{\beta}([-\frac{1}{2},\frac{1}{2}]^d)}^{\beta} + \sup_{|t'|=\lfloor s \rfloor+1} \left\|D^{t'} K\right\|_{L^{\beta}([-\frac{1}{2},\frac{1}{2}]^d)}^{\beta}\right)^{\frac{1}{\beta}} \lesssim 1.$$

(D.14)

From the arbitrariness of $f_*$, we can then deduce that $\mathcal{S} \subseteq C^s(\Omega)$, as desired. For any $p \in (0,1)$, below we again use $w_p$ to denote the discrete random variable satisfying $\mathbb{P}(w_p = -1) = p$ and $\mathbb{P}(w_p = 1) = 1-p$. Now let's pick $\kappa = \frac{1}{3}\sqrt{\frac{2}{3n}}$ and take $\{w_j^{(0)}\}_{j=1}^{m^d}$ and $\{w_j^{(1)}\}_{j=1}^{m^d}$ to be independent and identical copies of $w_{\frac{1+\kappa}{2}}$ and $w_{\frac{1-\kappa}{2}}$ respectively. Then we define $\mu_0, \mu_1$ to be two discrete measures supported on the finite set $\mathcal{S}$ such that the following condition holds for any $\eta_j \in \{\pm 1\}$ $(1 \le j \le m^d)$:

$$\mu_k\left(\left\{\sum_{j=1}^{m^d} \eta_j f_j\right\}\right) = \prod_{j=1}^{m^d} \mathbb{P}(w_j^{(k)} = \eta_j), \; k \in \{0,1\}.$$

(D.15)

Then we proceed to determine the separation distance $\Delta$ between the two priors $\mu_0$ and $\mu_1$. Similar to what we did in the proof of Theorem 2.1, we need to first define the following quantity $C := \int_{\Omega_j} f_j(x)dx$, which remains the same for any $1 \leq j \leq m^d$. Moreover, applying (D.7) helps us evaluate the quantity $C$ directly as follows

$$
\begin{aligned}
C &= \int_{\Omega_j} f_j(x)dx = \int_{\Omega_j} m^{-s} K(m(x - c_j))dx \\
&= m^{-s} \int_{[-\frac{1}{2}, \frac{1}{2}]^d} K(y)\frac{1}{m^d}dy = m^{-s-d}\|K\|_{L^1([-\frac{1}{2}, \frac{1}{2}]^d)}.
\end{aligned}
\tag{D.16}
$$

Moreover, by picking $\lambda = \frac{1}{2}$, we may apply Hoeffding's Inequality (Lemma 3) to the bounded random variables $\{w_j^{(0)}\}_{j=1}^{m^d}$ and $\{w_j^{(1)}\}_{j=1}^{m^d}$ to deduce that

$$
\begin{aligned}
\mathbb{P}\Big(\sum_{j=1}^{m^d} w_j^{(0)} \geq -(1-\lambda)m^d\kappa\Big) &\leq \exp\Big(-\frac{2(\lambda m^d\kappa)^2}{4m^d}\Big) = \exp\Big(-\frac{1}{2}\lambda^2\kappa^2 m^d\Big), \\
\mathbb{P}\Big(\sum_{j=1}^{m^d} w_j^{(1)} \leq (1-\lambda)m^d\kappa\Big) &\leq \exp\Big(-\frac{2(\lambda m^d\kappa)^2}{4m^d}\Big) = \exp\Big(-\frac{1}{2}\lambda^2\kappa^2 m^d\Big).
\end{aligned}
\tag{D.17}
$$

By taking $c := 0, \Delta := (1-\lambda)\kappa m^d C$ and $\beta_0 = \beta_1 = \exp\Big(-\frac{1}{2}\lambda^2\kappa^2 m^d\Big)$, we may use (D.17) justified above to get that

$$
\begin{aligned}
\mu_0(f \in C^s(\Omega) : I_f \leq c - \Delta) &= \mathbb{P}\Big(\sum_{j=1}^{m^d} I_{w_j^{(0)}f_j} \leq \frac{1-(1-\lambda)\kappa}{2}m^d C - \frac{1+(1-\lambda)\kappa}{2}m^d C\Big) \\
&\geq \mathbb{P}\Big(\sum_{j=1}^{m^d} w_j^{(0)} \leq -(1-\lambda)m^d\kappa\Big) = 1 - \mathbb{P}\Big(\sum_{j=1}^{m^d} w_j^{(0)} \geq -(1-\lambda)m^d\kappa\Big) \\
&\geq 1 - \exp\Big(-\frac{1}{2}\lambda^2\kappa^2 m^d\Big) = 1 - \beta_0, \\
\mu_1(f \in C^s(\Omega) : I_f \geq c + \Delta) &= \mathbb{P}\Big(\sum_{j=1}^{m^d} I_{w_j^{(1)}f_j} \geq \frac{1+(1-\lambda)\kappa}{2}m^d C - \frac{1-(1-\lambda)\kappa}{2}m^d C\Big) \\
&\geq \mathbb{P}\Big(\sum_{j=1}^{m^d} w_j^{(1)} \geq (1-\lambda)m^d\kappa\Big) = 1 - \mathbb{P}\Big(\sum_{j=1}^{m^d} w_j^{(0)} \leq (1-\lambda)m^d\kappa\Big) \\
&\geq 1 - \exp\Big(-\frac{1}{2}\lambda^2\kappa^2 m^d\Big) = 1 - \beta_1,
\end{aligned}
\tag{D.18}
$$

which indicates that (B.1) holds true. Now let's consider bounding the KL divergence between the two marginal distributions $\mathbb{P}_0, \mathbb{P}_1$ associated with $\mu_0, \mu_1$, respectively. Applying the fact that $\{x_1, \cdots, x_n\}$ and $\{\epsilon_1, \cdots, \epsilon_n\}$ are identical and independent samples from the uniform distribution on $\Omega$ and the normal distribution $\mathcal{N}(0, n^{-2\gamma})$ allows us to write the marginal distributions in an explicit form as follows:

$$
\mathbb{P}_0(\vec{x}, \vec{y}) = \prod_{j=1}^{m^d}\Big(\frac{1-\kappa}{2}\prod_{i:x_i\in\Omega_j}\frac{1}{\sqrt{2\pi}n^{-\gamma}}e^{-\frac{(y_i-f_j(x_i))^2}{2n^{-2\gamma}}} + \frac{1+\kappa}{2}\prod_{i:x_i\in\Omega_j}\frac{1}{\sqrt{2\pi}n^{-\gamma}}e^{-\frac{(y_i+f_j(x_i))^2}{2n^{-2\gamma}}}\Big),
$$

$$
\mathbb{P}_1(\vec{x}, \vec{y}) = \prod_{j=1}^{m^d}\Big(\frac{1+\kappa}{2}\prod_{i:x_i\in\Omega_j}\frac{1}{\sqrt{2\pi}n^{-\gamma}}e^{-\frac{(y_i-f_j(x_i))^2}{2n^{-2\gamma}}} + \frac{1-\kappa}{2}\prod_{i:x_i\in\Omega_j}\frac{1}{\sqrt{2\pi}n^{-\gamma}}e^{-\frac{(y_i+f_j(x_i))^2}{2n^{-2\gamma}}}\Big).
$$

$$\tag{D.19}$$

Furthermore, for any $n$ fixed quadrature points $\vec{x} = (x_1, x_2, \cdots, x_n)$, we use $\mathbb{P}_k(\cdot \mid \vec{x})$ to denote the marginal distribution of the observed function values $\vec{y} = (y_1, y_2, \cdots, y_n)$ conditioned on $\vec{x}$ for $k \in \{0, 1\}$. Since $\{x_i\}_{i=1}^n$ are identically and independently sampled from the uniform distribution

on $\Omega$, we have that the two probability densities $\mathbb{P}_k(\vec{x}, \vec{y})$ and $\mathbb{P}_k(\vec{y} \mid \vec{x})$ have the same mathematical expression for any $k \in \{0, 1\}$. Then we may further rewrite the KL divergence between the two marginal distributions $\mathbb{P}_0, \mathbb{P}_1$ as follows:

$$
\begin{aligned}
KL(\mathbb{P}_0 || \mathbb{P}_1) &= \int_\Omega \cdots \int_\Omega \Big( \int_{-\infty}^\infty \cdots \int_{-\infty}^\infty \log\Big(\frac{\mathbb{P}_0(\vec{x}, \vec{y})}{\mathbb{P}_1(\vec{x}, \vec{y})}\Big) \mathbb{P}_0(\vec{x}, \vec{y}) dy_1 \cdots dy_n \Big) dx_1 \cdots dx_n \\
&= \int_\Omega \cdots \int_\Omega \Big( \int_{-\infty}^\infty \cdots \int_{-\infty}^\infty \log\Big(\frac{\mathbb{P}_0(\vec{y} \mid \vec{x})}{\mathbb{P}_1(\vec{y} \mid \vec{x})}\Big) \mathbb{P}_0(\vec{y} \mid \vec{x}) dy_1 \cdots dy_n \Big) dx_1 \cdots dx_n \\
&= \int_\Omega \cdots \int_\Omega \Big( KL\Big(\mathbb{P}_0(\cdot \mid \vec{x}) || \mathbb{P}_1(\cdot \mid \vec{x})\Big)\Big) dx_1 \cdots dx_n.
\end{aligned}
$$
(D.20)

It now remains to upper bound the KL divergence between the two conditional distributions $\mathbb{P}_0(\cdot \mid \vec{x})$ and $\mathbb{P}_1(\cdot \mid \vec{x})$ for any fixed $\vec{x} = (x_1, \cdots, x_n)$. In order to derive such an upper bound, we need to introduce the following notations first. For any $n$ quadrature points $\{x_i\}_{i=1}^n$, we use $\mathcal{J}_n$ to denote the set of all indices $j$ satisfying that $\Omega_j$ contains at least one of the points in $\{x_i\}_{i=1}^n$, i.e,

$$
\mathcal{J}_n := \mathcal{J}_n(\vec{x}) = \Big\{ j : 1 \le j \le m^d \text{ and } \Omega_j \cap \{x_1, \cdots, x_n\} \ne \varnothing \Big\}. \tag{D.21}
$$

Moreover, we use $\vec{\omega}_{\mathcal{J}_n}^{(k)}$ to denote $|\mathcal{J}_n|$-dimensional vector formed by the random variables $\{\omega_j^{(k)} : j \in \mathcal{J}_n\}$ and $p_{\mathcal{J}_n}^{(k)}(\cdot)$ to denote the probability density function of $\vec{\omega}_{\mathcal{J}_n}^{(k)}$, where $k \in \{0, 1\}$. From our assumption on the distribution of the weights $\{w_j^{(0)}\}_{j=1}^n$ and $\{w_j^{(1)}\}_{j=1}^n$, we have that for any $\vec{\omega}_{\mathcal{J}_n} \in \{\pm 1\}^{|\mathcal{J}_n|}$,

$$
\begin{aligned}
p_{\mathcal{J}_n}^{(0)}(\vec{\omega}_{\mathcal{J}_n}) &= \prod_{j \in \mathcal{J}_n} \Big(\frac{1+\kappa}{2}\Big)^{\frac{1}{2}(1-\omega_j)} \Big(\frac{1-\kappa}{2}\Big)^{\frac{1}{2}(1+\omega_j)} \\
p_{\mathcal{J}_n}^{(1)}(\vec{\omega}_{\mathcal{J}_n}) &= \prod_{j \in \mathcal{J}_n} \Big(\frac{1+\kappa}{2}\Big)^{\frac{1}{2}(1+\omega_j)} \Big(\frac{1-\kappa}{2}\Big)^{\frac{1}{2}(1-\omega_j)}
\end{aligned}
\tag{D.22}
$$

Furthermore, for any fixed quadrature points $\vec{x} = (x_1, \cdots, x_n)$ and weights $\vec{\omega}_{\mathcal{J}_n} := \{\omega_j : j \in \mathcal{J}_n\} \subseteq \{\pm 1\}^{|\mathcal{J}_n|}$, we may define the transition kernel $G(\vec{x}, \vec{\omega}_{\mathcal{J}_n})$ as below

$$
G(\vec{x}, \vec{\omega}_{\mathcal{J}_n}) := \prod_{j \in \mathcal{J}_n} \Big( \prod_{i:x_i \in \Omega_j} \frac{1}{\sqrt{2\pi n^{-\gamma}}} e^{-\frac{(y_i + \omega_j f_j(x_i))^2}{2n^{-2\gamma}}}\Big) \tag{D.23}
$$

Combining the expressions in (D.19),(D.22) and (D.23) allows us to rewrite the two conditional distributions $\mathbb{P}_k(\cdot \mid \vec{x})$ as below:

$$
\mathbb{P}_k(\vec{y} \mid \vec{x}) = \mathbb{P}_k(\vec{x}, \vec{y}) = \int_{\{\pm 1\}^{|\mathcal{J}_n|}} G(\vec{x}, \vec{\omega}_{\mathcal{J}_n}) p_{\mathcal{J}_n}^{(k)}(\vec{\omega}_{\mathcal{J}_n}) d\vec{\omega}_{\mathcal{J}_n} \tag{D.24}
$$

where $k \in \{0, 1\}$. Applying the data processing inequality (Lemma 4) to (D.24) above then enables us to derive the following upper bound on $KL\Big(\mathbb{P}_0(\cdot \mid \vec{x}) || \mathbb{P}_1(\cdot \mid \vec{x})\Big)$ for any $n$ fixed quadrature points $\vec{x} = (x_1, \cdots, x_n)$:

$$
\begin{aligned}
KL\Big(\mathbb{P}_0(\cdot \mid \vec{x}) || \mathbb{P}_1(\cdot \mid \vec{x})\Big) &\le KL\Big(p_{\mathcal{J}_n}^{(0)} \| p_{\mathcal{J}_n}^{(1)}\Big) \\
&= |\mathcal{J}_n|\Big( \log\Big(\frac{1+\kappa}{1-\kappa}\Big)\frac{1+\kappa}{2} + \log\Big(\frac{1-\kappa}{1+\kappa}\Big)\frac{1-\kappa}{2}\Big) \\
&\le n\kappa \log\Big(\frac{1+\kappa}{1-\kappa}\Big)
\end{aligned}
\tag{D.25}
$$

where the equality in (D.25) above follows from the fact that $\{w_j^{(0)}\}_{j=1}^{m^d}$ and $\{w_j^{(1)}\}_{j=1}^{m^d}$ are independent and identical copies of $w_{\frac{1+\kappa}{2}}$ and $w_{\frac{1-\kappa}{2}}$ respectively. The last inequality of (D.25) above, however, is deduced from the fact that $m^d = 200n > n$, which implies $|\mathcal{J}_n| \le n$ for any $n$ quadrature points $\{x_i\}_{i=1}^n$. Substituting (D.25) into (D.20) and applying Pinkser's inequality yields the final upper bound on the TV distance between $\mathbb{P}_0$ and $\mathbb{P}_1$:

$$TV(\mathbb{P}_0\|\mathbb{P}_1) \leq \sqrt{\frac{1}{2}KL(\mathbb{P}_0\|\mathbb{P}_1)} \leq \sqrt{\int_\Omega \cdots \int_\Omega \frac{n\kappa}{2}\log\left(\frac{1+\kappa}{1-\kappa}\right)dx_1\cdots dx_n}$$

$$= \sqrt{\frac{n\kappa}{2}\log\left(\frac{1+\kappa}{1-\kappa}\right)} \leq \sqrt{\frac{3n}{2}}\kappa = \frac{1}{3}. \tag{D.26}$$

Finally, by substituting (D.16), (D.26), $\Delta = (1-\lambda)\kappa m^d C$ and $\beta_0 = \beta_1 = \exp\left(-\frac{1}{2}\lambda^2\kappa^2 m^d\right) = \exp(-\frac{50}{27}) < \frac{1}{6}$ into (B.2) and applying Markov's inequality, we obtain the final lower bound

$$\inf_{\hat{H}\in\mathcal{H}_n^f}\sup_{f\in C^s(\Omega)}\mathbb{E}_{\{x_i\}_{i=1}^n,\{y_i\}_{i=1}^n}\left[\left|\hat{H}\left(\{x_i\}_{i=1}^n,\{y_i\}_{i=1}^n\right)-I_f\right|\right]$$

$$\geq \Delta \inf_{\hat{H}\in\mathcal{H}_n^f}\sup_{f\in C^s(\Omega)}\mathbb{P}_{\{x_i\}_{i=1}^n,\{y_i\}_{i=1}^n}\left[\left|\hat{H}\left(\{x_i\}_{i=1}^n,\{y_i\}_{i=1}^n\right)-I_f\right|\geq\Delta\right] \tag{D.27}$$

$$\geq (1-\lambda)\kappa m^d C\frac{1-TV(\mathbb{P}_0\|\mathbb{P}_1)-\beta_0-\beta_1}{2} \geq \frac{1}{2}\frac{\sqrt{2}}{3\sqrt{3n}}\cdot(200n)\cdot\frac{C}{6}$$

$$\gtrsim \sqrt{n}C \gtrsim \sqrt{n}(200n)^{-\frac{s+d}{d}}\|K\|_{L^1([-\frac{1}{2},\frac{1}{2}]^d)} \gtrsim n^{-\frac{s}{d}-\frac{1}{2}},$$

which is exactly the second term in the RHS of (4.1). Combining the two lower bounds proved in (D.6) and (D.27) concludes our proof of Theorem 4.1

## D.3 Proof of Theorem 4.2 (Upper Bound on Integral Estimation)

Before proving the upper bound on integral estimation, we need to derive an upper bound on the expected error of the $k$-nearest neighbor estimator $\hat{f}_{k\text{-NN}}$, which is built based on the first half of the given dataset $\{(x_i,y_i)\}_{i=1}^n$, with respect to the $L^2$ norm. From our construction of $\hat{f}_{k\text{-NN}}$ given in Section 4.2, we have that for any fixed $\frac{n}{2}$ quadrature points $\{x_i\}_{i=1}^{\frac{n}{2}}$, $z\in\Omega$ and $k\in\{1,2,\cdots,\frac{n}{2}\}$, the expected value of $\hat{f}_{k\text{-NN}}(z)$ with respect to the observational noises $\{\epsilon_i\}_{i=1}^{\frac{n}{2}}$ is given by

$$\mathbb{E}_{\{\epsilon_i\}_{i=1}^{\frac{n}{2}}}\left[\hat{f}_{k\text{-NN}}(z)\right] = \frac{1}{k}\sum_{j=1}^k\mathbb{E}_{\{\epsilon_i\}_{i=1}^{\frac{n}{2}}}\left[f(x_{i_j^{(z)}})+\epsilon_{i_j^{(z)}}\right] = \frac{1}{k}\sum_{j=1}^k f(x_{i_j^{(z)}}), \tag{D.28}$$

where $\{x_{i_j^{(z)}}\}_{j=1}^k$ above are the $k$ nearest neighbors of $z$ among $\{x_i\}_{i=1}^{\frac{n}{2}}$. Now let's consider using the bias-variance decomposition to upper bound the error $\|\hat{f}_{k\text{-NN}}(z)-f(z)\|_{L^2(\Omega)}^2$. Based on the expected value computed in (D.28) above, we may decompose the function $\hat{f}_{k\text{-NN}}-f$ as a sum of the bias part and the variance part as follows:

$$B(z) := \mathbb{E}_{\{\epsilon_i\}_{i=1}^{\frac{n}{2}}}\left[\hat{f}_{k\text{-NN}}(z)\right]-f(z) = \frac{1}{k}\sum_{j=1}^k f(x_{i_j^{(z)}})-f(z) = \frac{1}{k}\sum_{j=1}^k\left(f(x_{i_j^{(z)}})-f(z)\right), \tag{D.29}$$

$$V(z) := \hat{f}_{k\text{-NN}}(z)-\mathbb{E}_{\{\epsilon_i\}_{i=1}^{\frac{n}{2}}}\left[\hat{f}_{k\text{-NN}}(z)\right] = \hat{f}_{k\text{-NN}}(z)-\frac{1}{k}\sum_{j=1}^k f(x_{i_j^{(z)}}) = \frac{1}{k}\sum_{j=1}^k\epsilon_{i_j^{(z)}}, \tag{D.30}$$

where the function $B$ corresponds to the bias part and the function $V$ corresponds to the variance part. Using the decomposition $\hat{f}_{k\text{-NN}}-f = B+V$ allows us to upper bound the expected error of

$\hat{f}_{k\text{-NN}}$ with respect to the $L^2$ norm as below:

$$\mathbb{E}_{\{x_i\}_{i=1}^{\frac{n}{2}},\{y_i\}_{i=1}^{\frac{n}{2}}}\left[\|\hat{f}_{k\text{-NN}} - f\|_{L^2(\Omega)}^2\right] = \mathbb{E}_{\{x_i\}_{i=1}^{\frac{n}{2}},\{y_i\}_{i=1}^{\frac{n}{2}}}\left[\|B + V\|_{L^2(\Omega)}^2\right]$$

$$\leq \mathbb{E}_{\{x_i\}_{i=1}^{\frac{n}{2}},\{y_i\}_{i=1}^{\frac{n}{2}}}\left[\left(\|B\|_{L^2(\Omega)} + \|V\|_{L^2(\Omega)}\right)^2\right]$$

$$\leq \mathbb{E}_{\{x_i\}_{i=1}^{\frac{n}{2}},\{y_i\}_{i=1}^{\frac{n}{2}}}\left[2\|B\|_{L^2(\Omega)}^2 + 2\|V\|_{L^2(\Omega)}^2\right] \tag{D.31}$$

$$\lesssim \mathbb{E}_{\{x_i\}_{i=1}^{\frac{n}{2}},\{y_i\}_{i=1}^{\frac{n}{2}}}\left[\|V\|_{L^2(\Omega)}^2\right] + \mathbb{E}_{\{x_i\}_{i=1}^{\frac{n}{2}},\{y_i\}_{i=1}^{\frac{n}{2}}}\left[\|B\|_{L^2(\Omega)}^2\right]$$

$$= \mathbb{E}_{z,\{x_i\}_{i=1}^{\frac{n}{2}},\{\epsilon_i\}_{i=1}^{\frac{n}{2}}}\left[|V(z)|^2\right] + \mathbb{E}_{z,\{x_i\}_{i=1}^{\frac{n}{2}},\{\epsilon_i\}_{i=1}^{\frac{n}{2}}}\left[|B(z)|^2\right],$$

where $z$ above is uniformly distributed over the domain $\Omega$ and independent of $x_i$ for any $1 \leq i \leq \frac{n}{2}$. On the one hand, using the expression of the variance part $V$ derived in (D.30) above and the fact that $\{\epsilon_i\}_{i=1}^{\frac{n}{2}}$ are independent and identical distributed noises, we may compute the first term in (D.31) above as follows:

$$\mathbb{E}_{z,\{x_i\}_{i=1}^{\frac{n}{2}},\{\epsilon_i\}_{i=1}^{\frac{n}{2}}}\left[|V(z)|^2\right] = \mathbb{E}_z\left[\mathbb{E}_{\{x_i\}_{i=1}^{\frac{n}{2}},\{\epsilon_i\}_{i=1}^{\frac{n}{2}}}\left[\left|\frac{1}{k}\sum_{j=1}^{k}\epsilon_{i_j^{(z)}}\right|^2\right]\right]$$

$$= \mathbb{E}_z\left[\frac{1}{k^2}\mathbb{E}_{\{x_i\}_{i=1}^{\frac{n}{2}},\{\epsilon_i\}_{i=1}^{\frac{n}{2}}}\left[\sum_{j=1}^{k}\epsilon_{i_j^{(z)}}^2\right]\right] \tag{D.32}$$

$$= \mathbb{E}_z\left[\frac{n^{-2\gamma}k}{k^2}\right] = \frac{n^{-2\gamma}}{k}.$$

On the other hand, since $s \in (0,1)$ and the given function $f$ is $s$-Hölder smooth, we have that the inequality $|f(x) - f(y)| \lesssim \|x - y\|^s$ holds true for any $x, y \in \Omega$. Combining this inequality with the expression of the bias part $B$ derived in (D.30) above helps us upper bound the second term in (D.31) as below:

$$\mathbb{E}_{z,\{x_i\}_{i=1}^{\frac{n}{2}},\{\epsilon_i\}_{i=1}^{\frac{n}{2}}}\left[|B(z)|^2\right] = \mathbb{E}_z\left[\mathbb{E}_{\{x_i\}_{i=1}^{\frac{n}{2}},\{\epsilon_i\}_{i=1}^{\frac{n}{2}}}\left[\left|\frac{1}{k}\sum_{j=1}^{k}\left(f(x_{i_j^{(z)}}) - f(z)\right)\right|^2\right]\right]$$

$$\leq \frac{1}{k}\mathbb{E}_z\left[\mathbb{E}_{\{x_i\}_{i=1}^{\frac{n}{2}}}\left[\sum_{j=1}^{k}\left|f(x_{i_j^{(z)}}) - f(z)\right|^2\right]\right]$$

$$\lesssim \frac{1}{k}\mathbb{E}_z\left[\mathbb{E}_{\{x_i\}_{i=1}^{\frac{n}{2}}}\left[\sum_{j=1}^{k}\left|x_{i_j^{(z)}} - z\right|^{2s}\right]\right] \tag{D.33}$$

$$\leq \mathbb{E}_{z,\{x_i\}_{i=1}^{\frac{n}{2}}}\left[\left|x_{i_k^{(z)}} - z\right|^{2s}\right]$$

$$\leq \left(\mathbb{E}_{z,\{x_i\}_{i=1}^{\frac{n}{2}}}\left[\left|x_{i_k^{(z)}} - z\right|^2\right]\right)^s \lesssim \left(\frac{k}{n}\right)^{\frac{2s}{d}}.$$

The second least inequality follows from the fact that $\omega(t) := t^s$ is a concave function when $s \in (0,1)$, while the last inequality is obtained by plugging in (D.1) given in Lemma 6. Substituting (D.32) and (D.33) into (D.31) then yields that for any $k \in \{1, 2, \cdots, \frac{n}{2}\}$, the expected error of $\hat{f}_{k\text{-NN}}$ with respect to the $L^2$ norm can be upper bounded as follows:

$$\mathbb{E}_{\{x_i\}_{i=1}^{\frac{n}{2}},\{y_i\}_{i=1}^{\frac{n}{2}}}\left[\|\hat{f}_{k\text{-NN}} - f\|_{L^2(\Omega)}^2\right] \lesssim \frac{n^{-2\gamma}}{k} + \left(\frac{k}{n}\right)^{\frac{2s}{d}}. \tag{D.34}$$

Furthermore, from our construction of the integral estimator $\hat{H}_{k\text{-NN}}$ given in Section 4.2, we may upper bound the expectation of the estimator $\hat{H}_{k\text{-NN}}$'s squared error via the expected error of $\hat{f}_{k\text{-NN}}$

with respect to the $L^2$ norm as below:

$$
\mathbb{E}_{\{x_i\}_{i=1}^n, \{y_i\}_{i=1}^n}\left[\left|\hat{H}_{k\text{-NN}}\left(\{x_i\}_{i=1}^n, \{y_i\}_{i=1}^n\right) - I_f\right|^2\right]
$$

$$
= \mathbb{E}_{\{x_i\}_{i=1}^n, \{y_i\}_{i=1}^n}\left[\left|\int_\Omega \hat{f}_{k\text{-NN}}(x)dx + \frac{2}{n}\sum_{i=\frac{n}{2}+1}^n \left(y_i - \hat{f}_{k\text{-NN}}(x_i)\right) - \int_\Omega f(x)dx\right|^2\right]
$$

$$
\lesssim \mathbb{E}_{\substack{\{x_i\}_{i=1}^{\frac{n}{2}},\\ \{y_i\}_{i=1}^{\frac{n}{2}}}}\left[\mathbb{E}_{\substack{\{x_i\}_{i=\frac{n}{2}+1}^n,\\ \{y_i\}_{i=\frac{n}{2}+1}^n}}\left[\left|\frac{1}{\frac{n}{2}}\sum_{i=\frac{n}{2}+1}^n \left(f(x_i) - \hat{f}_{k\text{-NN}}(x_i) - \int_\Omega (f(x) - \hat{f}_{k\text{-NN}}(x))dx\right)\right|^2\right]\right]
$$

$$
+ \mathbb{E}_{\{x_i\}_{i=1}^n, \{y_i\}_{i=1}^n}\left[\left|\frac{2}{n}\sum_{i=\frac{n}{2}+1}^n \epsilon_i\right|^2\right]
$$

$$
= \mathbb{E}_{\{x_i\}_{i=1}^{\frac{n}{2}}, \{y_i\}_{i=1}^{\frac{n}{2}}}\left[\frac{4}{n^2}\sum_{i=\frac{n}{2}+1}^n \mathbb{E}_{x_i}\left[\left|\left(f(x_i) - \hat{f}_{k\text{-NN}}(x_i) - \int_\Omega (f(x) - \hat{f}_{k\text{-NN}}(x))dx\right)\right|^2\right]\right]
$$

$$
+ \frac{4}{n^2}\sum_{i=\frac{n}{2}+1}^n \mathbb{E}_{x_i, y_i}\left[\epsilon_i^2\right] \lesssim \frac{1}{n}\left(\mathbb{E}_{\{x_i\}_{i=1}^{\frac{n}{2}}, \{y_i\}_{i=1}^{\frac{n}{2}}}\left[\|\hat{f}_{k\text{-NN}} - f\|_{L^2(\Omega)}^2\right] + n^{-2\gamma}\right)
$$

$$
\lesssim \frac{1}{n}\left(\frac{n^{-2\gamma}}{k} + \left(\frac{k}{n}\right)^{\frac{2s}{d}}\right) + n^{-2\gamma-1}.
$$
(D.35)

Based on the magnitude of the noises, we have the following two cases for the final upper bound:

When $\gamma \in [0, \frac{s}{d})$, the optimal $k$ is determined by balancing the two terms $\frac{n^{-2\gamma}}{k}$ and $\left(\frac{k}{n}\right)^{\frac{2s}{d}}$ in (D.35), which yields $\frac{n^{-2\gamma}}{k} = \left(\frac{k}{n}\right)^{\frac{2s}{d}} \Rightarrow k = \Theta(n^{\frac{2(s-\gamma d)}{d+2s}})$. The corresponding upper bound is given by

$$
\frac{1}{n}\left(\frac{n^{-2\gamma}}{k} + \left(\frac{k}{n}\right)^{\frac{2s}{d}}\right) + n^{-2\gamma-1} \lesssim \frac{1}{n}n^{-2\gamma - \frac{2(s-\gamma d)}{d+2s}} + n^{-1-2\gamma} = n^{-\frac{2s(1+2\gamma)}{2s+d}-1} + n^{-2\gamma-1}
$$

$$
\lesssim \max\{n^{-\frac{2s(1+2\gamma)}{2s+d}-1}, n^{-2\gamma-1}\} = n^{-2\gamma-1}.
$$
(D.36)

When $\gamma \in [\frac{s}{d}, \infty]$, we note that $k \in \{1, 2, \cdots, \frac{n}{2}\}$ must be of at least constant level. Therefore, the optimal $k$ is determined by balancing the two terms $\frac{n^{-2\gamma-1}}{k}$ and $n^{-2\gamma-1}$, which yields that $k = \Theta(1)$ is of constant level. The corresponding upper bound is given by

$$
\frac{1}{n}\left(\frac{n^{-2\gamma}}{k} + \left(\frac{k}{n}\right)^{\frac{2s}{d}}\right) + n^{-2\gamma-1} \lesssim n^{-\frac{2s}{d}-1} + n^{-2\gamma-1}
$$

$$
\lesssim \max\{n^{-\frac{2s}{d}-1}, n^{-2\gamma-1}\} = n^{-\frac{2s}{d}-1}.
$$
(D.37)

Finally, substituting (D.36) and (D.37) into (D.35) gives us the final upper bound:

$$
\mathbb{E}_{\{x_i\}_{i=1}^n, \{y_i\}_{i=1}^n}\left[\left|\hat{H}_{k\text{-NN}}\left(\{x_i\}_{i=1}^n, \{y_i\}_{i=1}^n\right) - I_f\right|\right]
$$

$$
\leq \sqrt{\mathbb{E}_{\{x_i\}_{i=1}^n, \{y_i\}_{i=1}^n}\left[\left|\hat{H}_{k\text{-NN}}\left(\{x_i\}_{i=1}^n, \{y_i\}_{i=1}^n\right) - I_f\right|^2\right]}
$$
(D.38)

$$
\lesssim \sqrt{\frac{1}{n}\left(\frac{n^{-2\gamma}}{k} + \left(\frac{k}{n}\right)^{\frac{2s}{d}}\right) + n^{-2\gamma-1}} \lesssim \sqrt{\max\{n^{-\frac{2s}{d}-1}, n^{-2\gamma-1}\}}
$$

$$
= n^{\max\{-\frac{1}{2}-\gamma, -\frac{1}{2}-\frac{s}{d}\}},
$$

which concludes our proof of Theorem 4.2.

# E    Construction of the Oracle in Assumption 3.1

For the cases when $s > \frac{d}{p}$, the optimal function estimator for any $r \in [1, \infty)$ is already constructed in a series of earlier work [49, 50], whose convergence rate has aleardy been proved to be $(n^{-\frac{s}{d} + (\frac{1}{p} - \frac{1}{r})_+})^r$. Hence, here we only need to focus on the cases when $s \in (\frac{2dq - dp}{2pq}, \frac{d}{p})$. Our goal is construct a desired oracle such that for any $r$ satisfying $\frac{1}{r} \in (\frac{d - sp}{pd}, \frac{1}{p}]$, the convergence rate of function estimation is given by $(n^{-\frac{s}{d} + \frac{1}{p} - \frac{1}{r}})^r$ up to logarithm factors. We need to introduce a few notations and key lemmas in the following two subsections beforehand.

## E.1    Preliminaries and Notations

We introduce a few notations used in our proofs in this subsection. For any compact region $R \subset \mathbb{R}^d$, the diameter $R$ is defined as $\mathrm{diam}(R) := \sup_{x, y \in R} \|x - y\|$. Moreover, for any two compact regions $R_1, R_2 \subset \mathbb{R}^d$, the distance $\mathrm{dist}(R_1, R_2)$ between them is defined as $\mathrm{dist}(R_1, R_2) := \|c_{R_1} - c_{R_2}\|_\infty$, where $\|\cdot\|_\infty$ denotes the $l_\infty$ norm in $\mathbb{R}^d$ and $c_{R_1}, c_{R_2}$ are the centroids of $R_1, R_2$ respectively. For any collection of $n$ data points $P := \{x_1, x_2, \cdots, x_n\}$, we use $\rho(P, \Omega)$ to denote the covering radius of $P$ in $\Omega = [0, 1]^d$, which is defined as follows:

$$\rho(P, \Omega) := \sup_{y \in \Omega} \inf_{1 \le i \le n} \|y - x_i\| \tag{E.1}$$

## E.2    Key Lemmas

In this subsection, we first list corollaries of some well-known theorems as lemmas used in our proofs.

**Lemma 7 (Bramble-Hilbert Lemma)** *For any $s \in \mathbb{N}$ and $u \in W^{s,p}(\Omega)$, there exists some polynomial $\pi$ with $\deg(\pi) \le s - 1$, such that for any $k \in \mathbb{N}$ and $0 \le k \le s$, the following inequality holds:*

$$\|u - \pi\|_{W^{k,p}(\Omega)} \lesssim diam(\Omega)^{s-k} \|u\|_{W^{s,p}(\Omega)} \tag{E.2}$$

**Lemma 8 (Gagliardo–Nirenberg interpolation inequality)** *Fix $s \in \mathbb{N}, p \in [1, \infty)$ and $r \in [1, \infty)$ such that $\frac{1}{r} \in (\frac{d - sp}{pd}, \frac{1}{p}]$. Let $\theta = \frac{d}{s}(\frac{1}{p} - \frac{1}{r}) \in (0, 1)$ such that the relation $\frac{1}{r} = \theta(\frac{1}{p} - \frac{s}{d}) + (1 - \theta)\frac{1}{p}$, holds. Then we have the following inequality:*

$$\|u\|_{L^r(\Omega)} \lesssim \|u\|^\theta_{W^{s,p}(\Omega)} \|u\|^{1-\theta}_{L^p(\Omega)} \tag{E.3}$$

Now let's proceed to list some other results developed in earlier works [66, 67, 70] as lemmas used for constructing the oracle here.

**Lemma 9 (Bound on the covering radius (Theorem 2.1 in [70]))** *Given $P := \{x_1, \cdots, x_n\}$ sampled independently and identically from the uniform distribution on $\Omega = [0, 1]^d$, we have that there exist constants $c_1, c_2 > 0$ and $\alpha_0 > 0$, which are all independent of $n$, such that the following inequality holds for any $\alpha > \alpha_0$:*

$$\mathbb{P}\left(\rho(P, \Omega) \ge c_1 \left(\frac{\alpha \log n}{n}\right)^{\frac{1}{d}}\right) \lesssim n^{1 - c_2 \alpha} \tag{E.4}$$

**Lemma 10 (Properties of the moving least squares estimator (Theorem 4.7 in [67]))** *For    any given collection of $n$ points $P = \{x_1, \cdots, x_n\}$ with covering radius $\rho(P, \Omega)$, there exist constants $a_1, a_2$ independent of $n$ and continuous functions $u_{x_i} : \Omega \to \mathbb{R}$ $(1 \le i \le n)$, such that*

- $\pi(y) = \sum_{i=1}^n \pi(x_i) u_{x_i}(y)$ *for any $y \in \Omega$ and any polynomial $\pi$ with $\deg(\pi) \le s - 1$*

- $\sum_{i=1}^n |u_{x_i}(y)| \le a_1$ *for any $y \in \Omega$*

- $u_x(y) = 0$ *for any $y \in \Omega$ and $x \in P$ with $\|x - y\| \ge a_2 \rho(P, \Omega)$*

Based on the functions $u_{x_i}$ ($1 \le i \le n$) given in Lemma 10, one may define a function estimator $K_n = K_n(\{x_i\}_{i=1}^n, \{f(x_i)\}_{i=1}^n)$ of $f$ as $K_n(\{x_i\}_{i=1}^n, \{f(x_i)\}_{i=1}^n) := \sum_{i=1}^n f(x_i)u_{x_i}$. Such an estimator is obtained via the moving least square approximation, which was first proposed in [66]. One may refer to [67] for more detail about it.

In addition, we also need upper bounds on the moments of any binomial random variable, which is given as the lemma below.

**Lemma 11 (Bound on the moment of binomial random variable ([71, 72]))** *Let $Z \sim Bin(m, p)$ be a binomial random variable binomial distribution with parameters $m$ and $p$. Then for any $k \in \mathbb{N}$, the $k$-th moment $\mathbb{E}[Z^k]$ of $Z$ can be upper bounded as below:*

$$\mathbb{E}[Z^k] \le \left( c' \frac{k}{\log(1 + \frac{k}{mp})} \right)^k \tag{E.5}$$

*where $c' > 1$ above is some universal constant.*

### E.3 Construction of the oracle and proof of its convergence rate

Finally, we will explain how the desired oracle is constructed and present a complete proof of its convergence rate in this subsection. We remark that our proof is similar to the one presented in section 2.1 of [49].

Let $c_1, c_2, \alpha_0, a_1, a_2$ be the positive constants specified in Lemma 9 and Lemma 10 above. We pick $\alpha > 0$ to be sufficiently large such that $\alpha > \max\{\alpha_0, \frac{1}{c_2}(2 + \frac{sr}{d} - \frac{r}{p})\}$, which implies that $1 - c_2\alpha < -\frac{sr}{d} + \frac{r}{p} - 1$. Now we pick $k = \min\left\{ \frac{\sqrt{d}}{c_1}(\frac{n}{\alpha \log n})^{\frac{1}{d}}, (\frac{n}{2\log n})^{\frac{1}{d}} \right\}$ and divide $\Omega = [0,1]^d$ into small cubes $C_1, C_2, \cdots, C_{k^d}$, each of which has side length $k^{-1} \ge 2c_1(\frac{\alpha \log n}{n})^{\frac{1}{d}}$. Furthermore, since the observed data points $\mathcal{X}_n := \{x_i\}_{i=1}^n$ are i.i.d samples from the uniform distribution on $\Omega$, we may define $A$ to be the following event:

$$A := \left\{ \mathcal{X}_n : |\mathcal{X}_n \cap C_l| \ge 1, \, \forall \, 1 \le l \le k^d \right\} \tag{E.6}$$

For any $1 \le l \le k^d$, we use $B_l$ to denote the event $\{\mathcal{X}_n : |\mathcal{X}_n \cap C_l| \ge 1\}$. Then we have $A = \cap_{l=1}^{k^d} B_l$ by definition. Based on $A$ defined above, we may describe our choice of the oracle $K_n = K_n(\{x_i\}_{i=1}^n, \{f(x_i)\}_{i=1}^n)$ as follows: When $A$ is false, we simply pick $K_n(\{x_i\}_{i=1}^n, \{f(x_i)\}_{i=1}^n)$ to be the zero function. When $A$ is true, we pick $K_n(\{x_i\}_{i=1}^n, \{f(x_i)\}_{i=1}^n) := \sum_{i=1}^n f(x_i)u_{x_i}$ to be the moving least square estimator specified in Lemma 10 above.

Before proving that the estimator given by the oracle above satisfies the desired upper bound, let's firstly derive lower bound the probability $\mathbb{P}(A)$ at first. We may apply the assumption $k^d \le \frac{n}{2\log n}$ above and union bound to deduce that

$$\mathbb{P}(A) = \mathbb{P}(\cap_{l=1}^{k^d} B_l) = 1 - \mathbb{P}(\cup_{l=1}^{k^d} B_l^c) \ge 1 - \sum_{l=1}^{k^d} \mathbb{P}(B_l^c) = 1 - k^d(1 - \frac{1}{k^d})^n$$

$$= 1 - k^d \left( (1 - \frac{1}{k^d})^{k^d} \right)^{\frac{n}{k^d}} \ge 1 - k^d e^{-\frac{n}{k^d}} \ge 1 - \frac{n}{2\log n} e^{-2\log n} = 1 - \frac{1}{2n\log n} \ge \frac{1}{2} \tag{E.7}$$

Moreover, we also need to derive an upper bound on the probability $\mathbb{P}(A^c)$ via Lemma 9. We use $E$ to denote the event that $\left\{ \mathcal{X}_n : \rho(\mathcal{X}_n, \Omega) > \frac{\sqrt{d}}{k} \right\}$. Below we will firstly justify that $E$ implies $A^c$, which indicates that $\mathbb{P}(A^c) \le \mathbb{P}(E)$. For the sake of contradiction, assume that $A$ and $E$ both hold true. Then for any point $y \in \Omega$, there must exist some $x_{i_y} \in \mathcal{X}_n$ and some $l_y \in \{1, 2, \cdots, k^d\}$, such that $\{y, x_{i_y}\} \in C_{l_y}$. This implies that $\inf_{1 \le i \le n} \|y - x_i\| \le \|y - x_{i_y}\| \le \text{diam}(C_{l_y}) \le \frac{\sqrt{d}}{k}$. Taking supremum with respect to all $y$ then implies $\rho(\mathcal{X}_n, \Omega) \le \frac{\sqrt{d}}{k}$, which contradicts the definition of $E$. Therefore we must have $\mathbb{P}(A^c) \le \mathbb{P}(E)$. Applying our assumption $k \le \frac{\sqrt{d}}{c_1}(\frac{n}{\alpha \log n})^{\frac{1}{d}}$ and Lemma 9 above further implies that

$$\mathbb{P}(A^c) \leq \mathbb{P}(E) = \mathbb{P}\left(\rho(\mathcal{X}_n, \Omega) > \frac{\sqrt{d}}{k}\right) \leq \mathbb{P}\left(\rho(\mathcal{X}_n, \Omega) \geq c_1\left(\frac{\alpha \log n}{n}\right)^{\frac{1}{d}}\right) \lesssim n^{1-c_2\alpha} \quad \text{(E.8)}$$

Using the law of total expectation and the two upper bounds derived in E.7 and E.8 above, we may obtain the following upper bound on the error of the function estimator $K_n$ as follows:

$$
\begin{aligned}
&\mathbb{E}_{\{x_i\}_{i=1}^n}\left[\|K_n(\{x_i\}_{i=1}^n, \{f(x_i)\}_{i=1}^n) - f\|_{L^r(\Omega)}^r\right] \\
&= \mathbb{E}_{\{x_i\}_{i=1}^n}\left[\left\|\sum_{i=1}^n f(x_i)u_{x_i} - f\right\|_{L^r(\Omega)}^r \Big| A\right]\mathbb{P}(A) + \mathbb{E}_{\{x_i\}_{i=1}^n}\left[\|0 - f\|_{L^r(\Omega)}^r \Big| A^c\right]\mathbb{P}(A^c) \\
&\lesssim \mathbb{E}_{\{x_i\}_{i=1}^n}\left[\left\|\sum_{i=1}^n f(x_i)u_{x_i} - f\right\|_{L^r(\Omega)}^r \Big| A\right]\mathbb{P}(A) + \|f\|_{L^r(\Omega)}^r n^{1-c_2\alpha} \\
&\lesssim \mathbb{E}_{\{x_i\}_{i=1}^n}\left[\left\|\sum_{i=1}^n f(x_i)u_{x_i} - f\right\|_{L^r(\Omega)}^r \Big| A\right] + n^{-\frac{sr}{d}+\frac{r}{p}-1} \\
&\leq \frac{\mathbb{E}_{\{x_i\}_{i=1}^n}\left[\|\sum_{i=1}^n f(x_i)u_{x_i} - f\|_{L^r(\Omega)}^r\right]}{\mathbb{P}(A)} + n^{-\frac{sr}{d}+\frac{r}{p}-1} \\
&\leq 2\mathbb{E}_{\{x_i\}_{i=1}^n}\left[\left\|\sum_{i=1}^n f(x_i)u_{x_i} - f\right\|_{L^r(\Omega)}^r\right] + n^{-\frac{sr}{d}+\frac{r}{p}-1} \\
&\lesssim \mathbb{E}_{\{x_i\}_{i=1}^n}\left[\left\|\sum_{i=1}^n f(x_i)u_{x_i} - f\right\|_{L^r(\Omega)}^r\right] + n^{-\frac{sr}{d}+\frac{r}{p}-1}
\end{aligned}
\quad \text{(E.9)}
$$

where the second inequality above follows from our assumption $\alpha > \max\{\alpha_0, \frac{1}{c_2}(2 + \frac{sr}{d} - \frac{r}{p})\}$ specified earlier. From the last expression above, we can see that now it suffices to show that the first term $\mathbb{E}_{\{x_i\}_{i=1}^n}\left[\|\sum_{i=1}^n f(x_i)u_{x_i} - f\|_{L^r(\Omega)}^r\right]$ in the last expression above is no larger than $n^{-\frac{sr}{d}+\frac{r}{p}-1}$ up to constants and logarithm factors.

Moreover, for any $1 \leq l \leq k^d$, we define $\mathcal{N}_l$ to be a collection of "neighbors" of the cube $C_l$ as below:

$$\mathcal{N}_l := \left\{C_j : \text{dist}(C_j, C_l) \leq \max\{a_2, 1\}\frac{\sqrt{d}}{c_1}\left(\frac{n}{\alpha \log n}\right)^{\frac{1}{d}}\right\} \quad \text{(E.10)}$$

Correspondingly, the union of all the cubes in $\mathcal{N}_l$ is defined as $M_l := \cup_{C_j \in \mathcal{N}_l} C_j$. Since the distance between any two cubes is measured by the $l_\infty$ distance between their centroids, we may deduce that each $M_l$ remains to be a cube for any $1 \leq l \leq k^d$. Also, the number of cubes in each $\mathcal{N}_l$ can be upper bounded by some constant $C_*$ independent of $n$, which implies that the diameter of each $M_l$ satisfies $\text{diam}(M_l) \lesssim k^{-s}$. Then for any $1 \leq l \leq k^d$, we may apply Lemma 7 above to deduce that there exists some polynomial $\pi_l$ such that $\deg(\pi_l) \leq s - 1$ and the following two inequalities hold:

$$\|f - \pi_l\|_{W^{s,p}(M_l)} \lesssim \|f\|_{W^{s,p}(M_l)}$$
$$\|f - \pi_l\|_{L^p(M_l)} = \|f - \pi_l\|_{W^{0,p}(M_l)} \lesssim \text{diam}(M_l)^s\|f\|_{W^{s,p}(M_l)} \lesssim k^{-s}\|f\|_{W^{s,p}(M_l)}$$
$$\quad \text{(E.11)}$$

Combining the two inequalities above with Gagliardo–Nirenberg interpolation inequality E.3 listed in Lemma 8 further implies that for any $1 \leq l \leq k^d$,

$$\|f - \pi_l\|_{L^r(M_l)}^r \lesssim \left(\|f - \pi_l\|_{W^{s,p}(M_l)}^\theta \|f - \pi_l\|_{L^p(M_l)}^{1-\theta}\right)^r \lesssim k^{-sr(1-\theta)}\|f\|_{W^{s,p}(M_l)}^r \quad \text{(E.12)}$$

Using the first and third property of the moving least square estimator listed in Lemma 10 above, we may deduce that the following inequality holds for any $1 \leq l \leq k^d$ and any $y \in C_l$:

$$\left| f(y) - \sum_{i=1}^{n} f(x_i) u_{x_i}(y) \right|^r = \left| f(y) - \pi_l(y) - \left( \sum_{i=1}^{n} f(x_i) u_{x_i}(y) - \sum_{i=1}^{n} \pi_l(x_i) u_{x_i}(y) \right) \right|^r$$

$$= \left| f(y) - \pi_l(y) - \sum_{i:x_i \in M_l} (f(x_i) - \pi_l(x_i)) u_{x_i}(y) \right|^r$$

$$\leq \left( \left| (f - \pi_l)(y) \right| + \sum_{i:x_i \in M_l} \left| (f - \pi_l)(x_i) u_{x_i}(y) \right| \right)^r$$

$$\leq \left( 1 + |\mathcal{X}_n \cap M_l| \right)^{r-1} \left( \left| (f - \pi_l)(y) \right|^r + \sum_{i:x_i \in M_l} \left| (f - \pi_l)(x_i) u_{x_i}(y) \right|^r \right)$$

$$\lesssim \max \left\{ 1, |\mathcal{X}_n \cap M_l|^{r-1} \right\} \cdot \max\{1, a_1^r\} \cdot \left( \left| (f - \pi_l)(y) \right|^r + \sum_{i:x_i \in M_l} \left| (f - \pi_l)(x_i) \right|^r \right)$$

$$\lesssim \max \left\{ 1, |\mathcal{X}_n \cap M_l|^{r-1} \right\} \left( \left| (f - \pi_l)(y) \right|^r + \sum_{i:x_i \in M_l} \left| (f - \pi_l)(x_i) \right|^r \right)$$

$$\tag{E.13}$$

By rewriting $\| \sum_{i=1}^{n} f(x_i) u_{x_i} - f \|_{L^r(\Omega)}^r$ as an integral and plugging in the inequality derived above, we then obtain that

$$\mathbb{E}_{\{x_i\}_{i=1}^n} \left[ \left\| \sum_{i=1}^{n} f(x_i) u_{x_i} - f \right\|_{L^r(\Omega)}^r \right] = \mathbb{E}_{\{x_i\}_{i=1}^n} \left[ \sum_{l=1}^{k^d} \int_{C_l} \left| \sum_{i=1}^{n} f(x_i) u_{x_i}(y) - f(y) \right|^r dy \right]$$

$$\lesssim \mathbb{E}_{\{x_i\}_{i=1}^n} \left[ \sum_{l=1}^{k^d} \int_{C_l} \max \left\{ 1, |\mathcal{X}_n \cap M_l|^{r-1} \right\} \left( \left| (f - \pi_l)(y) \right|^r + \sum_{i:x_i \in M_l} \left| (f - \pi_l)(x_i) \right|^r \right) dy \right]$$

$$\lesssim \sum_{l=1}^{k^d} \left( \int_{C_l} \left| (f - \pi_l)(y) \right|^r dy \right) \mathbb{E}_{\{x_i\}_{i=1}^n} \left[ \max \left\{ 1, |\mathcal{X}_n \cap M_l|^{r-1} \right\} \right]$$

$$+ \sum_{l=1}^{k^d} \mathbb{E}_{\{x_i\}_{i=1}^n} \left[ \max \left\{ 1, |\mathcal{X}_n \cap M_l|^{r-1} \right\} \left( \sum_{i:x_i \in M_l} \left| (f - \pi_l)(x_i) \right|^r \right) k^{-d} \right]$$

$$\leq \sum_{l=1}^{k^d} \| f - \pi_l \|_{L^r(M_l)}^r \mathbb{E}_{\{x_i\}_{i=1}^n} \left[ 1 + |\mathcal{X}_n \cap M_l|^{r-1} \right]$$

$$+ k^{-d} \sum_{l=1}^{k^d} \mathbb{E}_{\{x_i\}_{i=1}^n} \left[ \max \left\{ 1, |\mathcal{X}_n \cap M_l|^{r-1} \right\} \left( \sum_{i:x_i \in M_l} \left| (f - \pi_l)(x_i) \right|^r \right) \right]$$

$$\tag{E.14}$$

Before deriving upper bounds on the final expression in E.14, let's consider bounding the two expectations $\mathbb{E}_{\{x_i\}_{i=1}^n} \left[ |\mathcal{X}_n \cap M_l|^{r-1} \right]$ and $\mathbb{E}_{\{x_i\}_{i=1}^n} \left[ |\mathcal{X}_n \cap M_l|^r \right]$ $(1 \leq l \leq k^d)$ at first. Given that the datapoints $\mathcal{X}_n = \{x_i\}_{i=1}^n$ are i.i.d samples from the uniform distribution on $\Omega$, we have that each $|\mathcal{X}_n \cap M_l| \sim \text{Bin}(k^d, p_l)$ is a binomial random variable, where $|\mathcal{N}_l| \leq c_* \Rightarrow p_l \leq \frac{c_*}{k^d}$ for any $1 \leq l \leq k^d$. Applying Lemma 11 above then yields that for any $1 \leq l \leq k^d$:

$$\mathbb{E}_{\{x_i\}_{i=1}^n} \left[ |\mathcal{X}_n \cap M_l|^{r-1} \right] \leq \left( c' \frac{r-1}{\log(1 + \frac{r-1}{k^d p_l})} \right)^{r-1} \leq \left( c' \frac{r-1}{\log(1 + \frac{r-1}{c_*})} \right)^{r-1},$$

$$\mathbb{E}_{\{x_i\}_{i=1}^n} \left[ |\mathcal{X}_n \cap M_l|^r \right] \leq \left( c' \frac{r}{\log(1 + \frac{r}{k^d p_l})} \right)^r \leq \left( c' \frac{r}{\log(1 + \frac{r}{c_*})} \right)^r$$

$$\tag{E.15}$$

where the two upper bounds in E.15 are all constants independent of the sample size $n$.

Secondly, we need to derive an upper bound on the summation $\sum_{l=1}^{k^d} \|f - \pi_l\|_{L^r(M_l)}^r$. Recall our definition of the cubes $M_l$ $(1 \leq l \leq k^d)$, we can deduce that each small cube $C_i$ is contained within at most constantly many big cubes $M_l$. That is to say, there exists some constant $c_*'$ independent of sample size $n$, such that $|l : C_i \subset M_l| \leq c_*'$ for any $1 \leq i \leq k^d$. By combining this fact with inequality E.12 proved above, we can deduce that

$$
\sum_{l=1}^{k^d} \|f - \pi_l\|_{L^r(M_l)}^r \lesssim \sum_{l=1}^{k^d} k^{-sr(1-\theta)} \|f\|_{W^{s,p}(M_l)}^r = k^{-sr(1-\theta)} \sum_{l=1}^{k^d} \sum_{i:C_i \in M_l} \|f\|_{W^{s,p}(C_i)}^r
$$

$$
= k^{-sr(1-\theta)} \sum_{i=1}^{k^d} \sum_{l:C_i \in M_l} \|f\|_{W^{s,p}(C_i)}^r \leq k^{-sr(1-\theta)} \sum_{i=1}^{k^d} c_*' \|f\|_{W^{s,p}(C_i)}^r \tag{E.16}
$$

$$
\lesssim k^{-sr(1-\theta)} \sum_{i=1}^{k^d} \|f\|_{W^{s,p}(C_i)}^r \leq k^{-sr(1-\theta)} \Big( \sum_{i=1}^{k^d} \|f\|_{W^{s,p}(C_i)}^p \Big)^{\frac{r}{p}} = k^{-sr(1-\theta)} \|f\|_{W^{s,p}(\Omega)}^r
$$

Furthermore, to derive the final upper bound, let's simplify the second term in E.14 and try to upper bound it. Using the law of total expectation again, we have that

$$
k^{-d} \sum_{l=1}^{k^d} \mathbb{E}_{\{x_i\}_{i=1}^n} \left[ \max\left\{1, |\mathcal{X}_n \cap M_l|^{r-1}\right\} \Big( \sum_{i:x_i \in M_l} \big|(f - \pi_l)(x_i)\big|^r \Big) \right]
$$

$$
= k^{-d} \sum_{l=1}^{k^d} \sum_{l_1,\cdots,l_n=1}^{k^d} \mathbb{E}_{\{x_i\}_{i=1}^n} \left[ \max\left\{1, |\mathcal{X}_n \cap M_l|^{r-1}\right\} \cdot \right.
$$

$$
\Big( \sum_{i:x_i \in M_l} \big|(f - \pi_l)(x_i)\big|^r \Big) \Big| x_1 \in C_{l_1}, \cdots, x_n \in C_{l_n} \right] \cdot \mathbb{P}(x_1 \in C_{l_1}, \cdots, x_n \in C_{l_n})
$$

$$
= k^{-d} \sum_{l=1}^{k^d} \sum_{l_1,\cdots,l_n=1}^{k^d} \max\left\{1, |\mathcal{X}_n \cap M_l|^{r-1}\right\} \Big( \sum_{i:x_i \in M_l} \int_{C_{l_i}} \big|(f - \pi_l)(x_i)\big|^r dx_i \Big)
$$

$$
\cdot \mathbb{P}(x_1 \in C_{l_1}, \cdots, x_n \in C_{l_n})
$$

$$
= k^{-d} \sum_{l=1}^{k^d} \sum_{l_1,\cdots,l_n=1}^{k^d} \max\left\{1, |\mathcal{X}_n \cap M_l|^{r-1}\right\} \Big( \sum_{i:x_i \in M_l} \|f - \pi_l\|_{L^r(C_{l_i})}^r \Big)
$$

$$
\cdot \mathbb{P}(x_1 \in C_{l_1}, \cdots, x_n \in C_{l_n})
$$

$$
\leq k^{-d} \sum_{l=1}^{k^d} \sum_{l_1,\cdots,l_n=1}^{k^d} \max\left\{1, |\mathcal{X}_n \cap M_l|^{r-1}\right\} \Big( |\mathcal{X}_n \cap M_l| \sum_{C_j \in N_l} \|f - \pi_l\|_{L^r(C_j)}^r \Big)
$$

$$
\cdot \mathbb{P}(x_1 \in C_{l_1}, \cdots, x_n \in C_{l_n})
$$

$$
\leq k^{-d} \sum_{l=1}^{k^d} \Big( \sum_{l_1,\cdots,l_n=1}^{k^d} \max\left\{1, |\mathcal{X}_n \cap M_l|^r\right\} \mathbb{P}(x_1 \in C_{l_1}, \cdots, x_n \in C_{l_n}) \Big) \|f - \pi_l\|_{L^r(M_l)}^r
$$

$$
= k^{-d} \sum_{l=1}^{k^d} \mathbb{E}_{\{x_i\}_{i=1}^n} \left[ \max\left\{1, |\mathcal{X}_n \cap M_l|^r\right\} \right] \cdot \|f - \pi_l\|_{L^r(M_l)}^r
$$

$$
\leq k^{-d} \sum_{l=1}^{k^d} \mathbb{E}_{\{x_i\}_{i=1}^n} \left[ 1 + |\mathcal{X}_n \cap M_l|^r \right] \cdot \|f - \pi_l\|_{L^r(M_l)}^r
$$

$$
\tag{E.17}
$$

Finally, by substituting the bounds derived in E.15, E.16, E.17 into E.14, we may further plug in our choice of $k$ and ignore the logarithm terms to obtain the desired upper bound:

$$\mathbb{E}_{\{x_i\}_{i=1}^n}\left[\left\|\sum_{i=1}^n f(x_i)u_{x_i} - f\right\|_{L^r(\Omega)}^r\right] \lesssim (1 + k^{-d})\sum_{l=1}^{k^d} \|f - \pi_l\|_{L^r(M_l)}^r$$

$$\lesssim \sum_{l=1}^{k^d} \|f - \pi_l\|_{L^r(M_l)}^r \lesssim k^{-sr(1-\theta)}\|f\|_{W^{s,p}(\Omega)}^r \qquad \text{(E.18)}$$

$$\lesssim \left((\frac{n}{\log n})^{\frac{1}{d}}\right)^{-sr(1-\frac{d}{s}(\frac{1}{p}-\frac{1}{r}))} \lesssim n^{-\frac{sr}{d}+\frac{r}{p}-1}$$

where we ignore the logarithm term in the last step above. This concludes our proof of the convergence rate of the oracle we specified above. Substituting E.18 into E.9 completes our proof of Assumption 3.1.