# OpenReview forum: "When can Regression-Adjusted Control Variate Help? Rare Events, Sobolev Embedding and Minimax Optimality"
_NeurIPS.cc/2023/Conference — NeurIPS 2023 poster_

### Official Review · Reviewer_Z34k · 2023-06-09

**Soundness:** 3 good
**Presentation:** 2 fair
**Contribution:** 3 good
**Rating:** 5
**Confidence:** 3

**Summary:**

This paper presents theoretical results on estimates of the integral of f^q based, and when regression adjusted control variates can lead to better Monte Carlo estimators.

**Strengths:**

The results look to be novel and improve existing ones in this area. Furthermore, this is an area that is important in practice and for which strong theory can be impactful.

I am not familiar with the historical work on this problem, and thus find it hard to say just how impressive the theoretical developments are -- thus the main focus of my review has been on their practical usefulness. It seemed that there is potential for these to be important, but (see Weaknesses) the presentation currently limits this.

**Weaknesses:**

The paper's presentation limits its accessibility to those who are very familiar with the theoretical area, which means that it is unlikely that the work would have impact on practice. As just one example, the introduction is written under the assumption that the reader knows what a Sobolev space is.

Similarly, there is little attempt to relate the theory to practice. E.g. how would a user know what Sobolev space their function is in? This may be straightforward in some situations (but is not explained). In general settings where we want to estimate E_pi(f(X)) and area using MCMC to simulate from pi -- things are complicated if the domain of X is unbounded, as I believe the theoretical results assume require to transform X to e.g. lie on the unit hyper-cube. The expectation is still an integral of a function over the hyper-cube, but perhaps it is less clear what the function is/what its properties are.



**Questions:**

The main thing that would improve my opinion of the work would be if the authors could add some details linking the theory to practice.

These were questions/comments I had when reading the paper that relate to the current presentation, and are suggestions that may make the paper more accessible to a wider audience:

The paper would be more accessible if an informal definition and intuition for a Sobolev space is given.

I like the idea of something like Figure 1 to try and get some intuition about the dichotomy between situations where rare/extreme events are and are not possible. But at the moment I think this is of limited use — it just shows a curve with and without one. Could more be said, I.e. how does the Sobolev spaces in (a) and (b) differ (i.e. a has larger s perhaps?) and what does this mean about the functions (e.g. a is smoother than b). It may not be clear why smoothness as defined by the Sobolev space immediately means that you do not get extreme events as in the figure, as compared to assumptions on the derivatives of the function. Also the definition of extreme event seems to depend on q. Again some intuition here would help.

Presumably there is some link in that if f is in a Sobolev space, then f^q is in a different space — and really what matters are the properties of f^q?

What if Omega is unbounded (as you use a uniform distribution for the points in Theorem 2.1)? Or equivalently how does the distribution on the points affect the results in (2.1). At one point in the introduction you mention x_i uniform on the unit cube, but initially you just say that Omega is in R^d. I think things could be clearer.

Figure 2 mentions truncated Monte Carlo — has this been defined at this stage (even briefly/informally)?

Section 4: I think the introduction here could be clearer. That is stating that you now consider situations where you observe f(x_i) with error, and evaluate how the noise level impacts performance.

---

> ### Author Rebuttal · Authors · 2023-08-09
>
> We would like to thank the reviewer for the feedback on our work. Below are our responses to the questions raised in the review:
>
> 1. Connection between application and our theoretical work
>
> We already discussed related application in subsection 1.1 "Regression-Adjusted Control Variate (RACV)" of the paper. Here we will list the most important applications of RACV along with their references. These include gradient estimation [1], statistical inference in biology [2], causal inference [3], estimation of the normalization factor and sampling [4], MCMC simulation [5].
>
> Moreover, we would like to emphasize that our main contribution is to provide a theoretical understanding of RACV. We examine all possible cases and pinpoint the regimes when RACV can help us obtain a minimax optimal estimator. We will make sure to clarify our main contributions in our next version.
>
> We admit that our theoretical results are derived under a few assumptions (The given function is in a sobolev space; The data points are uniformly sampled from a unit cube; etc). We will discuss how these assumptions limit the application of our theory while revising the conclusion section.
>
> 2. Intuitive Explanation of mathematical concepts (Sobolev spaces and embedding)
>
> We have given the definition of Sobolev space in section 1.3. Intuitively, one may interpret $W^{s,p}(\Omega)$ as the space formed by all the functions $f$ satisfies the following two conditions: (1) $f$ can be differentiated up to $s$ times; (2) The $i$-th derivative of $f$ has finite $L^p$ norm for any $0 \leq i \leq s$.
>
> Moreover, we also list a simplified version of the Sobolev Embedding Theorem below:
>
> For any non-negative integer $s$ and $1 \leq p < r \leq \infty$ satisfying $\frac{1}{p}-\frac{s}{d} = \frac{1}{r}$, we have $W^{s,p}(\Omega) \subset L^r(\Omega)$, $\textit{i.e,}$ every sobolev function in $W^{s,p}$ is also a $L^r$ function.
>
> Now let's consider dividing the unit cube $\Omega$ into $n$ grids, each of which has side length $n^{-1/d}$. We will then use a bump function supported on one grid  $\Omega'$ as an example to provide some intuition on the connection between existence of rare events and the smoothness parameter $s$. For simplicity, here we just take $s=\frac{d}{p}$ to be the threshold. Our bump function is given by
>
> $$
> g(x) = n^{(-\frac{s}{d}+\frac{1}{p})}K(n^{\frac{1}{d}}(x-c))
> $$
>
> where $c$ denotes center of the grid and $K(y) :=  \prod_{i=1}^{d}\exp(-(1-y_{i}^2)^{-1})$ is compactly supported on $[-1,1]^d$. We will then proceed to verify that  $g \in W^{s,p}(\Omega)$. In fact, for any $|t| \leq s$, we may use change of variable to directly compute the $L^p$ norm $\|D^{t}g\|_{L^p}$ of the $t$-th derivative $D^{t}g$, which is given by:
>
> $$n^{\frac{|t|-s}{d}} \|D^{t}K\|_{L^p}$$
>
> Now let's consider different values of $s$. On the one hand, when $s >\frac{d}{p}$, we have that $f \in L^{\infty}(\Omega)$ for any Sobolev function $f \in W^{s,p}(\Omega)$ via the Sobolev Embedding Theorem above. This corresponds to Part (a) of Figure 1 in our paper, where the function is uniformly bounded without any extreme event. On the other hand, when $s <\frac{d}{p}$, from the bump function above we have that the power $-\frac{s}{d}+\frac{1}{p}$ in the leading coefficient $n^{-\frac{s}{d}+\frac{1}{p}}$ is positive. As the bump function $g$ constructed above lies in the Sobolev space $W^{s,p}(\Omega)$ for any $n$, we may take the limit $n \rightarrow \infty$ to deduce that a peak (rare event) can exist in this bump function, which corresponds to Part (b) of Figure 1 in our paper. Essentially speaking, Part (a) corresponds to $W^{s,p}$ with larger $s$, while Part (b) corresponds to $W^{s,p}$ with smaller $s$.
>
> Furthermore, we will explain how Sobolev Embedding Theorem above can help us find the space that $f^q$ lies in. Let $r = (\frac{1}{p}-\frac{s}{d})^{-1}$. Then we have $f \in L^{r}(\Omega)$, which implies that $f^q$ is a $L^{\frac{r}{q}}$ function. When proving that our estimator is minimax optimal, our main strategy is to embed the influence function $f^{q-1}$ and the error function $f-\hat{f}$ into "dual" $L^p$ spaces. Hence, it's crucial to figure out how to use some $\hat{f}$ to approximate the original $f$, which is also the main technique contribution we made in the paper.
>
> 3. Miscellaneous questions
>
> The phrase "Truncated Monte Carlo" is defined in Subsection 3.2 of our paper. Regarding the introduction of Section 4, we will follow your advice to revise it in the new version.
>
> We will take all suggestions into account while revising our manuscript. Finally, we would like to express our gratitude to the reviewer for your time and dedication, which helps us improve the quality of our manuscript.
>
> [1] Shi, J., Zhou, Y., Hwang, J., Titsias, M. and Mackey, L., 2022. Gradient estimation with discrete Stein operators. Advances in neural information processing systems, 35, pp.25829-25841.
>
> [2] Angelopoulos, A.N., Bates, S., Fannjiang, C., Jordan, M.I. and Zrnic, T., 2023. Prediction-powered inference. arXiv preprint arXiv:2301.09633.
>
> [3] Liu, H. and Yang, Y., 2020. Regression-adjusted average treatment effect estimates in stratified randomized experiments. Biometrika, 107(4), pp.935-948.
>
> [4] Holzmüller, D. and Bach, F., 2023. Convergence rates for non-log-concave sampling and log-partition estimation. arXiv preprint arXiv:2303.03237.
>
> [5] Belomestny, D., Goldman, A., Naumov, A. and Samsonov, S., 2023. Theoretical guarantees for neural control variates in MCMC. arXiv preprint arXiv:2304.01111.

---

> > ### Comment · Reviewer_Z34k · 2023-08-16
> >
> > Thanks for the response. However, I do not think it addressed my main concern -- about making the paper accessible to a wider audience, giving clearer intuition and linking the results in with practice a bit more clearly.
> >
> > Looking at the other reviews -- I think this was also the main concern of the most negative review. By comparison, those who are most familiar with the underlying mathematics are more positive about the paper. I accept that it has some new theoretical results, to me it is just a shame that the impact of the work will be limited by a lack of interest in making these accessible to practitioners, as this would increase the likely impact of the work.
> >
> > Given the theoretical aspects of the work are good, I have increased my score to 5.

---

> > > ### Author Response · Authors · 2023-08-19
> > > **Thank you**
> > >
> > > Thank you for your response. We are grateful for your detailed review, which helped us improve the presentation of our paper a lot. We believe the new version of our manuscript has already become more approachable for the general audience. We also think that it will capture broader attention if it gets accepted. After receiving your review, we’ve already included all the explanations about the link between our work and applications in the rebuttal and added a broader impact section to the main text. Additionally, we also added a background section to the appendix, which provides preliminaries for researchers without sufficient mathematical background. Just as we pointed out earlier, there were lots of related papers (already cited in the manuscript) studying regression-adjusted control variates (RACV) from an application perspective. Hence, we believe that the story presented in our paper will catch the interest of the general NeurIPS audience.
> > >
> > > Here's what we've already changed to our manuscript:
> > >
> > > We've extended the description from lines 79-91 to detailly discuss the relationship of our work with empirical work [1-7] and changed the subtitle from "Regression-adjusted Control Variate" to "relationship to empirical works". We believe such subtitles will let empirical users easier to find who to find insight for their application.
> > >
> > > We'll add the following description to section 1.2 contribution and section 5 conclusion
> > >
> > > All previous works made strong assumptions that $s<d/p$, which made the function uniformly bounded and neglected the possibility of spike functions. As a result, they were unable to discover the transition between the two regimes that our paper found. Our paper also introduces a new technique using the Sobolev embedding theorem to embed the influence function into the dual norm of the function estimation evaluation norm. This technique could have a significant impact on the semi-parametric literature.
> > >
> > > We've added a section in the appendix to introduce the Sobolev space and Sobolev embedding theorem in the appendix. This is an extension of our previous rebuttal.
> > >
> > >
> > > [1] Shi, J., Zhou, Y., Hwang, J., Titsias, M. and Mackey, L., 2022. Gradient estimation with discrete Stein operators. Advances in neural information processing systems, 35, pp.25829-25841.
> > >
> > >  [2] Angelopoulos, A.N., Bates, S., Fannjiang, C., Jordan, M.I. and Zrnic, T., 2023. Prediction-powered inference. arXiv preprint arXiv:2301.09633.
> > >
> > >  [3] Liu, H. and Yang, Y., 2020. Regression-adjusted average treatment effect estimates in stratified randomized experiments. Biometrika, 107(4), pp.935-948.
> > >
> > > [4] Holzmüller, D. and Bach, F., 2023. Convergence rates for non-log-concave sampling and log-partition estimation. arXiv preprint arXiv:2303.03237.
> > >
> > >  [5] Sun Z, Barp A, Briol F X. Vector-valued control variates International Conference on Machine Learning. PMLR, 2023: 32819-32846.
> > >
> > > [6] Yaniv Romano, Evan Patterson, and Emmanuel Candes. Conformalized quantile regression.  Advances in neural information processing systems, 32, 2019.
> > >
> > > [7] Aleksandros Sobczyk and Mathieu Luisier. Approximate euclidean lengths and distances  beyond johnson-lindenstrauss. Neurips 2022.

---

### Official Review · Reviewer_gXt5 · 2023-06-19

**Soundness:** 2 fair
**Presentation:** 2 fair
**Contribution:** 2 fair
**Rating:** 3
**Confidence:** 3

**Summary:**

This paper studies whether we can learn a control variate to reduce variance in Monte Carlo sampling.

**Strengths:**

This paper studies whether we can learn a control variate to reduce variance in Monte Carlo sampling.

**Weaknesses:**

To someone who's not familiar with this area of research, the paper lacks introduction and is very confusing.

**Questions:**

Is there a simple and intuitive toy example to better understand problem and proposed solutions?

**Limitations:**

To someone who's not familiar with this area of research, the paper lacks introduction and is very confusing.

---

### Official Review · Reviewer_Qqae · 2023-07-06

**Soundness:** 3 good
**Presentation:** 2 fair
**Contribution:** 3 good
**Rating:** 7
**Confidence:** 3

**Summary:**

The authors study the theoretical properties of Monte Carlo estimators of the moments of a Sobolev function with nonparametric regression-adjusted control variates. In particular, they show that when a certain smoothness assumption is satisfied, then the regression-adjusted rule achieves the minimax optimal rate (where the minimum is over estimators, the maximum over test functions/integrands). When the smoothness assumption considered is not satisfied, the Monte Carlo estimate of the moments has infinite variance, and the authors show that to again achieve the minimax optimal rate one needs to resort to a truncated estimator. Further, the authors study the performance of the regression-adjusted estimator for integral estimation with *noisy* observations, a somewhat nonstandard setting, and provide some lower and upper bounds for the estimator's absolute error.

**Strengths:**

- A good theoretical contribution to the literature on minimax-like results for Monte Carlo estimators
- Good effort in making the notation and results accessible to a wider audience
- Interesting setting with noisy function evaluations

**Weaknesses:**

- The paper reads a bit like a laundry list where results are presented in a sequence without much of a "story" connecting them; in particular, sections 3 and 4 feel disconnected, the former being about minimax results with noiseless observations specifically about moment estimation, the latter being about integral estimation (q=1 ? ) with noisy observations. Why does the setting change, what is the motivation?
- Regarding your contribution, you mention "existing proof techniques are only applicable in scenarios where there is either no noise or a constant level of noise" - here are you talking about results for moment estimation or integral estimation ? Can you clarify in both settings which aspects are missing from the existing literature and how your results filll that gap ?
- While I understand that the contribution is of theoretical nature, could you make more effort motivating why the setting of noisy observations is interesting / relevant to applications of Monte Carlo ? Further on this point, could you give intuitions for when the main smoothness assumption is expected to hold or not hold (beyond rare events) ?

**Questions:**

- What are other valid choices beyond KNN for Section 4?
- It would improve the paper to give example classes of $f$'s of interest

**Limitations:**

The authors should add a couple of sentences about limitations in the conclusions.

---

> ### Author Rebuttal · Authors · 2023-08-09
>
> We are really grateful for the reviewer's valuable feedback on our work. For questions raised in the review, we list our answers below:
>
> 1. A coherent story connecting all the results
>
> We already built a “story” connecting the results presented in the paper as the reviewer suggested. The story is to investigate whether a machine learning based control variate can help improve the Monte Carlo methods in terms of convergence rate.  We find out that if the random variable simulated via the Monte Carlo method is of finite variance, then we can always use a non-parametric control variate to improve the convergence rate. However, if rare events of infinite variance exist, a non-parametric control variate no longer helps to improve the convergence rate. This "story" is repeated several times in the title, abstract, introduction, and contribution sections.
>
> 2. Our contributions in the paper
>
> We've already listed all our contributions in section 1.2 (which is the "story" introduced before) in an intuitive way. We discuss our contributions in proof techniques here.
>
> Firstly, we emphasize that our paper is the first to consider the case when the underlying function $f$ is in the Sobolev space $W^{s,p}(\Omega)$, where $s < \frac{d}{p}$. In this regime, $f \in W^{s,p}(\Omega)$ may not be embedded in $L^{\infty}(\Omega)$. Different from literatures of doubly robust estimators, here we are considering a nonlinear functional, so the influence function is not exactly zero in our estimator. Therefore, we need to use Sobolev Embedding Theorem to embed the influence function $f^{q-1}$ and the error function $f-\hat{f}$ into "dual" $L^p$ spaces to obtain the desired upper bound. For the sake of completeness, we list a simplified version of the Sobolev Embedding Theorem below:
>
> For any non-negative integer $s$ and $1 \leq p < r \leq \infty$ satisfying $\frac{1}{p}-\frac{s}{d} = \frac{1}{r}$, we have $W^{s,p}(\Omega) \subset L^r(\Omega)$, $\textit{i.e,}$ every sobolev function in $W^{s,p}$ is also a $L^r$ function.
>
> Secondly, the technique used in our proof of the information-theoretic lower bounds is a bit different from that of previous work [1]. In one case of our proof, we pick the two priors to be two discrete distributions of functions such that any function’s sign on each grid in the divided domain is a discrete random variable whose distribution depends on $n$ (the number of data points). This enables us to calculate the amount of information required to distinguish between the two hypotheses even when there is no observational noise. We think that the technique we used here might be useful for the nonparametric statistics community.
>
> In addition, for the problem of estimating integral based on noisy observations, our contribution is to characterize the minimax optimal convergence rate for any noise level ranging from zero to $O(1)$.
>
> 3. Applications of our theoretical analysis
>
> We already discussed related application in subsection 1.1 related work "Regression-Adjusted Control Variate (RACV)" of the paper. Here we will list the most important applications of RACV along with their references. These include gradient estimation [2], statistical inference in biology [3], causal inference [4], estimation of the normalization factor and sampling [5], MCMC simulation [6].
>
> We admit that our theoretical results are derived under a few assumptions (The given function is in a sobolev space; The data points are uniformly sampled from a unit cube; etc). We will discuss how these assumptions limit the application of our theory while revising the conclusion section.
>
> Finally, we would like to thank the reviewer again for all the helpful comments, which help us improve the quality of our manuscript.
>
> [1] Tsybakov, A.B., 2004. Introduction to nonparametric estimation, 2009. URL https://doi.org/10.1007/b13794.
>
> [2] Shi, J., Zhou, Y., Hwang, J., Titsias, M. and Mackey, L., 2022. Gradient estimation with discrete Stein operators. Advances in neural information processing systems, 35, pp.25829-25841.
>
> [3]  Angelopoulos, A.N., Bates, S., Fannjiang, C., Jordan, M.I. and Zrnic, T., 2023. Prediction-powered inference. arXiv preprint arXiv:2301.09633.
>
> [4] Liu, H. and Yang, Y., 2020. Regression-adjusted average treatment effect estimates in stratified randomized experiments. Biometrika, 107(4), pp.935-948.
>
> [5] Holzmüller, D. and Bach, F., 2023. Convergence rates for non-log-concave sampling and log-partition estimation. arXiv preprint arXiv:2303.03237.
>
> [6] Belomestny, D., Goldman, A., Naumov, A. and Samsonov, S., 2023. Theoretical guarantees for neural control variates in MCMC. arXiv preprint arXiv:2304.01111.

---

> > ### Comment · Reviewer_Qqae · 2023-08-21
> > **Reply to authors' rebuttal**
> >
> > I am satisfied with the response to my questions and therefore increase my score.
> > I still highly recommend to the authors to not simply cite relevant potential applications (such as gradient estimation), but to actually instantiate a realistic problems with equations in an example paragraph (without necessarily an experiment needed, actually).

---

> > > ### Author Response · Authors · 2023-08-21
> > >
> > > Thank you for your response and feedback on our work. We appreciate it!

---

### Official Review · Reviewer_pkMo · 2023-07-06

**Soundness:** 3 good
**Presentation:** 3 good
**Contribution:** 3 good
**Rating:** 6
**Confidence:** 4

**Summary:**

In this papers the authors consider estimating the $q^{\rm th}$-moment of a function $f$ (i.e. $\int f^q(x)dx$) by observing samples of $x_i,f(x_i)$ when $x_i$-essentially follows a uniform distribution over (a compact) domain of integration. The paper first develops an information theoretic lower bound (using standard testing arguments) for estimating $\int f^q(x)dx$ for $f\in W^{s,p}$ (Sobolev Space of smoothness $s\in \mathbb{N}$ and order $p\geq 1$). In order to match the lower bounds the authors subsequently specialize the results in two domains -- high and low smoothness respectively. In the high smoothness regime, a suitably bias corrected estimator based on an initial ML based estimator of $f$ (that satisfies some desirable initial properties) is shown to be optimal whereas in low smoothness regimes, owing to existence of rare and extreme events due to unboundedness of the $2q^{\rm th}$-moment of $f$, the problem needs to addressed differently and the authors show that a truncated version of the classical Monte Carlo method can provide optimality guarantees. Finally the paper also provides some details on the noisy version of the problem (i.e. when one observes $x_i,f(x_i)+\varepsilon_i$, $\varepsilon_i\sim N(0,\sigma_n^2)$) to obtain connections to the nonparametric regression literature.

**Strengths:**

(i) A complete paper with minimax upper and lower bounds for estimating $q^{\rm th}$-moment of a function $f$ (i.e. $\int f^q(x)dx$) by observing samples of $x_i,f(x_i)$.

(ii) Considers both high and low smoothness regimes of the problem.

**Weaknesses:**

Not much discussion on a a very well developed non-parametric theory of estimating $\int f^q(x)dx$) by observing samples of $x_i,f(x_i)+\epsilon_i,\epsilon_i\sim N(0,\sigma^2)$ where both $1/\sqrt{n}$ and (slower than)non-$1/\sqrt{n}$-rates along with efficiency bounds  $1/\sqrt{n}$ are presented.

**Questions:**

If one is allowed to choose the design points carefully from a suitable class of distributions, can the authors discuss how the minimax rates change based on the class of distributions of $x$ -- indeed for uniform distribution one understands the problem and the problem does not change much for compactly supported (known/sufficiently regular) densities of $x$ which are bounded away from $0$ on its compact support. However, when one goes beyond this class, some properties might/might not change (e.g. for entropy estimation in density models it makes a fundamental density when the density is bounded away from $0$ versus when its not) since not having uniform distribution like samples in all sub-domains renders the integration complexity different.

**Limitations:**

None noted.

---

> ### Author Rebuttal · Authors · 2023-08-09
>
> We sincerely thank the reviewer for the insightful feedback on our work. Below are our responses to the questions raised in the review:
>
> 1. Relation between our work and classical non-parametric theory
>
> Regarding a missing discussion on the classical non-parametric theory of estimating functionals via noisy data samples, we would like to mention that previous work on non-parametric estimation of functionals have been cited and discussed in line 97-105 of the paper. Below we also provide a more detailed discussion on the relation between our work and previous work on the estimation of nonlinear functionals (moments).
>
> We remark that our work is the first to consider the case when the underlying function $f$ is in the Sobolev space $W^{s,p}(\Omega)$, where $s < \frac{d}{p}$. This implies that $f$ is not embedded in $L^{\infty}(\Omega)$ and may have a spike. In contrast, previous work all assumed that the underlying function is sufficiently smooth.
>
>
> Secondly, the technique used in our proof of the information-theoretic lower bounds is a bit different from that of previous work [1,2]. In one case of our proof, we pick the two priors to be two discrete distributions of functions such that any function’s sign on each grid in the divided domain is a discrete random variable whose distribution depends on $n$ (the number of data points). This enables us to calculate the amount of information required to distinguish between the two hypotheses even when there is no observational noise. We think that the technique we used here might be useful for the nonparametric statistics community.
>
> In addition, our proof of the upper bound illustrates how the idea of doubly robust estimation can help us design estimators to achieve minimax optimality when there’s no observational noise. Under the other regime when observational noise exists, however, the behavior is completely different. As the influence function is not exactly zero in our estimator, we use Sobolev Embedding Theorem to embed the influence function $f^{q-1}$ and the error function $f-\hat{f}$ into different $L^p$ spaces, which yields the desired upper bound. For the sake of completeness, we list a simplified version of the Sobolev Embedding Theorem below:
>
> For any non-negative integer $s$ and $1 \leq p < r \leq \infty$ satisfying $\frac{1}{p}-\frac{s}{d} = \frac{1}{r}$, we have $W^{s,p}(\Omega) \subset L^r(\Omega)$, $\textit{i.e,}$ every sobolev function in $W^{s,p}$ is also a $L^r$ function.
>
> 2. Design of points
>
> As a response to the question on the design of points, we would like to point out that if the given density supported on the unit cube $\Omega$ is both upper bounded and lower bounded by fixed constants, we can claim that the convergence rate remains unchanged and minimax optimal. However, the difficult part is that we don't have any prior information about the distribution from the given data points.
>
> Moreover, the design of points is in fact related to one important take home message emphasized in our work. Specifically, the minimax optimal convergence rate derived in our paper matches that of the Quasi-Monte Carlo method. From the perspective of information theory, our result reveals that a set of uniformly sampled points doesn’t lead to any loss in information compared to a set of carefully chosen points. For the Quasi-Monte Carlo method, most of the computational resources are used to locate the positions of desired points. After the points are picked, we may simply take the sample average of the function’s values on those points as our estimation. However, when we choose to use a set of uniformly sampled points, we no longer need to spend resources locating the desired points. The tradeoff is that our algorithm for estimation is more complicated, which leads to an extra computational cost.
>
> As a side-note, we also provide some intuitive explanations that may help people understand the tradeoff between algorithms and design of points in Reproducing Kernel Hilbert Space (RKHS). Authors of [3,4] showed that one can compress the data points to present functions in RKHS with convergence rate $n^{-1}$. Once the data points are selected, one only need to take an expectation over the function values at the selected points to estimate the function's integral. In our algorithm, we first use half of the data to run kernel regression, which has a convergence rate of $n^{-\frac{1}{2}}$. Then we use the Monte Carlo method to estimate the error between the original function and the result given by kernel regression. Thus, the convergence rate of our algorithm is also $n^{-1} =n^{-\frac{1}{2}} * n^{-\frac{1}{2}}$, where the first $n^{-\frac{1}{2}}$ is the magnitude of the error induced by kernel regression and the second corresponds to the convergence rate of the Monte-Carlo Method.
>
> We will take all the suggestions given by the reviewer into account while revising our manuscript. Finally, we would like to thank the reviewers again for their time and efforts, which help us improve the quality of our manuscript.
>
> [1] Han, Y., Jiao, J. and Mukherjee, R., 2020. On estimation of $l_r$-norms in Gaussian white noise models. Probability Theory and Related Fields, 177(3-4), pp.1243-1294.
>
> [2] Tsybakov, A.B., 2004. Introduction to nonparametric estimation, 2009. URL https://doi.org/10.1007/b13794.
>
> [3] Dwivedi, R. and Mackey, L., 2021. Kernel thinning. arXiv preprint arXiv:2105.05842.
>
> [4] Dwivedi, R. and Mackey, L., 2021. Generalized kernel thinning. arXiv preprint arXiv:2110.01593.

---

### Decision · Program_Chairs · 2023-09-21

**Decision:**

Accept (poster)

**Comment:**

This paper considers estimating moments of a Sobolev function with control variates adjusted by regression. It shows that the resulting estimator satisfies an information theoretic lower-bound, and thus is in a sense unimprovable. Reviewers agreed the paper makes a useful contribution. However, there were also many comments from reviewers on presentation and "story" that might make the paper more accessible and increase impact. The authors are encouraged to take advantage of these.